# The role of plume-scale processes in long-term impacts of aircraft emissions

**Thibaud M. Fritz, Sebastian D. Eastham, Raymond L. Speth, and Steven R.H. Barrett**

Laboratory for Aviation and the Environment, Department of Aeronautics and Astronautics, Massachusetts Institute of Technology, Cambridge, MA 02139, USA

*Correspondence to:* S.D. Eastham (seastham@mit.edu)

**Abstract.** Emissions from aircraft engines contribute to atmospheric $NO_x$, driving changes in both the climate and in surface air quality. Existing atmospheric models typically assume instant dilution of emissions into large-scale grid cells, neglecting non-linear, small-scale processes occurring in aircraft wakes. They also do not explicitly simulate the formation of ice crystals, which could drive local chemical processing. This assumption may lead to errors in estimates of aircraft-attributable ozone production, and in turn to biased estimates of aviation's current impacts on the atmosphere and the effect of future changes in emissions. This includes black carbon emissions, on which contrail ice forms. These emissions are expected to reduce as biofuel usage increases, but their chemical effects are not well captured by existing models.

To address this problem, we develop a Lagrangian model that explicitly models the chemical and microphysical evolution of an aircraft plume. It includes a unified tropospheric-stratospheric chemical mechanism that incorporates heterogeneous chemistry on background and aircraft-induced aerosols. Microphysical processes are also simulated, including the formation, persistence, and chemical influence of contrails. The plume model is used to quantify how the long-term (24-hour) atmospheric chemical response to an aircraft plume varies in response to different environmental conditions, engine characteristics, and fuel properties. We find that an instant dilution model consistently overestimates ozone production compared to the plume model, up to a maximum error of ~200% at cruise altitudes. Instant dilution of emissions also underestimates the fraction of remaining $NO_x$, although the magnitude and sign of the error vary with season, altitude, and latitude. We also quantify how changes in black carbon emissions affect plume behavior. Our results suggest that a 50% reduction in black carbon emissions, as may be possible through blending with certain biofuels, may lead to thinner, shorter-lived contrails. For the cases that we modeled, these contrails sublimate ~5 to 15% sooner and are 10 to 22% optically thinner. The conversion of emitted $NO_x$ to $HNO_3$ and $N_2O_5$ falls by 16% and 33% respectively, resulting in chemical feedbacks that are not resolved by instant-dilution approaches. The persistent discrepancies between results from the instant dilution approach and from the aircraft plume model demonstrate that a parametrization of effective emission indices should be incorporated into 3-D atmospheric chemistry transport models.

## 1 Introduction

Worldwide air passenger traffic is projected to grow at an annual rate of 5% over the next two decades (Airbus, 2017; Boeing, 2017). Commercial aviation fuel usage has continuously increased (Mazraati, 2010) as demand for air transport has outpaced improvements in efficiency (Lee et al., 2001). Combined with difficulties in reducing emissions of pollutants such as nitrogen oxides ($NO_x$) from aircraft engines, aviation has a unique and growing influence on the chemical composition of the atmosphere.

The release of chemically reactive substances from aircraft exhausts induces perturbations in the environmental chemical balance that can persist for days (Meijer, 2001). Additionally, aviation is a unique sector in terms of its environmental challenges as it is the most significant anthropogenic source of pollution at high altitude (8-12 km). In 2015, an estimated 240 Tg of jet fuel were burned for commercial aviation according to the global inventory from the FAA Aviation Environmental Design Tool (AEDT). For comparison, even under a very conservative assumption - that every rocket launch

in 2015 was performed with the high-capacity, kerosene-burning "Falcon Heavy" - we estimate that rockets burned at most 11 Gg of fuel below the stratopause in that year. Nitrogen oxides ($NO_x = NO + NO_2$) released from aircraft engines have been estimated to increase ozone concentrations in the Northern hemisphere by 2 to 9% (Penner, 1999; Schumann, 1997; Brasseur et al., 1996), while the ice clouds that form in aircraft exhausts ("contrails") have been estimated as having climate impacts of the same order of magnitude as the carbon-dioxide released in the plume (Kärcher, 2018).

The chemical effects of these emissions are typically simulated using global, Eulerian, 3-D atmospheric chemistry transport models. These models simulate aircraft exhaust as being released instantaneously into homogeneously-mixed grid cells that are orders of magnitude larger than the aircraft plume (Brasseur et al., 1998; Meijer et al., 2000; Eyring et al., 2007). This approach does not explicitly capture the high initial species concentrations within the plume, including the effects of non-linear chemistry in the early stages or the formation (and chemical effects) of aerosols and ice crystals (i.e. contrails) in the exhaust plumes. Shortly after release into the atmosphere, species concentrations in the aircraft plume can be several orders of magnitude larger than their background levels. $NO_x$ concentrations at cruise altitude can exceed values up to 20 ppbv in the early stages of the plume, whereas background $NO_x$ levels are typically between 0.007 ppbv and 0.15 ppbv in flight corridor such as the North Atlantic Flight Corridor (NAFC) (Schumann et al., 1998).

The impact of plume-scale modeling of aircraft wakes has been investigated over the past few decades mostly for its relevance to the environmental impact of aviation (Hidalgo, 1974; Thompson et al., 1996). Paoli et al. (2011) extensively covers the different approaches adopted to account for plume scale effects. They also list previous efforts to incorporate plume-scale processing of aircraft emissions into global chemistry-transport models. Prior studies have explicitly modeled the gas-phase components of the plume and have shown that the "instant dilution" approach results in inaccurate estimation of the plume's chemical effects on the environment (Petry et al., 1998; Kraabøl et al., 2000; Cariolle et al., 2009; Huszar et al., 2013). Furthermore, the effects of interactions between contrail ice and the plume chemistry - including as a surface for rapid heterogeneous chemistry - have not yet been quantified.

Field measurements over the past decades, such as the SUCCESS (Toon and Miake-Lye, 1998), POLINAT (Schumann et al., 2000) and SULFUR experiments (Schumann et al., 2002), have measured the microphysical characteristics of both liquid aerosol and ice particles (contrails) in aircraft plumes. Contrail modeling efforts based on these measurements have shown that these aerosols are sensitive to ambient relative humidity, fuel sulfur content, and the amount of emitted solid particles (Kärcher, 1998; Wong and Miake-Lye, 2010). In the early stages, non-volatile aerosols take up a significant amount of the emitted water vapor through con-densation and heterogeneous freezing, potentially leading to the formation of liquid aerosols and ice crystals. During the plume expansion regime, gas species react and diffuse, potentially reacting with one another through heterogeneous chemistry on their surface. This suggests that the formation of ice in aircraft exhausts may result in additional chemical processing that is not captured in either global atmospheric models or gas-phase aircraft plume models.

This gap also affects assessment of new fuels for aviation. Biofuels have been identified as an option to reduce aviation's climate impacts by reducing the net contribution of aviation to atmospheric $CO_2$. However, several of these alternative fuels are also expected to produce less black carbon (Speth et al., 2015) and to have a lower sulfur content (Gupta et al., 2010; Rojo et al., 2015). The effect that these changes will have on aircraft plume chemistry and contrail evolution - and therefore on the total environmental impact of aircraft emissions - depend on the microphysical response of the plume. As such, the atmospheric effects of changing from conventional jet fuel to alternative fuels are not yet fully understood. In the following, soot emissions are identified as black carbon emissions.

To address these issues we develop the Aircraft Plume Chemistry, Emissions, and Microphysics Model (APCEMM). APCEMM is applied under a variety of conditions to simulate the influence of changes in environment, aircraft characteristics, and fuel properties on in-plume chemistry and aerosol size distribution. Finally, the effects of these changes are presented in terms of their impact on large-scale properties such as net 24-hour ozone production, end-of-lifetime $NO_x$ partitioning, and contrail optical thickness.

## 2 Methods

We first describe the overall modeling approach used by APCEMM to simulate the chemistry and physics of an aircraft plume (Section 2.1). Sections 2.2 and 2.3 describe the details of the different models used for the initial and mature plume evolution phases, respectively. Finally, Section 2.4 describes the experimental design used to determine the overall impact of plume-scale processes on long-term aircraft emissions impacts. In this paper, APCEMM is used to quantify the role of non-linear plume chemistry and to obtain a first estimate of contrail impacts on chemistry. Estimating precisely the interdependence of contrail microphysics and chemistry would require expensive Large Eddy Simulations (LES). Using simplifying assumptions regarding plume dynamics, APCEMM aims to bridge the gap between Gaussian plume models and LES.

## 2.1 Model overview

APCEMM models the growth and chemical evolution of a single aircraft plume. Chemical concentrations and aerosol

characteristics are calculated for a 2-D cross-section of the plume, perpendicular to the flight path. Dynamics, chemistry, and microphysics are explicitly modelled within the plume, using two different approaches depending on the age of the plume.

Observations and high-resolution modeling of aircraft wakes has shown three dynamical regimes in the first few minutes after emission, before the wake develops into a "mature" plume. Typical timescales and dilution ratios for an aircraft plume are shown in Table 1.

During the initial "early jet" and "jet" regimes, compressibility effects arise from the momentum-driven jet that last for a short amount of time of the order of a few seconds (Kärcher, 1995). After ~10 seconds the wing-tip vortices have formed and begin to affect the emissions plume. During this "vortex regime", the counter-rotating vortex pair causes the plume to descend by distances of the order of several hundred meters (Kärcher, 1995; Schumann, 2012). The wake is prone to instabilities triggered by atmospheric or aircraft-induced turbulence. Crow instability can occur in aircraft plumes and cause small sinusoidal distortions in the vortex shape to be amplified (Paoli and Shariff, 2016). These enhanced oscillations cause the vortex system to collapse (Naiman et al., 2011). We do not explicitly simulate these processes, instead treating the early-plume as well-mixed.

Over the period of these three initial regimes, the plume cools rapidly to ambient temperatures (~220 K) from an initial temperature of 500-600 K, leading to a spike in ice and liquid water saturations approximately 100 ms after emission and triggering a range of microphysical processes (Kärcher et al., 2015). During this period, formation of sulfate aerosols, freezing on solid nuclei, condensation, heterogeneous nucleation, and coagulation also occur. Homogeneous freezing is not included. Previous studies have suggested that homogeneous freezing is unlikely in aircraft plumes given the number of pre-existing nuclei (Wong and Miake-Lye, 2010). This is because combustion particles can acquire an ice coating at temperatures much higher than cruise temperatures, implying that ice crystals formed in the vicinity of the engines freeze by virtue of heterogeneous nucleation. In APCEMM, the plume is assumed to be well mixed during these first three regimes - the "early plume phase" (Section 2.2). We model this early plume as a uniform, well-mixed air mass evolving through time. In the following, we refer to this early-plume representation as a "box model".

The output of this box model is then provided as the initial condition for the model of the long term diffusion regime (Section 2.3). This regime begins when the aircraft-induced vortices break apart. In this regime, the plume expands in ambient air. The rate of diffusion is controlled by the vertical stratification of the atmosphere and by the vertical gradient of the wind speed (wind shear). Unlike the early plume phase, spatial heterogeneity of the plume is explicitly accounted for in APCEMM during the diffusion regime, allowing for

cross-plume concentration gradients. For the first hour of this regime, we simulate an upward motion of the plume. This is because the vortex sinking, modeled as a simple vertical displacement, results in adiabatic compression of the plume. In a stably-stratified atmosphere, this causes the plume to be warmer than its surroundings. The resulting buoyancy and radiative imbalance causes the plume to rise back to its original emission altitude, which we simulate as taking place over a one-hour timescale (Heymsfield et al., 1998).

## 2.2 Modeling of the early plume

During the early plume phase, the plume is treated as a single, well-mixed air mass. The air mass grows, dilutes, and cools through turbulent mixing with ambient air (entrainment). It also sinks and heats up due to the effect of the aircraft wing-tip vortices (vortex sinking). Throughout this phase we simulate rapid chemical changes, including the formation of liquid and solid aerosols.

### 2.2.1 Dilution and temperature evolution of the early plume

In the jet and vortex regimes, we adopt a formulation similar to the box model used in Kärcher (1995). The rate of change of chemical concentrations within the plume is dominated in this regime by dilution due to turbulent mixing. The contribution of wake mixing is approximated as a first-order decay term proportional to a time-dependent entrainment rate, i.e. $\omega_C(t)$.

$$\left(\frac{DC_k}{Dt}\right)\bigg|_{\text{mix}} = -\omega_{C_k}(t)(C_k - C_{\text{Amb},k}), \tag{1}$$

where $C_k$ is the molecular concentration of species $k$.

This entrainment rate agrees with the experimental data and curve fit provided in Schumann et al. (1998) for times greater than 1 s. $C_{\text{Amb},k}$ is the ambient molecular concentration of species $k$ and is assumed to be constant during the jet and vortex regimes considering that the timescale associated with gas-phase chemistry is much greater than the time taken to reach the diffusion regime. We assume that, during this time, gas-phase chemistry does not influence the concentrations of most species within the plume. The only exceptions are conversion of S(IV) to S(VI) and $NO_x$ to HONO and $HNO_3$ as described in Section 2.2.2, which evolve on similar timescales due to high initial concentrations (Kärcher et al., 1996).

The temperature of the plume during this initial phase is controlled by two processes. Firstly, prior experiments have shown that typical Lewis numbers are close to unity in coaxial jets, such as aircraft plumes (Forstall and Shapiro, 1950). Mixing of cold air with the hot exhaust stream (cooling the plume) is therefore assumed to occur at the same rate as entrainment of ambient chemical species. Secondly, the downward motion induced by the wing tip vortices also causes the

**Table 1.** Plume timescales and dilution ratios (Kärcher, 1995). Dilution ratios are the ratio of the initial plume air mass to the air mass at the target time.

| Phases | Early-plume model | | | Long-term plume model |
|---|---|---|---|---|
|  | Early Jet Regime | Jet Regime | Vortex Regime | Diffusion Regime |
| Timescale | 0.1 s | 10 s | 100 s | Few hours up to a day |
| Dilution ratio at end of phase [-] | $5.5 \times 10^{-1}$ | $2.6 \times 10^{-3}$ | $9.9 \times 10^{-4}$ | $< 1.0 \times 10^{-4}$ |

air to heat up adiabatically, independently of the local lapse rate (Unterstrasser et al., 2008). The plume temperature evolution is therefore expressed as the sum of a positive heating due to vortex sinking and a first-order decay term representing entrainment, i.e.

$$\frac{dT_p}{dt} = \Gamma_d v_z - \omega_T(t)(T_p - T_{\text{Amb}}(z)), \quad (2)$$

In the present section, units are provided for the purpose of demonstration only, and are not fundamental to the formulae. In Equation 2, $T_p$ is the plume temperature in K, $\Gamma_d$ the adiabatic lapse rate expressed in K/m, $v_z$ the vertical velocity in m/s of the plume and $T_{\text{Amb}}$ the ambient temperature in K evaluated as a function of the ambient lapse rate, which has been obtained as a function of latitude and altitude from monthly-averaged meteorological data obtained from the Modern-Era Retrospective analysis for Research and Applications, Version 2 (MERRA-2). The plume acquires a vertical motion during the vortex regime such that $v_z$ is assumed to be non-zero only between the time at which the vortices start inducing a vertical displacement of the plume and the vortex break-up time. The wake vortex sinking is computed according to a parametric formulation described in Schumann (2012) from which we evaluate the mean downward displacement as a function of aircraft and ambient atmospheric characteristics. This parameterization of the aircraft-induced mixing does not allow us to accurately capture the vortex dynamics and neglects the spatial heterogeneity that could arise. The inclusion of a more accurate representation of the early-plume dynamics in APCEMM is an area for future development.

### 2.2.2 Chemical conversions in the early plume

In the early stages of the plume, oxidation of NO and $NO_2$ results in the formation of HONO and $HNO_3$. As described in Kärcher (1999), conversion efficiencies of NO, $NO_2$ and

**Table 2.** Assumed conversion efficiencies of $NO_x$ and $SO_2$ to secondary species

| NO | $\rightarrow$ | HONO | 1.5% |
|---|---|---|---|
| $NO_2$ | $\rightarrow$ | $HNO_3$ | 4.0% |
| $SO_2$ | $\rightarrow$ | $H_2SO_4$ | 0.5% |
| OH | $\rightarrow$ | $H_2O_2$ | 2.0% |

$SO_2$ depend on the exit plane hydroxyl radical concentration. Tremmel et al. (1998) inferred initial OH concentrations at the combustor and engine exit through measurements of NO, HONO, $HNO_3$ as well as $CO_2$ to account for plume dilution. Their results indicate that the OH emission index ranges between 0.32 and 0.39 $g/kg_{\text{fuel}}$ for the JT9D-7A, which corresponds to an engine exit mixing ratio lying between 9.0 and 14.4 ppmv. Conversion efficiencies used in APCEMM are depicted in Table 2. Even though the conversion efficiencies remain of the order of a few percent, they increase monotonically with the OH engine exit mixing ratio, as more radicals are available for the following reactions (Kärcher et al., 1996).

$$NO + OH \rightarrow HONO$$
$$NO_2 + OH \rightarrow HNO_3$$
$$OH + OH \rightarrow H_2O_2$$

Oxidation of S(IV) to gaseous S(VI) is not simulated during this period. This process mostly occurs in the engine's turbines and only a negligible fraction is converted in the young aircraft plume (Lukachko et al. (1998); Tremmel and Schumann (1999)).

### 2.2.3 Microphysical representation of the early plume

As the plume cools down and mixes with ambient air, aerosols begin to form, supplementing those that were emitted directly from the engine (e.g. black carbon particles). This both modifies the local chemical concentrations and changes the initial aerosol size distribution during the second phase of the plume. Four microphysical processes are explicitly considered: freezing of liquid particles into solid ones, condensation of gas onto liquid particles, nucleation of new liquid particles, and coagulation of both solid and liquid particles.

We first consider growth of an existing particle population. The mathematical formulation is given in detail in Appendix A, but is covered briefly here. The microphysical model for growth of ice particles is adapted from Kärcher (1998). According to Kärcher and Yu (2009) and *in situ* measurements of black carbon number emissions at cruise altitude from Petzold et al. (1999), the plume is in a "soot-rich" regime, favoring freezing of water around black carbon cores rather than freezing of liquid and ambient particles. We thus assume that solid particles (black carbon and metal) emitted by the aircraft serve as condensation nuclei for water vapor. Under supersaturated conditions, deposition induces ice crystal growth, depleting gaseous water vapor. During this initial phase, ice crystals are treated as mono-disperse (single size) and are considered spherical. Under these assumptions, we need only consider the growth of a single "representative" particle, rather than analyzing the population as a whole. Because of the low ambient temperatures, water that condenses is assumed to freeze instantaneously, such that ice crystals grow by deposition of water molecules onto their surface. The rate of change in the ice mass of a particle, $m_p$, is then given by

$$\frac{dm_p}{dt} = H_p^{\text{act}}(m_p) \times 4\pi C_p D_{\text{v,eff}} \left( P_{\text{H}_2\text{O}} - P_{\text{H}_2\text{O}}^{\text{sat}} \right), \qquad (3)$$

where $H_p^{\text{act}}$ is a function accounting for nucleus activation (equation A3), $C_p$ is the ice crystal capacitance (equal to the particle radius $r_p$ for spherical nuclei), $D_{\text{v,eff}}$ is the effective water vapor diffusion coefficient in air (equation A4), and $P_{\text{H}_2\text{O}}$ the water partial pressure. Assuming that each ice particle is nucleated on a black carbon particle with a dry radius of 20 nm, and using a fixed mass density for ice of 916.7 kg/m$^3$, this calculation also gives the rate of change of radius of solid particles in the plume. Calculation of each of the terms in equation 3 is described in Appendix A1.

Black carbon and ice particles can also grow by condensation of water vapor, sulfuric acid, and nitric acid into a partial liquid surface layer. The growth of this layer is related to the condensation (or evaporation) rate of H$_2$O, H$_2$SO$_4$ and HNO$_3$, calculated as

$$\frac{dN_{k,p}}{dt} = 4\pi r_p D_k \beta(r_p) \left( \frac{P_k - P_k^{\text{sat}}}{k_B T} \right) \times \theta, \qquad (4)$$

where $N_{k,p}$ is the number of molecules of type $k$ on a particle of type $p$, $D_k$ is the gas diffusivity in m$^2$/s, and $P_k$ and $P_k^{\text{sat}}$ are the partial and saturation pressures of species $k$, respectively, expressed in Pa. The function $\beta$ accounts for changes in uptake in different gas regimes, and is described in equation A5. Experimentally-derived deposition coefficients for heteromolecular condensation, used in the calculation of $\beta$, are taken from Kärcher (1998). On black carbon particles, $\theta$ describes the fractional surface coverage of the particle liquid coating and is calculated according to Kärcher (1998). For all other particles, this limitation is ignored and $\theta$ is set to 1. Gas diffusivities for H$_2$SO$_4$ and HNO$_3$ are taken from Tang et al. (2014).

Similar to sulfur, organic compounds in the upper troposphere have been found to alter the freezing behavior of aerosols and the black carbon coating fraction, $\theta$, even in natural conditions (Cziczo et al., 2004; Kärcher and Koop, 2005; Murray et al., 2010). In aircraft plumes, the formation of condensable organic species originates in the production of electrically charged clusters (chemi-ions) (Kärcher et al., 2015). These organic compounds have been found to be either aqueous aerosols or soluble in aqueous H$_2$SO$_4$ solutions (Yu et al., 1999; Kärcher et al., 2015). Their high solubility makes organic matter a prime contributor to the mass of ultrafine plume particles and could also enhance the black carbon particle coating (Rojo et al., 2015). Previous studies estimated the mass of particulate organic matter in aqueous form to be approximately 20 mg/kg$_{\text{fuel}}$ (Kärcher et al., 2000). The theory behind the role of organics on particle growth and their chemical speciation is still limited. In most of the experiments described in this paper we neglect the role of organic aerosol. However, as a sensitivity study, we estimate the effect of organic matter on the fractional coating by prescribing an initial fraction of coated black carbon (Section 3.1). We also perform simulations in which we calculate the effect of changes in black carbon emissions. These simulations can be considered as a proxy for the effect of organic aerosol if the aerosols are capable of acting as ice nuclei (Section 3.1.3).

In addition to growth of existing particles, new liquid particles can form through binary homogeneous and heterogeneous nucleation. Several nucleation parameterizations have been established to simulate binary homogeneous nucleation in a sulfur-rich environment (Jaecker-Voirol and Mirabel, 1989; Napari et al., 2002; Vehkamäki et al., 2002). Jung et al. (2008) have computed different sensitivities using these models and provided further validation of the models cited previously, comparing the results to field measurements. Given the range of ambient conditions relevant to an aircraft plume, we calculate cluster size, composition, and nucleation rate using the parameterization from Vehkamäki et al. (2002). While this model is only considered valid between 230.15 K and 305.15 K, we expect that most nucleation of fresh sulfate aerosol will occur while the plume is still cooling down, within this temperature range. Liquid aerosols are assumed to remain liquid throughout the plume lifetime. Previous stud-

ies (e.g. Kärcher, 1998; Tabazadeh et al., 1997) have quanti-
fied the freezing behavior of sulfate aerosols and liquid sulfur
coating at low temperatures and found that freezing of sul-
fate aerosols requires an ice supersaturation of about 1.5 at
210 K. Additionally, Kärcher et al. (1998) conclude that het-
erogeneous freezing on coated black carbon particles drives
the contrail formation phase. We thus neglect the freezing of
sulfate aerosols similarly to Wong and Miake-Lye (2010).

The number concentration of aerosol particles in the plume
can also change through coagulation, as emitted and en-
trained particles collide and coalesce. During the early plume
phase, we consider only the coagulation of liquid aerosols,
and the scavenging of liquid aerosols by ice and black carbon
particles. Self-aggregation of ice and black carbon particles
on the time scale of the early plume is assumed to be negligi-
ble. Since all aerosols during this phase are likely to be small,
all collisions are assumed to result in coagulation (a coa-
lescence efficiency of 1) (Jacobson, 2011). Particle breakup
and shattering is neglected for the same reason (Beard and
Ochs III, 1995; Jacobson, 2011). The effect of coagulation
on the number concentration of aerosols in size bin $k$, cover-
ing the size interval $[r_k, r_{k+1}]$, is modeled as

$$\frac{dn_k}{dt} = \frac{1}{2} \sum_{j=1}^{k-1} K_{j,k-j} n_j n_{k-j} - \sum_{j=1}^{+\infty} K_{k,j} n_k n_j, \qquad (5)$$

where $n_k$ is the number density of particles in bin $k$ and $K_{i,j}$ is
the coagulation kernel appropriate to collisions between size
bins $i$ and $j$, which represents the physics of the problem. A
full description of the coagulation kernel and its calculation
is given in Appendix B. Equation (5) states that the rate of
change in the number density in bin $k$ corresponds to the rate
at which smaller particles of size $k - j$ coagulate with parti-
cles of size $j$ minus the rate at which the particles of size $k$
are lost due to coagulation with particle of all sizes.

During the early plume phase, liquid aerosols are modeled
using 64 size bins, from a minimum radius of 0.1 nm to a
maximum of 0.5 μm. Ice and black carbon aerosols are con-
sidered to have a single size, as estimated based on equa-
tions (3) and (4). Instead of solving equation (5) directly
for every size bin, aerosol coagulation is computed using a
semi-implicit, non-iterative, volume-conserving and uncon-
ditionally stable numerical scheme described in Jacobson
et al. (1994). This model has been used extensively in aerosol
modeling and aircraft plume simulations (Paoli et al., 2008).
The rate of particle coagulation peaks shortly after emis-
sion and then significantly reduces as entrainment of ambient
air into the plume decreases the number of aerosol particles
present per unit volume of air.

The number and size of the aerosol particles present at the
end of the early phase is used to provide the initial conditions
for the mature plume phase, with one adjustment. The down-
ward movement induced by the aircraft wake vortices (Unter-
strasser et al., 2008) increases the depth of the contrail, while
adiabatic heating and turbulent temperature fluctuations re-

sult in crystal losses through sublimation. These losses are
represented using a survival fraction, which we compute us-
ing a parameterization based on large eddy simulations (Un-
terstrasser, 2016). This survival fraction is typically of the or-
der of 0.5, such that the initial aerosol population for the ma-
ture plume phase includes roughly half the number of aerosol
particles as were present at the end of the early phase.

## 2.3    Modeling of the mature plume

Following breakdown of the wingtip vortices, the plume is
considered to enter the "diffusion regime". In this regime, the
plume is no longer considered to be well mixed, and diffusion
of chemical constituents becomes important.

In APCEMM, we use an operator splitting method that al-
lows us to treat the chemical kinetics terms separately from
the turbulent diffusion terms, and to apply optimized solution
methods for these different processes. For chemistry calcula-
tions, the domain is represented using a set of fixed concen-
tric elliptical rings (Figure 1). The central ring (semi-major
and semi-minor axis of ~75 m and ~30 m respectively) is ini-
tialized using chemical concentrations and aerosol properties
as calculated at the end of the "early plume" stage (Section
2.2), and after accounting for losses due to vortex sinking.
All other rings are initialized with ambient air. Each ring is
further discretized into a lower and upper half-ring to allow
for vertical variations in temperature, and to account for sed-
imentation of aerosols.

Diffusion and advection of pollutants relative to the plume
centerline (due to wind shear), in addition to sedimentation
of aerosols and buoyant motion, are simulated on a regular,
rectilinear grid with a horizontal and vertical grid spacing
of ~100 m horizontally and ~5 m vertically. Prior to these
"transport" processes, concentrations of constituents in the
rings are mapped to the rectilinear grid. Diffusion, advec-
tion, and settling of the constituents is then simulated using
a spectral scheme (Gottlieb and Orszag, 1977). This scheme
is also used to allow shear to distort the chemical rings. Fol-
lowing transport, the constituents are mapped back to the ring
discretization.

Diffusion of pollutants, chemistry, and aerosol micro-
physics are all explicitly accounted for using a time step-
ping scheme. All processes are simulated using a variable
timestep. During the first 10 minutes, and within 10 minutes
of local sunrise or sunset, the time step is restricted to 30 sec-
onds to ensure that rapid chemical changes are captured. At
all other times, a time step of 5 minutes is used.

### 2.3.1    Diffusion and shear in the mature plume

The rate of diffusion of the plume's constituents is modeled
using directional diffusion coefficients. The degree of diffu-
sion anisotropy is dictated by the Richardson number, a mea-
sure of local atmospheric stability (Dürbeck and Gerz, 1996;
Schumann et al., 1998). In APCEMM, the vertical diffusion

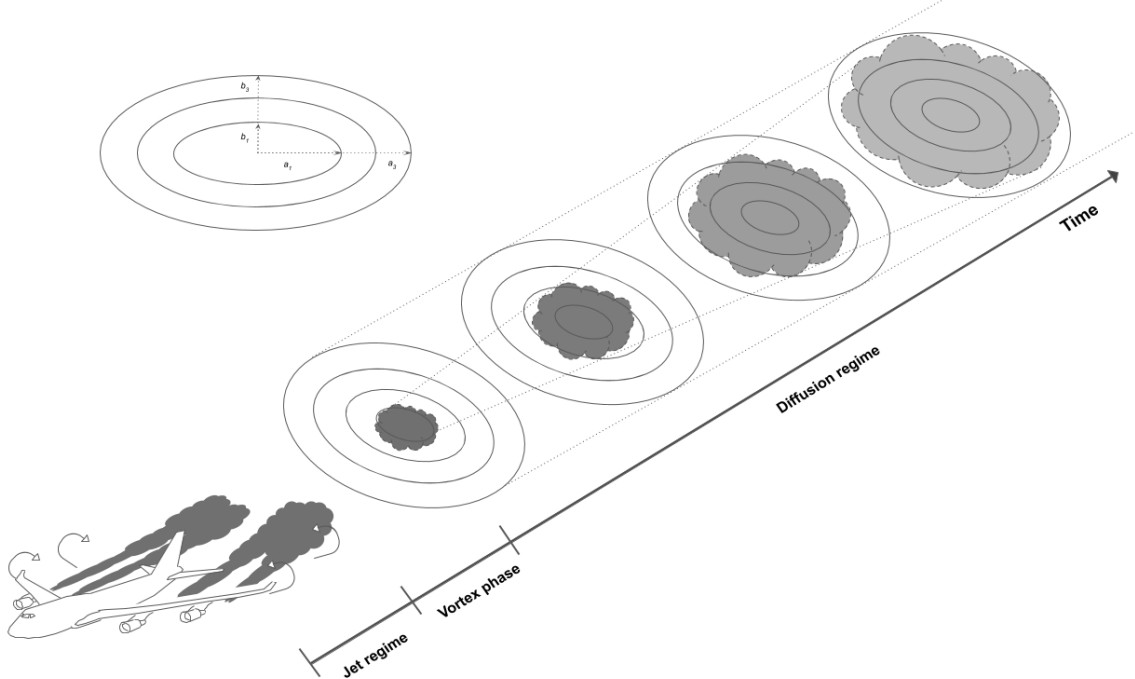

**Figure 1.** Schematic of the discretized ring approach used in APCEMM. The rings' major and minor axis are denoted by $a_i$ and $b_i$ respectively.

coefficient $D_v$ is estimated using the approach from Schumann et al. (1995) using the local Brunt-Väisälä frequency and Richardson number, which is in turn computed from the local wind shear. To account for initial turbulence, $D_v$ is increased to 1.1 m$^2$/s, after which it is reduced exponentially to its background value with an e-folding time of 13 minutes (Kraabøl et al., 2000). The horizontal diffusion coefficient is required as an input to APCEMM. Dürbeck and Gerz (1996) found that, for Richardson numbers above unity, the horizontal and vertical diffusion coefficients lie in the range 15 m$^2$/s $\leq D_h \leq$ 23 m$^2$/s and 0.15 m$^2$/s $\leq D_v \leq$ 0.18 m$^2$/s respectively, which agree with measurements from Schumann et al. (1995). Although computationally efficient, our current representation of the aircraft-induced turbulence is idealized and does not allow us to model the spatial heterogeneity that would arise after the dissipation of the vortex pair. A higher-fidelity approach would be needed to capture the effects of vortex structures on the optical and chemical properties of the plume.

In addition to diffusion across the domain, advection due to wind shear is explicitly accounted for using the same spectral scheme as is applied to simulate diffusion. The horizontal wind speed relative to the plume center line is calculated assuming a constant vertical wind shear across the domain.

Several physical processes can result in not only a vertical displacement but also a vertical stretching of an aircraft plume. Aircraft-induced vortex sinking leads to adiabatic compression and leads to the formation of spatially non-uniform primary and secondary wakes (Paoli and Shariff, 2016). Following the initial descent, the plume rises due to buoyancy. Simultaneously, the plume absorbs upwelling infrared radiation causing it to heat up relatively to the surrounding air (Jensen et al., 1998b). Non-uniformities in the local ice water content lead to a vertical stretching of the contrail. The contrail net heating drives a local plume updraft with speeds of up to 10 cm s$^{-1}$ during the first hour (Heymsfield et al., 1998), consistent with estimates that the contrail core can rise over the first hour after its formation (Jensen et al., 1998b).

In this study, we adopt a simple representation of vertical plume motions. Vortex sinking is represented as a uniform-velocity downdraft over the first few minutes, as described in Section 2.2. We represent the effects of buoyancy by applying an exponentially-decaying velocity profile, with the aim of resolving the vertical updraft. The time constant associated with the local updraft is one hour (Jensen et al., 1998b; Heymsfield et al., 1998), such that an initial vertical speed of 10 cm/s has reduced to 6 cm/s within 30 minutes. Although simulation of large-scale synoptic uplift of the aircraft plume and surrounding air are possible in APCEMM, we do not include them in this paper.

This representation of vertical motions is highly simplified. The focus of this paper is to quantify the effect of plume-scale non-linear chemistry on the atmospheric

impacts of aviation exhaust, including a first assessment of role played by contrail ice. Although APCEMM can also give some insight into the behavior of contrails, it is still an intermediate-fidelity tool between the widely-used Gaussian plume approximation (Schumann, 2012) and computationally-expensive large eddy simulations (Paugam et al., 2010; Sölch and Kärcher, 2010; Unterstrasser and Gierens, 2010a; Picot et al., 2015; Paoli et al., 2017). Future research using APCEMM to investigate the details of contrail evolution would benefit from the implementation of a more sophisticated representation of the plume's movement and internal dynamics.

### 2.3.2 Chemistry in the mature plume

The gas-phase chemistry mechanism is taken from GEOS-Chem v11 (Eastham et al., 2014). Heterogeneous halogen, $N_2O_5$ and $HO_x$ chemistry, as well as formation and evaporation of stratospheric aerosols are considered. Due to their long lifetimes, reactions involving CFCs and HCFCs are neglected. The set of chemical reactions is solved numerically with the Kinetic Pre-Processor (KPP) (Damian et al., 2002). KPP is a software tool, which from a set of chemical reactions and rate coefficients, generates code to integrate the differential equations and compute the time evolution of chemical species with a suitable numerical integration scheme.

### 2.3.3 Aerosol modelling in the mature plume

The aerosol distributions in the mature plume phase are initialized based on the output from the early-plume module. The distribution of sulfate aerosols is unchanged while ice particles are distributed assuming a log-normal distribution, using the mean ice particle radius and a geometric standard deviation of 1.6 (Goodman et al., 1998; Jensen et al., 1998a). The use of a log-normal distribution is based on *in situ* measurements (Schröder et al., 2018) and this assumption has been used in previous work to initialize the contrail ice particle size distribution (Jensen et al., 1998b; Picot et al., 2015).

As the plume expands, the ice crystal size distribution changes due to growth, sublimation, gravitational settling and coagulation. Even without the application of a log-normal distribution, a polydisperse distribution would arise due to coagulation and the different meteorological conditions throughout the plume. Particle growth is treated using a moving-center size structure (Jacobson, 1997). Ice crystal growth is characterized by the "advection" of the particle density distribution across diameter space (Jacobson, 2003).

Ice crystal growth modifies the particle volume but leaves the number of particles constant. Sublimation mechanisms lead to a loss of ice crystals and act as a source of water vapor, modifying the cell's relative humidity and release a dry particle core that is then considered "deactivated" and unable to take up water vapor as ice. The extent of sublimation is moderated by the size of the droplet cores, as larger hydrometeors can persist in subsaturated air. Evaporation and sublimation are both endothermic processes that cool down the surface of an ice crystal. The equilibrium surface temperature is obtained through an iterative process that allows us to compute the particle sublimation rate (Jacobson, 2003).

Aggregation of ice particles uses the same algorithm and the same coagulation kernel described previously for sulfate aerosols. Following the approach from Sölch and Kärcher (2010), we assume a constant aggregation efficiency for ice particles.

Gravitational settling causes the ice particles to fall vertically, thus entering warmer regions. Ice particle terminal velocities are computed according to Stokes law, accounting for the slip correction, as in Pruppacher et al. (1998). The settling velocity of an ice crystal depends on its size, with larger particles falling faster. Different parts of a contrail have different crystal sizes, meaning that they settle at different speeds (Unterstrasser et al., 2016). This differential settling effect is often neglected in reduced-order contrail models.

Finally, liquid (sulfate) aerosols are modeled using the same 64-bin approach as in the early plume phase. In the mature plume phase, the distribution of sulfate aerosols is affected only by coagulation, using the same coagulation kernel as before.

### 2.4 Experimental description

We first present the chemical and microphysical evolution of the plume in the first 15 minutes after emissions (Section 3.1), including the effect of changes in fuel sulfur content (FSC) and the potential role of condensable organic species. These factors are likely to be important for the formation and growth of ultrafine volatile particles and ice crystals in the aircraft wake. In addition, the theory behind the role of organic compounds is still poorly understood but Kärcher et al. (2000) estimate that particulate organic matter is a primary contributor to the mass of volatile particles. In this study, the impact of organic matter on the early evolution of black carbon particles and ice crystal growth is quantified by prescribing an initial fraction of coated black carbon. We also perform simulations to assess the role of the fuel sulfur content and the black carbon number emission index on the surface coating of black carbon particles and the formation of liquid sulfate particles. FSC varies from an average of ~2 to 600 ppm, depending on fuel type (Rojo et al., 2015). A range of black carbon emissions has been measured in a number of different aircraft wakes and has been found to vary by an order of magnitude (11 - 100 mg/kg$_{fuel}$ as estimated by Petzold et al. (1999)). Results for different black carbon emission rates may also provide insight into the effect of organic aerosol, if said aerosol is acting as a nucleation site rather than as a coating for existing black carbon.

We then simulate the chemical evolution of a plume from a single flight using both approaches and compare the results to the literature (Section 3.2). We compute the impact

of changes in background conditions (Section 3.3), engine emissions (Section 3.4), and flight location (Section 3.5). These simulations are intended to both validate the ability of APCEMM to accurately model non-linear plume chemistry and to quantify the extent to which the output of an instant dilution approach differs from that of a fully resolved plume model,

We also perform a set of dedicated experiments to quantify the relationship between different parameters and the behavior of a contrail forming in the plume. This includes the role of relative humidity and contrail updraft velocity (Section 3.6.1), and the effect of changes in black carbon emissions (Section 3.6.2). Finally, we combine these assessments to determine how accounting for contrail ice could directly affect the chemistry of the plume, and how this effect is modified by changes in black carbon emissions (Section 3.6.3). Most global models do not include contrail simulation, so only APCEMM results are provided in these sections.

All plumes are simulated for 24 hours. For typical diffusion parameters of $D_h$ = 15 m²/s and $D_v$ = 0.15 m²/s, this results in 0.03% of emitted material reaching the edge of the computational domain by the end of the simulation.

### 2.4.1 Model setup

Both models, APCEMM and the instant-dilution approach, are initialized with background mixing ratios obtained from a year-long GEOS-Chem simulation. Noontime photolysis rates are retrieved from that same run. Atmospheric background conditions are obtained from a spin-up run over 5 days. Meteorological data for each altitude, latitude, longitude, and time are taken from the Modern-Era Retrospective analysis for Research and Applications, Version 2 (MERRA-2) for 2013. This includes the vertical wind shear and (longitudinally-averaged) Brunt-Väisäla frequency, used for calculation of the vertical diffusion coefficient. This approach provides an upper bound on vertical diffusion, as it overpredicts the diffusion parameter at large Richardson numbers. The probability distribution of the Brunt-Väisäla frequency, Richardson number, and resulting vertical diffusion parameter are given in the SI for different pressure levels. As shown in Dürbeck and Gerz (1996), the Brunt-Väisäla distribution is unimodal in the troposphere and peaks around $N = 0.01$ s⁻¹. A second mode appears in the stratosphere at approximately $N = 0.02$ s⁻¹. Some further analysis shows that 90% of the distribution lies at Richardson numbers greater than 5, indicating weak and/or fast decaying turbulence. The distribution and its support agree with the values in Schumann et al. (1995) and the mean values lie in the range given by Dürbeck and Gerz (1996). A horizontal diffusion parameter $D_h$ of 20 m²/s is assumed for all simulations. Unless otherwise specified, we assume zero wind shear, although the effects of this assumption are investigated. A list of all the input parameters expected by APCEMM is provided in Table S1 of the Supplementary Information.

All the cases in this analysis consider emissions from a B747-8 equipped with GEnx engines, which are released in the innermost ring, with a cross-sectional area of 6,000 m². The representation of the GEnx engines in APCEMM at cruise altitude uses the equations of the Boeing Fuel Flow Method 2 (DuBois and Paynter, 2006) to compute a cruise $NO_x$ emission index from the ones provided by the ICAO Engine Emissions Databank. For black carbon emissions, we use the "$SN - C_{BC}$" method described in Stettler et al. (2013), equation (5) in their paper. The black carbon mass emission index varied from 10 to 14 mg/kg$_{fuel}$ for this particular engine except in Sections 3.1.3 and in 3.6.2, 3.6.3 and 3.6.4 where we prescribe different black carbon emission indices representing the heterogeneity in current cruise soot emissions. The emission rates for black carbon particles or "number emissions indices", $EI_{\#}$, are derived from the mass indices $EI_m$ as

$$EI_{\#} = \frac{3EI_m}{4\pi\rho_{BC}r_{BC}^3}\exp(-\frac{9}{2}\ln^2(\sigma_{BC})), \qquad (6)$$

where $\rho_{BC}$, $r_{BC}$ and $\sigma_{BC}$ are the black carbon mass density, median radius and geometric standard deviation. For all the cases described in this paper, we use a black carbon median radius of 20 nm and a geometric standard deviation equal to 1.6. This means that changes in the overall mass of black carbon emitted are modelled as a change in the number of particles.

The fuel sulfur content is kept constant at 500 ppm throughout all cases, except in Section 3.1.2 where we quantify the effect of changes in fuel sulfur content on the early-plume black carbon fractional coating and liquid sulfate aerosol formation and evolution. Analysis of results using other aircraft/engine combinations is a future research opportunity.

### 2.4.2 Metrics of the chemical response

To evaluate the error resulting from neglecting non-linear plume chemistry, we compare results generated using the instant dilution approach and using APCEMM. The discrepancies between both models are first compared in terms of total ozone mass per unit length in flight direction (kg/km). In addition, the conversion of short-lived nitrogen oxides to reservoir species affects long-term ozone production, heterogeneous chemistry, particle formation and/or growth, with known long-term impacts on air quality (Eastham and Barrett, 2016). The evolution of nitrogen partitioning is therefore computed for both models.

The total emitted $NO_y$ ($E_{NO_y}$) is a conserved quantity throughout the plume lifetime and is equal to the plume-integrated $NO_y$ perturbation $\left(E_{NO_y} = \Delta\left(NO_y\right) = \iint_A \left(\left[NO_y\right] - \left[NO_y\right]^{Amb}\right) dA\right)$, where the notation $[\cdot]^{Amb}$ refers to ambient conditions. At

any given instant $\Delta\left(NO_y\right)(t) = \Delta\left(NO_x\right)(t = 0)$, which is proportional to the $NO_x$ emission index. Averaging the perturbation due to aircraft emissions allows us to compute the time-dependent chemical conversions from one species to another. The emission conversion factor of species X, $ECF_X$, is then defined as:

$$ECF_X(t) = \frac{1}{E_{NO_y}} \iint\limits_A \left([X](t) - [X]^{Amb}(t)\right) dA. \tag{7}$$

The emission conversion factor quantifies how many moles of species X are obtained for one mole of emitted $NO_y$. For ozone, this is similar to the ozone production efficiency (OPE), although the ECF is time-dependent and does not include ozone that has been produced and later destroyed. Given that nitrogen oxides are converted to reservoir species over the plume lifetime, $ECF_{NO_x}$ decreases with time.

We use the ECF to quantify the discrepancy between the two approaches. For a species X, we define the error as the difference in ECF after 24 hours:

$$\varepsilon_X = ECF_X^{Box}(t = 24\text{ hrs}) - ECF_X^{APCEMM}(t = 24\text{ hrs}). \tag{8}$$

A positive error means that instant dilution of aircraft emissions overestimates the chemical production of species X compared to the aircraft plume model.

Evaluation after 24 hours ensures that the domain is in the same photochemical state as at initialization. This guarantees that we make a fair comparison for photochemically-active species. However, the plume may still be sufficiently concentrated that adding it to a grid cell in a larger simulation may still result in misrepresentation of plume chemistry. Additional work will be needed to quantify the magnitude of this error if plume processing is embedded into a global-scale model.

## 3   Results

### 3.1   Results of the early-plume phase

In this section, we model the evolution of the plume in the first 15 minutes after initial emission. In addition to chemical effects, we quantify the role that organic matter, fuel sulfur content, and black carbon emissions have on the early size distribution of aerosols in the plume.

In these simulations, the background air is initially sub-saturated with respect to water at a temperature of 215 K. As described in Section 2.2.3, organic matter influences the total mass of aqueous aerosols and the coating on black carbon particles, thus indirectly modifying the water deposition rate. Section 3.1.1 describes the early-plume microphysical evolution and quantifies the role of organic matter on particle growth. The formation of ultrafine volatile plume particles (composed of dissolved $H_2SO_4$ and organic compounds)

modifies the gaseous composition of the plume and thus the condensation of soluble species on black carbon particles. Section 3.1.2 quantifies the sensitivity of the black carbon coating fraction to fuel sulfur content, and examines the partitioning between liquid and gaseous $H_2SO_4$. An analogy is then used to identify the impact of organic aqueous aerosols on the black carbon surface composition. Section 3.1.3 then considers how changes in black carbon emissions affect its surface coating. The simulations carried out in this last section could also be used to provide some insight into a potential role for ultrafine organic particles at low temperatures. We consider these simulations and outcomes because of their relevance to the aged plume in terms of ice uptake, aerosol surface area, and contrail lifetime.

### 3.1.1   Early-plume evolution and role of organic matter

This section first describes the evolution of the aerosol size distribution during the early plume phase, from the engine exit plane to approximately 15 minutes after initial emission. We then quantify one potential effect of organic species on plume chemistry and aerosol size distribution through their effect on black carbon surface composition, by varying the initial fractional coating between 0 and 25%.

Figure 2 displays the evolution of the early plume over different timescales. The air at the exit plane of the engine is assumed to be at a temperature of ~550 K. The plume undergoes rapid cooling in the jet phase and fresh ambient air is entrained into the warm plume. As the temperature decreases, the plume reaches saturation and remains saturated for ~2 seconds. During this time, rapid deposition of water increases the particle radius of ice nuclei (as shown in the upper-right plot). As plume mixing continues, fresh, dry air enters the plume. As the relative humidity with respect to ice falls below saturation, the particles start to sublimate. The loss of ice mass serves to maintain the plume's relative humidity at 100% with respect to ice. This continues until all ice mass has melted, after which point mixing drives the plume relative humidity to the background humidity.

The history of the plume can be displayed as a "mixing line" (on the bottom-right plot in Figure 2) on which the water partial pressure is plotted against plume temperature. The line starts in warm and moist conditions at the engine exit. The dilution of the plume, acting similarly on temperature and water mixing ratio, leads to a straight mixing line until the plume becomes supersaturated. Uptake of gaseous water onto aerosol then reduces the gaseous water partial pressure. Further mixing brings the plume to background state.

Some fraction of the fuel sulfur, which is released in the engine combustor, is converted to gaseous $H_2SO_4$ and quickly condenses. This results in both liquid sulfate aerosol particles, but also coats already-existing particles (including engine-emitted black carbon particles). The bottom-left plot in Figure 2 shows the coating fraction of black carbon particles over time. This fraction is the result of both adsorption

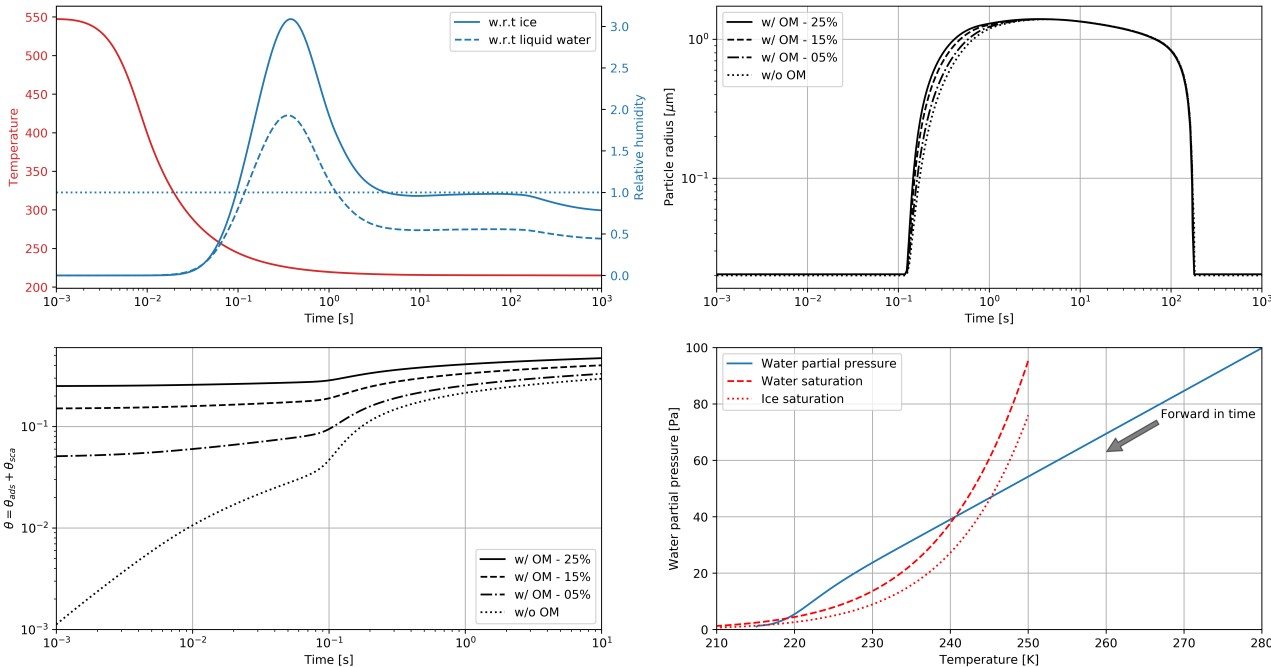

**Figure 2.** Clockwise, from top left: Simulated plume temperature and relative humidities (the saturation threshold is plotted as a dotted line), particle radius (different line styles represent the effect of organic compounds, a mass emission index of 15 mg/kg$_{fuel}$ is prescribed), water mixing line and black carbon fraction coverage. We here consider subsaturated conditions with a background temperature of 215 K. We assume that organic species coat black carbon early-on in the plume (in the first few milliseconds). "OM" refers to the percentage coating by organic matter.

(gas-phase $H_2SO_4$ condensing onto black carbon directly) and scavenging (collision with existing liquid droplets) as described in Kärcher (1998). In the absence of organic matter, the fractional coating is dominated by sulfuric acid. Adsorption of sulfur particles is initially the prevailing pathway to the formation of a black carbon coating. The coating fraction attributed to adsorption plateaus after ~0.1 seconds as the gaseous molecular concentration of sulfuric acid becomes negligible. The remaining growth in the coating fraction is attributed to the scavenging of liquid sulfur aerosols onto black carbon particles.

As described in Section 2.2.3, organic matter can influence the black carbon fractional coating. Given the short timescales associated with formation of particulate organic matter, we assume that the black carbon particles are partially coated with condensable organic compounds in the first few milliseconds. As a sensitivity test, we vary the coating fraction attributable to organic species between 0 and 25%. The bottom-left plot of Figure 2 shows the effect on total fractional coating. Our simplified treatment of organic species shows that the inclusion of organic compounds results in faster particle growth, affecting the transient regime in the first second after emission, but does not have any effect on particle radius (upper-right) after approximately one second. Similarly, the gaseous chemical composition is unaffected by

the condensation of organic compounds onto black carbon particles, even under supersaturated conditions.

### 3.1.2 Effects of fuel sulfur content

We next quantify the role of sulfur emissions on the plume gaseous composition and microphysical evolution. Ultrafine volatile particles are generated early-on in the plume by gas-to-particle conversion, containing both sulfuric acid and organic compounds. As already discussed, these highly-soluble species can condense onto the surface of black carbon particles, thus enhancing the coating fraction.

Figure 3 shows the effect of the fuel sulfur content on the black carbon fractional coverage and the partitioning between gaseous and liquid $H_2SO_4$. In this set of simulations, we vary the FSC between 50 and 5000 ppm. The results from our early-plume representation are in good agreement with the results from Kärcher (1998). The total sulfur concentration is the sum of gaseous $H_2SO_4$ and liquid sulfate and only decreases over time because of plume dilution.

In the low sulfur case, the black carbon particles are only coated on less than 5% of their surface after 100 seconds. This value compares to ~70% in the high FSC case. Our median scenario, assuming a typical fuel sulfur content of 500 ppm by weight, reaches 35%, with 30% originating from the scavenging contribution. These median values are com-

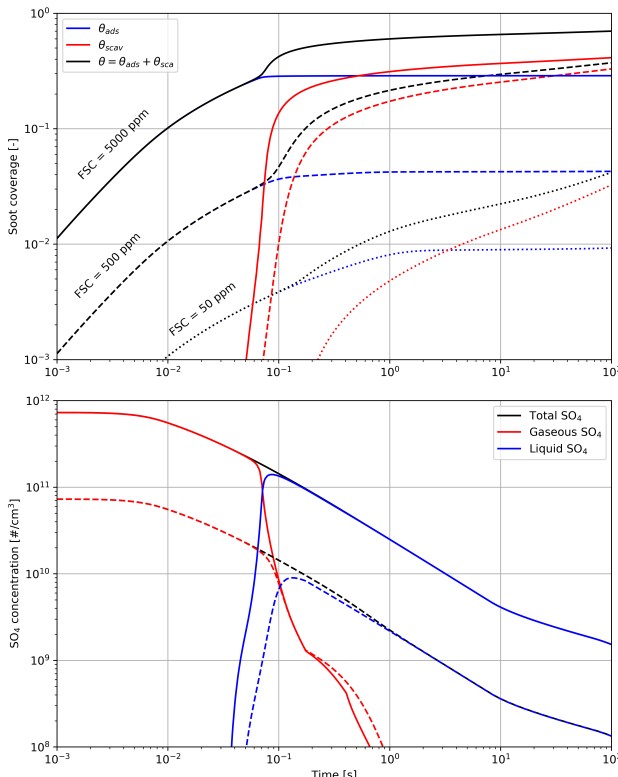

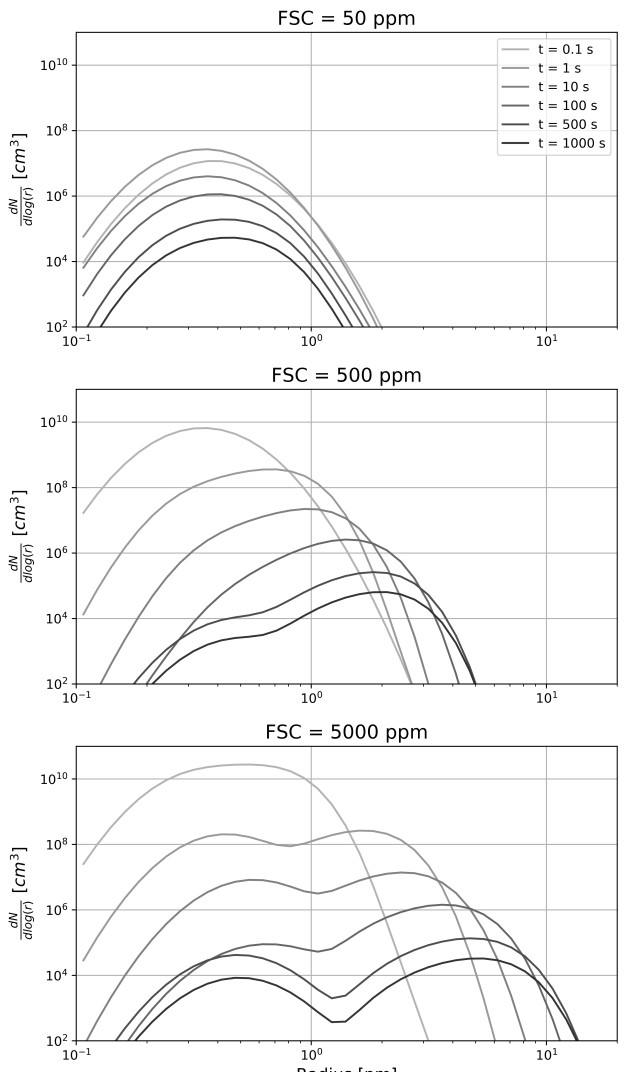

**Figure 3.** Black carbon fractional coverage (top) and number density of $H_2SO_4$ molecules (bottom), in gas and liquid form, as a function of time after emission. The total black carbon fraction coverage $\theta$ is the sum of the adsorption and scavenging contributions. Different line styles correspond to different fuel sulfur content (50, 500 and 5000 ppm).

parable with other studies of the early-plume microphysics (Wong and Miake-Lye, 2010; Kärcher, 1998).

As the plume cools down, $H_2SO_4$ undergoes conversion from gaseous phase to liquid. A large fraction of the emit-
5 ted sulfur is consumed to form liquid particles and coat the aircraft-emitted particles over the first 100 milliseconds after emissions (see Figure 3).

Figure 3 indicates that increasing FSC enhances the particle coatings and the number of liquid plume particles. In
addition, a smaller activated fraction on black carbon particles reduces the initial water condensation rate. However, when particles grow in diameter (of the order of ~0.1 μm or more), the coating fraction becomes unimportant because of the much larger ice surface area. We observe no significant
difference in the peak particle radius when varying the fuel sulfur content. This is in agreement with Busen and Schumann (1995) where no visual impact on contrail properties is detected after drastically reducing the FSC to 2 ppm.

Figure 4 describes the sensitivity of the sulfate aerosol dis-
20 tribution to the fuel sulfur content and its evolution throughout the first few minutes. In three experiments, we set the

**Figure 4.** Time evolution of the sulfate aerosol distribution as a function of the fuel sulfur content. From top to bottom, the FSC is set to 50, 500 and 5000 ppm on a mass basis. The color shading depicts the time evolution, with lighter shading corresponding to a younger plume.

FSC to 50, 500 and 5000 ppm. Two regimes appear. In the low sulfur emission case, very few liquid particles are formed at the nucleation mode (~0.5 nm). In this regime, coagulation is slow and the shape of the aerosol distribution is dictated 25 by the dilution and the entrainment of background aerosols (whose mode is not represented in Figure 4, given their larger size and smaller concentrations). In the high sulfur emission scenario, coagulation is more efficient and leads to the apparition of a second mode (~3 nm). After 100 seconds, the 30 two modes have approximately the same number of particles. At later times, the rate particle formation is reduced, thus promoting the coarser mode. At intermediate FSC val-

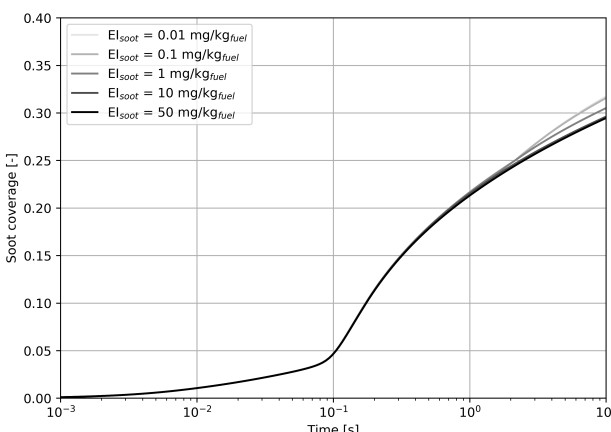

**Figure 5.** Simulated black carbon fraction coverage as a function of the black carbon mass emission index, varied from 0.01 to 50 mg/kg$_{fuel}$.

ues, the two modes still coexist but the larger mode contains more particles.

Similarly to liquid sulfate aerosols, particulate organic matter in aqueous aerosols is a primary contributor to the mass of ultrafine volatile particles. Previous studies have estimated its emission index at ~20 mg/kg$_{fuel}$, comparable to the sulfuric acid mass emission index, assuming an average fuel composition and conversion fraction from S(IV) to S(VI). Their similar emission indices and high solubility allow us to treat organic matter in an analogous way as sulfate aerosols and to draw conclusions that only differ quantitatively.

### 3.1.3 Role of black carbon emissions

In this section, we investigate the role of the black carbon emission index on the early plume aerosol size distribution. We perform simulations in which the black carbon mass emission index is varied between 0.01 mg/kg$_{fuel}$ and 50 mg/kg$_{fuel}$, to cover the potential range of current and future engine emission characteristics. We first focus on the impact of the emitted black carbon mass on its surface coating and then consider the effect on the particle radius over time, as both components could ice uptake and contrail lifetime.

Figure 5 indicates that we find no significant difference on the fractional coverage when varying the black carbon emissions by three orders of magnitude. In all cases, the contribution of sulfur adsorption to the black carbon area coverage is identical, and the contribution from liquid particle scavenging differs by a negligible amount.

Based on the results from Figure 5, all cases studied in this section have similar coating fractions, independent of the black carbon emission index. However, Figure 6 shows the in-plume relative humidity and particle radius when the black carbon emission index is varied. When the plume reaches saturation with respect to ice, the particle growth rate is the

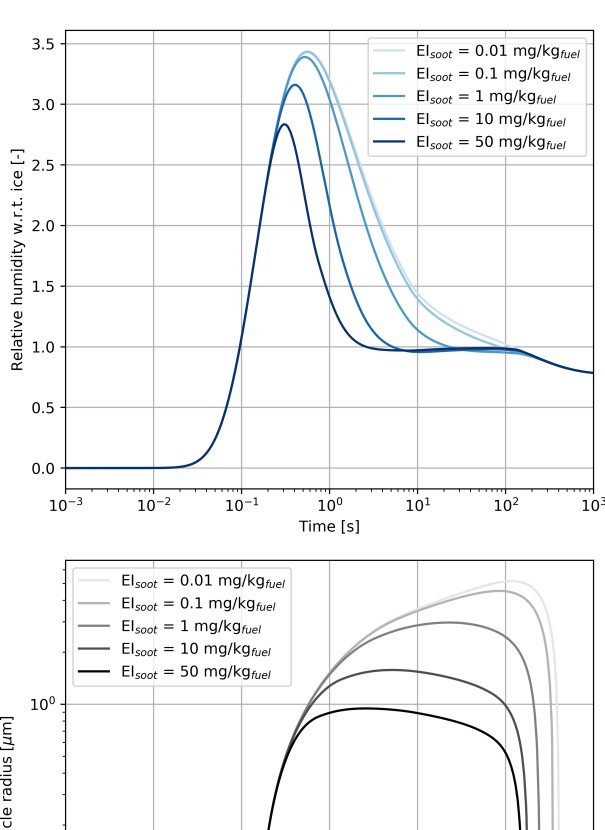

**Figure 6.** Relative humidity with respect to ice (top) and particle radius (bottom) for a black carbon mass emission index varying from 0.01 to 50 mg/kg$_{fuel}$.

same regardless of the number of particles. In all simulations, the particles are competing for the same quantity of water. As a result, increasing the black carbon particle number density leads to smaller crystals, which in turn reduces the length of the supersaturated period for exhaust plumes in an initially subsaturated air mass. However, the same mechanism would be expected to result in longer contrail lifetimes for higher emissions indices if the air was initially saturated or supersaturated, due to the lower settling velocities of smaller particles.

Further reductions in the black carbon emission index could change the contrail formation pathway. The studies from Kärcher and Yu (2009) demonstrate that liquid plume particles (containing dissolved H$_2$SO$_4$ and organic compounds) could play a role in contrail ice formation at low black carbon number emissions and low temperatures. Am-

bient liquid particles are characterized by low number concentrations compared to an aircraft plume and thus cannot by themselves explain contrail formation. However, recent numerical simulations from Rojo et al. (2015) suggest that volatile particles could influence contrail formation, especially for alternative fuels in which the fuel sulfur content and black carbon emissions may be lower. Although we do not explicitly model the freezing of liquid plume particles, our simulations with increased black carbon number emissions provide some insight into the possible effect of organic and sulfate aerosols if they are capable of acting as ice nuclei.

### 3.2 Limitations of instant dilution

We first simulate the evolution of an aircraft plume as simulated using APCEMM. Figure 7 shows the time series of the ozone and $NO_x$ perturbations over the first 24 hours after emission. The results as calculated under an instant dilution assumption (single, well-mixed box) are also shown.

The chemical evolution of the plume can be split into three regimes, distinct from the dynamical regimes described in Section 2.1 (Song et al., 2003; Vinken et al., 2011). The first regime is characterized by very high $NO_x$ mixing ratios (>1 ppmv), causing ozone titration. In this period, typically lasting 10 minutes, high mixing ratios of nitric oxide (NO) rapidly deplete local ozone concentrations, resulting in a burst of $NO_2$ production through reaction [A1] (see Table 3). In this regime, $HO_x$ (= $OH + HO_2$) production is suppressed by the lack of ozone (reactions [A5-A6]).

As the plume dilutes and $NO_x$ mixing ratios fall below 1 ppmv, it enters the second regime. With little ozone remaining, $HO_2$ reacts with the remaining NO (reaction [A4]), producing OH and $NO_2$ without depleting ozone. This leads to increased OH levels and enhanced ozone production. Meanwhile, photolysis of $NO_2$ through reaction [A2] results in the recovery of ozone, which had been depleted during the first regime. Between one and two hours after emission, ozone has been restored to its background value. Reactions [A7] through [A10] lead to conversion of emitted $NO_x$ to nitrogen reservoir species.

A few hours after emission, the third regime begins, characterized by $NO_x$ mixing ratios below 1 ppbv. Reaction [A4] and reactions including organic peroxides (such as [A12]) cause increasing levels of ozone and additional conversion to reservoir species. Aircraft plumes, similarly to ship plumes, are characterized by a high $NO_x$ to volatile organic compound (VOC) ratio, therefore favoring termination reactions (e.g. [A7]) over catalytic ozone formation (Song et al., 2003).

Differences between the model outputs are dominated by the behavior during the first two regimes. Explicitly modeling the plume allows the initial ozone destruction to be captured because the highly-concentrated plume is resolved. Although a recovery in ozone is later simulated once the plume diffuses, additional production that would have occurred during the early plume is prevented.

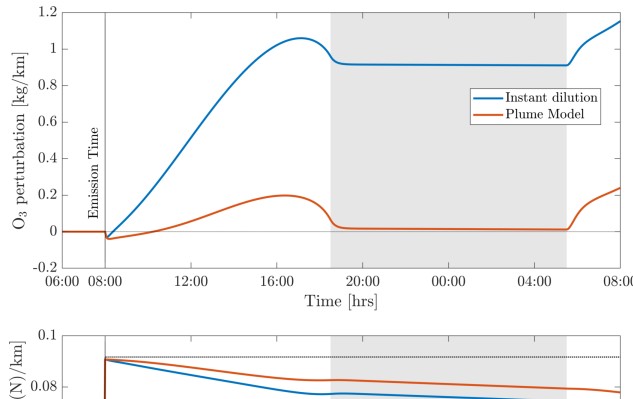

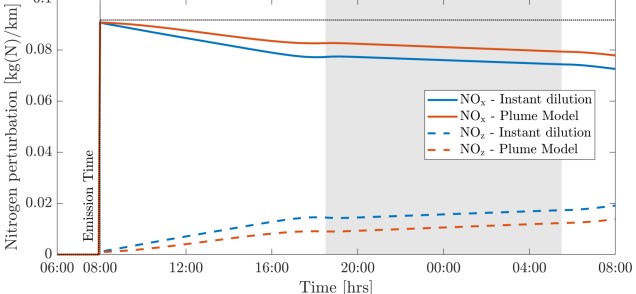

**Figure 7.** Perturbations in ozone ($O_3$), nitrogen oxides ($NO_x$) and the nitrogen reservoir species ($NO_z$) according to simulations using an instant dilution approach and the plume model. Emissions are released at 8:00 AM local time in a polluted environment. The black dotted line represents all nitrogen species ($NO_y$), which is a conserved quantity. The shaded areas correspond to nighttime.

**Table 3.** Dominant $O_3/NO_y/HO_x$ reaction pathways in APCEMM.

| Reaction # | Reaction | | |
|---|---|---|---|
| [A1] | $NO + O_3$ | $\longrightarrow$ | $NO_2 + O_2$ |
| [A2] | $NO_2 + h\nu$ | $\longrightarrow$ | $O(^3P) + NO$ |
| [A3] | $O(^3P) + O_2$ | $\longrightarrow$ | $O_3$ |
| [A4] | $NO + HO_2$ | $\longrightarrow$ | $NO_2 + OH$ |
| [A5] | $O_3 + h\nu$ | $\longrightarrow$ | $O(^1D) + O_2$ |
| [A6] | $O(^1D) + H_2O$ | $\longrightarrow$ | $2OH$ |
| [A7] | $NO_2 + OH + M$ | $\longrightarrow$ | $HNO_3 + M$ |
| [A8] | $NO_2 + O_3$ | $\longrightarrow$ | $NO_3 + O_2$ |
| [A9] | $NO_3 + NO_2$ | $\longrightarrow$ | $N_2O_5$ |
| [A10] | $N_2O_5 + H_2O$ | $\xrightarrow{Aerosol}$ | $2HNO_3$ |
| [A11] | $CH_4 + OH$ | $\longrightarrow$ | $CH_3O_2 + H_2O$ |
| [A12] | $CH_3O_2 + NO$ | $\longrightarrow$ | $HCHO + \dots$ $HO_2 + NO_2$ |
| [A13] | $CO + OH$ | $\longrightarrow$ | $CO_2 + H$ |
| [A14] | $H + O_2 + M$ | $\longrightarrow$ | $HO_2 + M$ |
| [A15] | $RH + OH$ | $\xrightarrow{O_2}$ | $RO_2 + H_2O$ |
| [A16] | $RO2 + NO$ | $\longrightarrow$ | $RO + NO_2$ |

R represents an organic compound. The RH notation is used to describe Volatile Organic Compounds (VOCs).

In the instant dilution model, this ozone destruction and production cut-off is not captured. Because ozone is not locally depleted, the instant dilution model instead simulates a prolonged period of net ozone production, as $HO_x$ concen-
5 trations remain close to background values. The instant dilution approach, unlike APCEMM, bypasses the first two $HO_x$-limited regimes and is therefore in a $NO_x$-rich, $HO_x$-rich environment, favoring daytime ozone production and conversion of $NO_x$ to reservoir species. Additionally, instant dilu-
10 tion of aircraft emissions results in shorter $NO_x$ lifetimes (see Appendix C for more details).

The net result is that, after 24 hours, the instant dilution approach estimates that the aircraft plume has produced ~1.2 kg of ozone per kilometer flown, compared to ~0.2 kg per
15 kilometer estimated by APCEMM for a $NO_x$ emission index of 11.5 $g/kg_{fuel}$. By this stage in the simulation both models show similar chemical behavior, as the plume has become sufficiently dilute to be well-represented by the instant dilution model. However, the erroneous simulation of ozone
production in the initial phase leads to a persistent and significant error in the net ozone production of the plume.

This behavior, and the discrepancy between APCEMM and an instant dilution model, is strongly affected by local meteorology. Increased diffusion, or equivalently higher
wind shear, dilutes the plume with a larger mass of air, minimizing ozone depletion. Therefore, total ozone production scales directly with mixing parameters. Table 4 shows the remaining $NO_x$ and total mass of produced ozone after 24 hours as a function of the local diffusion coefficients. The
results for instant dilution are shown in the last row. As diffusion rates increase and dilution becomes faster, the discrepancy between APCEMM and the instant dilution model decreases towards zero. Errors in global simulation of aircraft impacts will therefore be maximized in regions with low dif-
fusion and/or wind shear.

### 3.3 Influence of background conditions

The in-plume ozone perturbation $(\Delta[O_3](t))$ and the conversion efficiency of $NO_x$ to $NO_y$ are influenced by parameters such as the emission time and background conditions. We
first investigate the influence of changes in background $NO_x$. Figure 8 shows how the 24-hour ozone emission conversion factor, $ECF_{O_3}$, varies as a function of $NO_x$ background concentration and date of emission. Both simulations have been integrated over 24 hours. All simulations are conducted after
a 5-day spin-up, and are simulated as occurring at 220 hPa altitude and $60°$ N.

The instant dilution approach overestimates ozone production for any emission time, with emission conversion factors in the box model that are up to three times their respective
values in the plume model. These discrepancies are greatest in summertime due to the larger ozone production term. The size of the ozone perturbation is sensitive to background concentrations of $NO_x$ in both models.

**Table 4.** Influence of diffusion parameters and wind shear on in-plume chemistry

| | Diffusion coefficients [m²/s] | Remain. $NO_x$ [%] | $O_3$ perturbation [kg/km] |
|---|---|---|---|
| $s = 0.000\ s^{-1}$ | $D_h = 05, D_v = 0.05$ | 88 | 0.063 |
| | $D_h = 10, D_v = 0.10$ | 86 | 0.17 |
| | $D_h = 15, D_v = 0.15$ | 85 | 0.26 |
| | $D_h = 20, D_v = 0.20$ | 84 | 0.34 |
| | $D_h = 25, D_v = 0.25$ | 83 | 0.41 |
| $s = 0.003\ s^{-1}$ | $D_h = 05, D_v = 0.05$ | 86 | 0.078 |
| | $D_h = 10, D_v = 0.10$ | 84 | 0.21 |
| | $D_h = 15, D_v = 0.15$ | 83 | 0.31 |
| | $D_h = 20, D_v = 0.20$ | 82 | 0.42 |
| | $D_h = 25, D_v = 0.25$ | 82 | 0.47 |
| | Instant dilution: | 79 | 1.3 |

Data obtained 24 hours after emission.

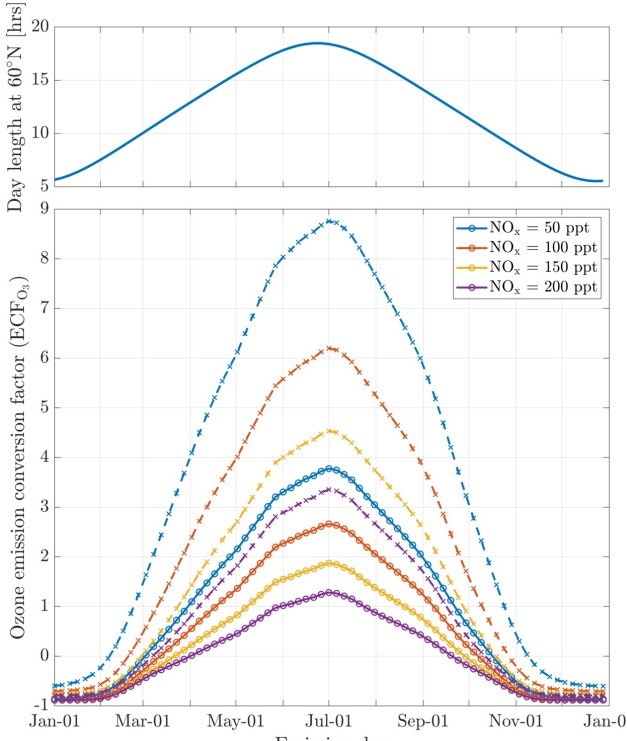

**Figure 8.** The bottom figure displays the 24-hour ozone emission conversion factor ($ECF_{O_3}$) from the emission of a B747-8 equipped with GEnx engines at 08:00 local time, at $60°$ N, for an instant-dilution approach (dotted lines) and the APCEMM plume model (continuous lines) as a function of day of the year. Different scenarios representing different background $NO_x$ mixing ratios are displayed. The cases correspond to a background $O_3$ mixing ratio of 52 ppb. The figure on the top displays the amount of sunlight received at $60°$ N as a function of day of the year, expressed in hours of daytime.

**Table 5.** Influence of $NO_x$ emission index on emission conversion factors and in-plume ozone perturbation

| $EI_{NO_x}$ [g/kg$_{fuel}$] | $ECF_{O_3}$ [-] | $ECF_{NO_x}$ [-] | $\Delta[O_3]$ (24 h) [pptv] |
|---|---|---|---|
| $EI_{NO_x}$ = 8.0 | 3.2 | 0.77 | 59 |
| $EI_{NO_x}$ = 12.0 | 2.4 | 0.79 | 67 |
| $EI_{NO_x}$ = 16.0 | 1.9 | 0.80 | 70 |
| $EI_{NO_x}$ = 20.0 | 1.5 | 0.82 | 70 |

Data obtained 24 hours after emission and for emission at 8:00 on June 16[th].

During summertime, increasing the background concentration of $NO_x$ from 100 to 200 pptv reduces the net (positive) ozone perturbation by 30-45% in both models. During wintertime, the same change in background $NO_x$ has a negligible effect in the plume model, as shown in Figure 8. However, the instant dilution approach is still sensitive to this change. It produces a larger (more negative) ozone perturbation when the background $NO_x$ is increased during wintertime. This pattern is explained by a less efficient conversion of $NO_x$ to reservoir species at night. The transition between net positive and net negative ozone also changes as a function of the background $NO_x$. At 50 pptv of background $NO_x$, the plume model simulates net ozone production for 10 months, compared to 8 months in the instant dilution model. At 200 pptv, net production is simulated for 6 and 5 months by the two models respectively. This inconsistency in the magnitude and sign of the error between the two models means that the true impact of aviation emissions will be inconsistently modeled by an instant dilution approach.

At a finer scale, we observe variations in emissions impacts depending on the time of day of the emission. Figure 9 shows contours of ozone and $NO_x$ emission conversion factors for different times of day over the course of a year. For most of the year, the total production of ozone is relatively insensitive to the exact time of day of the emission. The exception is during summertime, when emissions immediately before local sunset (the upper dotted line) cause almost twice as much ozone to be produced as an emission during late morning. This is discussed in more detail in Appendix C.

## 3.4 Influence of $NO_x$ emission index

In this section, we vary the $NO_x$ emission index while keeping other emissions unchanged. The total $NO_x$ emitted into the plume also affects chemical outcomes. Table 5 shows how a range of impact metrics are affected by changes in the $NO_x$ emission index. The overall ozone ECF decreases as the $NO_x$ emission index increases, falling from 3.2 for an EI of 8 g/kg$_{fuel}$ to 1.5 for an EI of 20 g/kg$_{fuel}$. However, the product of these two factors - proportional to the total ozone present after 24 hours - still increase monotonically with the emission index of $NO_x$ over the range of values considered.

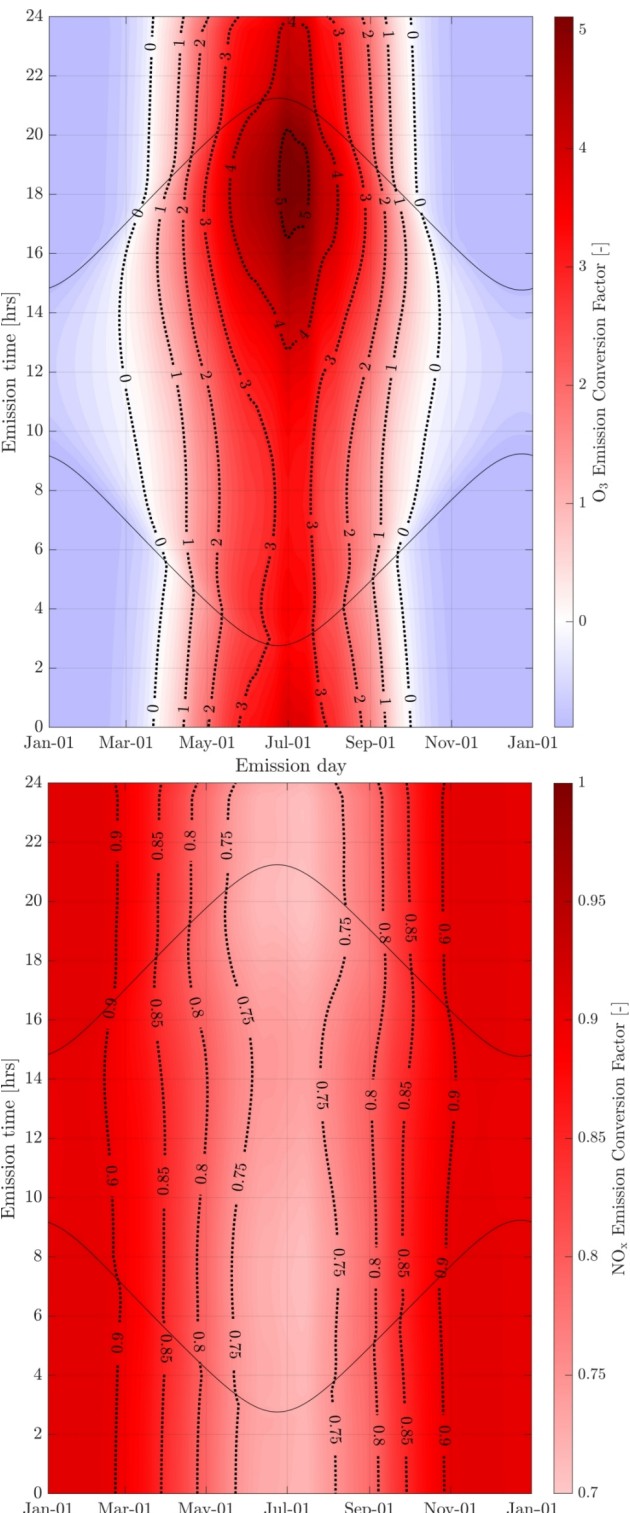

**Figure 9.** 24-hour $O_3$ (top) and $NO_x$ (bottom) emission conversion factors from the emission of a B747-8 equipped with GEnx engines at 60° N. Dotted lines represent sunrise and sunset at the given latitude.

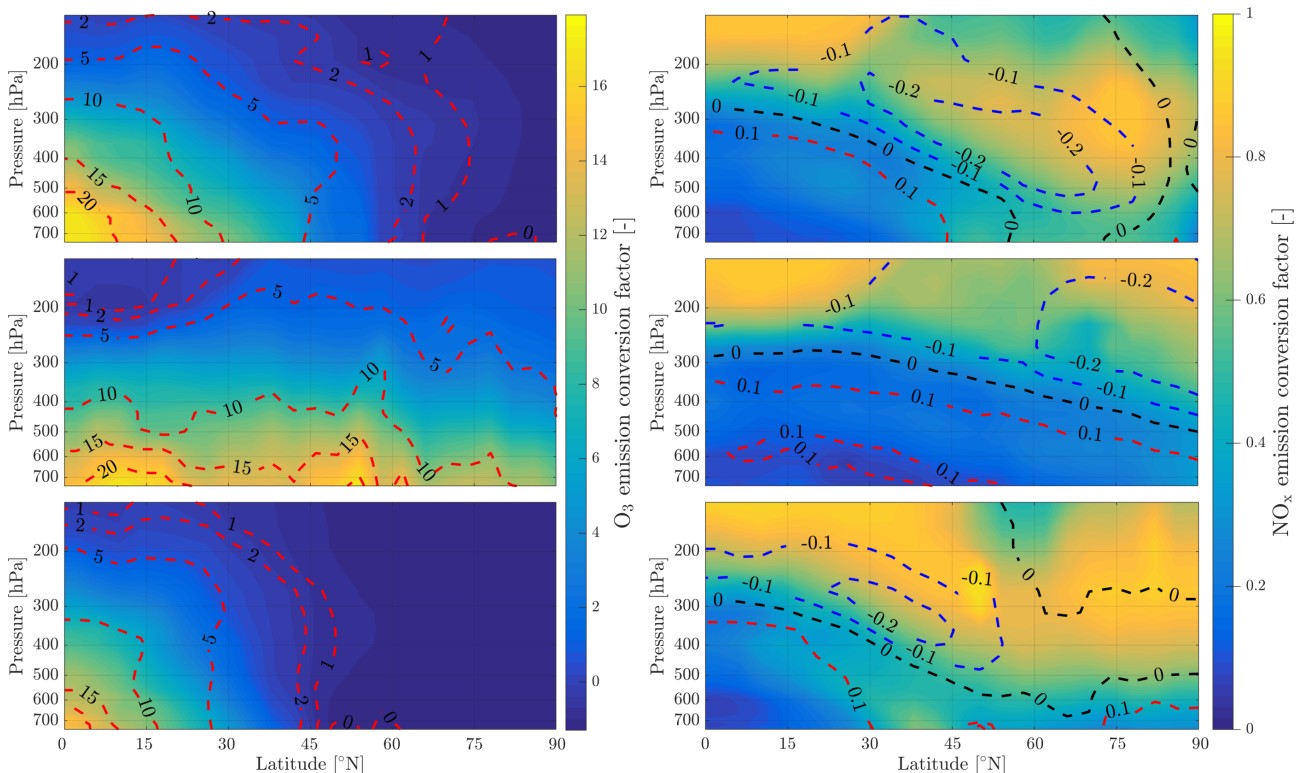

**Figure 10.** Contour plots of $O_3$ (left) and $NO_x$ (right) conversion emission factor, 24 hours after emission from APCEMM. The isolines represent the discrepancy between the instant dilution approach and APCEMM. Blue isolines represent cases where the species ECF is underestimated, whereas the red isolines signify that the quantity is overestimated by instant dilution. Simulations have been carried out for emissions at 8:00 on March 21st, June 21st and December 21st (from top to bottom).

This implies that decreasing the $NO_x$ emission index provides non-linear benefits in terms of total ozone production. A one unit increase in the $NO_x$ emission index (expressed in $g/kg_{fuel}$) leads to a reduction in $ECF_{O_3}$ of 0.08 mol/mol under high $EI_{NO_x}$ scenarios, but this increases to 0.2 mol/mol for the same absolute reduction in the $NO_x$ under low $EI_{NO_x}$ scenarios.

These results agree with the findings from Petry et al. (1998), Meijer (2001) and Vohralik et al. (2008). These simulations suggest that, relative to a plume-scale treatment of chemical processes, conventional instant-dilution approaches overestimate ozone production by up to a factor of three, and overestimate conversion of nitrogen oxides to reservoir species. We also find that decreasing aircraft emissions of $NO_x$ emissions yields accelerating returns in terms of total in-plume ozone production, but that these results are sensitive to background $NO_x$ concentrations.

### 3.5 Influence of pressure and latitude

The atmospheric response to aircraft emissions also varies as a function of the pressure and latitude of the emission. Although latitude is not a physical parameter of the model, it is equivalent to defining the amount of sunlight received, which affects photolysis rates and background conditions. We simulate pressures from 750 hPa to 150 hPa and latitudes from 0°N to 90°N. Temperature data is taken from monthly-averaged MERRA-2 meteorological data, for 2013. To capture variation of a single flight's emission conversion factors with geographic location and altitude, background conditions and photolysis rates are taken from GEOS-Chem. To also capture seasonal effects, simulations are carried out for emissions taking place on the winter and summer solstices as well as during the spring equinox, on March 21st. We perform simulations using both models. Results are presented in Figure 10 in terms of ozone and $NO_x$ emission conversion factors. Isolines of the discrepancy between both models are plotted on Figure 10 for $O_3$ and $NO_x$ ($\varepsilon_{O_3}$ and $\varepsilon_{NO_x}$).

The results show a link between ozone production efficiency and latitude and pressure. Increasing pressure enhances the ozone emission conversion for the same amount of emitted $NO_x$, given sufficient sunlight. The amount of sunlight drives ozone production, as little ozone is generated in the most northern latitudes during winter. At high flight altitudes or in cold regions, the daytime $NO_x$-driven ozone production is of the order of magnitude of the ozone loss at dusk and the early titration effect. This cancellation leads to a

**Table 6.** Discrepancies between the instant-dilution approach and APCEMM at cruise altitudes. The left-most column shows the average ozone ECF as calculated in APCEMM, while the central column shows the average discrepancy in ECF between the instant dilution model and APCEMM. The right-most column shows the maximum calculated error. A positive error value means that the instant dilution model overestimates ozone production. All variables are evaluated and averaged over cruise altitudes only.

| Date | $\overline{ECF_{O_3}}$ | $\overline{\varepsilon_{O_3}}$ | $\max(\varepsilon_{O_3})$ |
|---|---|---|---|
| 03/21 | +0.18 | +2.6 | +5.5 |
| 06/21 | +0.98 | +2.4 | +4.4 |
| 12/21 | -0.31 | +2.0 | +5.3 |

small in-plume ozone perturbation of varying sign as shown previously (Vohralik et al., 2008).

The instant dilution approach consistently overestimates the amount of ozone produced at cruise altitudes (~150 to ~240 hPa), as shown in Table 6. In absolute terms, the instant-dilution approach performs worst during summertime when ozone production is enhanced across the Northern hemisphere and the discrepancy in the ozone emission conversion factor ($\varepsilon_{O_3}$) is larger. The maximum ozone discrepancy (Table 6) reaches values around 5 in all seasons, corresponding to a relative error of approximately +200%.

$NO_x$ conversion shows different sensitivities to location than the ozone ECF. As shown in the right panels of Figure 10, the $NO_x$ ECF is positively correlated with ambient temperature but is insensitive to the amount of sunlight and season. As the temperature decreases with increasing altitude in the troposphere, the conversion of $NO_x$ to $NO_y$ is lowest at high altitude, going from an average value of 0.3 at 700 hPa to approximately 0.75 at 150 hPa. Greater conversion occurs in warmer air, around the equator and the tropics. Furthermore, the instant-dilution approach underestimates the amount of remaining $NO_x$ at high altitudes but overestimates at lower levels. The crossover point varies significantly with season and latitude.

## 3.6 Contrail microphysical, optical, and chemical properties

All analysis thus far has concerned conditions that are subsaturated with respect to ice. The simulated plumes have therefore been made up only of gas-phase constituents and non-ice aerosol particles. However, a second discrepancy between instant dilution models and real aircraft plumes is the lack of condensation trails in simulated aircraft exhaust, which can cause both climate and chemical impacts. This section assesses APCEMM's ability to simulate aircraft-induced condensation trails ("contrails"), quantifying how changes in background conditions affect the properties of the contrail. We then quantify the effect that these condensation trails

have on the long-term atmospheric effects of aircraft emissions. This includes an investigation of differences in contrail lifetime and effects when forming in the stratosphere, as may result from supersonic flight.

In the following sections, we use similar background meteorological conditions to Unterstrasser and Gierens (2010a). We simulate a 1 km thick supersaturated layer, below which there is a linear decrease in relative humidity at a rate of 12% per hundred meters, to a background value of 50%. The average supersaturation in the upper troposphere has been estimated to be around 15% (Gierens et al., 1999), corresponding to a saturation of 115% with respect to ice. Atmospheric shear is set to 0.002 s$^{-1}$. The flight-level temperature is to 217 K. A diurnal temperature variation with a 0.1 K amplitude is applied, corresponding to daily temperature fluctuations in the upper-troposphere (Seidel et al., 2005). We also consider a range of contrail updraft velocities, from 0 to 10 cm/s. This is based on previous studies that have shown that heating of contrail ice induces an upward convection motion with velocities in this range, such that the plume enters a colder environment (Unterstrasser and Gierens, 2010b; Unterstrasser et al., 2017). This updraft can cause the contrail to advect out of the supersaturated layer. An intercomparison between APCEMM and Unterstrasser and Gierens (2010a) is provided in Appendix D. As described in Appendix D2, the environment surrounding a contrail is characterized by local oscillations in the humidity field that arise from turbulent motion. This phenomenon is not explicitly in APCEMM and parameterization of these fluctuations is considered a future research priority.

### 3.6.1 Contrail simulations and the impact of relative humidity and initial contrail updraft velocity

We first simulate the formation and evolution of a contrail in aircraft exhaust under a variety of conditions, to quantify the range of likely behaviors and verify behavior consistent with observations. For these purposes, we simulate an aircraft plume in locally supersaturated air, with flight-level relative humidities ranging from 104% to 110% and initial contrail updraft velocities, from 0 to 10 cm/s.

Figure 11 shows the temporal evolution of total contrail ice crystal number and mass for each combination of parameters, as well as the extinction-weighted effective diameter (Unterstrasser and Gierens, 2010a). In all cases, the contrail persists for at least 5 hours, but has sublimated after 10 hours. These lifetimes are consistent with observations, in spite of the idealized meteorological conditions considered here (Minnis et al., 1998; Iwabuchi et al., 2012). Most ice crystals are lost through sublimation and *in situ* losses. *In situ* losses correspond to the sublimation of small crystals in favor of larger crystals when the relative humidity approaches 100%. This phenomenon is attributed to Ostwald ripening through the Kelvin effect (Lewellen et al., 2014; Unterstrasser et al., 2017). Losses through Brownian coag-

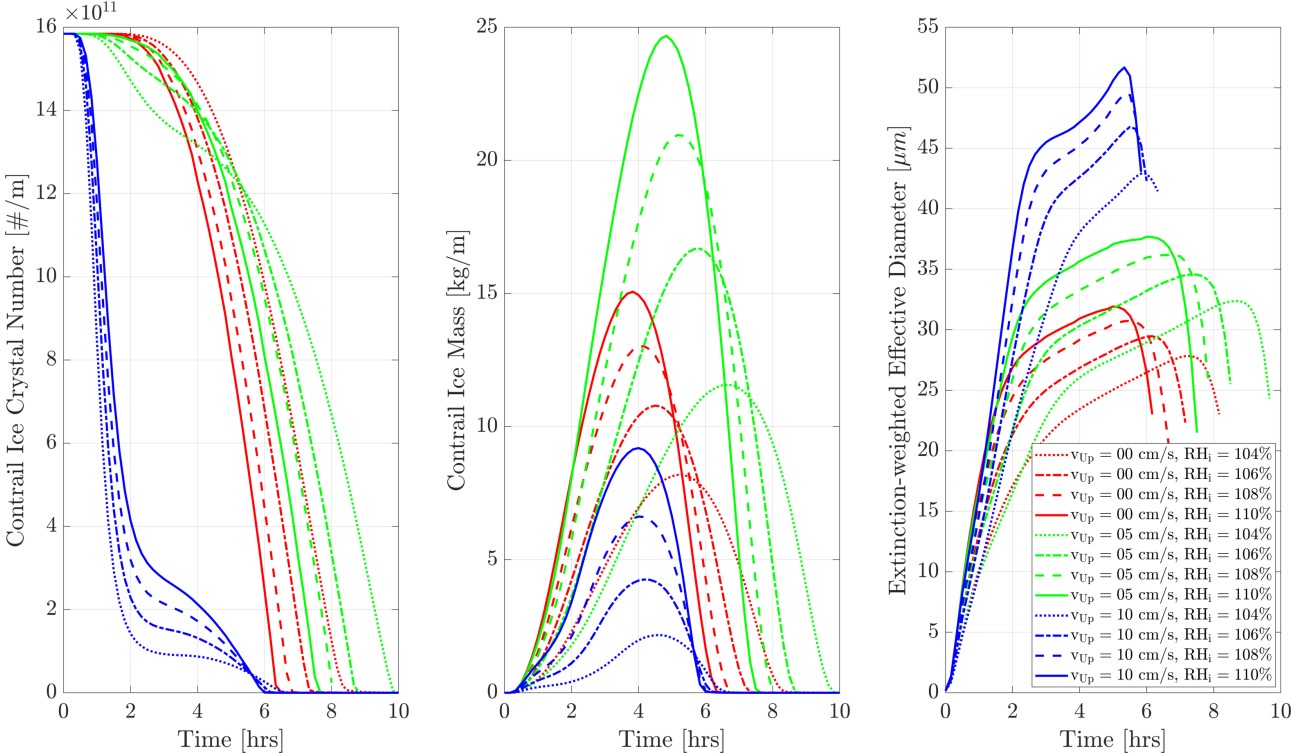

**Figure 11.** Contrail ice crystal number (left), ice crystal mass (middle) and extinction-weighted effective diameter (right) when varying background relative humidity and the contrail updraft velocity magnitude.

ulation are negligible as they account for less than 1% of particle losses. As the contrail expands, the contrail core is dehydrated through gravitational sedimentation of the largest particles, leaving behind a population of smaller ice crystals with little ice mass but a significant surface area. The formation and settling of large ice crystals (with a radius greater than 30 μm) lead to early variations in contrail ice mass. This means that growth in contrail ice mass slows earlier than would be captured by reduced order models that consider only the mean settling velocity.

The contrail ice crystal number is reduced when the updraft velocity is set to 10 cm/s. This is because the ice particles are advected out of the supersaturated region and thus sublimate quickly. The remaining crystals in the supersaturated region experience rapid growth, and their large settling velocity reduces the contrail lifetime. For updraft velocities of 0 and 5 cm/s, the contrail remains in the supersaturated layer, thus increasing the total ice mass over a larger number of particles, resulting in an increased contrail lifetime.

The contrail ice mass peaks between 4 and 5 hours after formation, with variations in timing affected by both the contrail updraft velocity and background relative humidity. In both cases, moister air result in greater overall water uptake and therefore larger ice particles. These larger particles then fall to drier altitudes and melt, reducing the available number of particles. Higher relative humidity also increase

the extinction-weighted effective diameter for the same reason. This results in a trade-off between the "size" of the contrail, in terms of either the ice mass or effective diameter, and its lifetime, with more massive contrails having shorter lifetimes.

### 3.6.2 Influence of engine black carbon emission index on contrail properties

We next model how changes in black carbon emissions affect the properties of the contrail. We simulate an aircraft plume in which the black carbon mass emission indices are varied between 10 and 60 mg/kg$_{fuel}$, compared to 10 to 14 mg/kg$_{fuel}$ estimated using the $SN-C_{BC}$ method for the GEnx engine. All other aircraft and engine emissions parameters are fixed for this sensitivity analysis. As a comparison, Stettler et al. (2013) estimate a fleet-wide average black carbon mass emission index of 28 mg/kg$_{fuel}$ while *in situ* observations at cruise altitude have shown that different engines are estimated to have emission indices between 11 to 100 mg/kg fuel (Petzold et al., 1999). Each simulation assumes a flight-level temperature and relative humidity of 217 K and 110% respectively, and a post-vortex sinking updraft of 5 cm s$^{-1}$.

Figure 12 shows the evolution of the total ice particle number, total ice mass, and extinction-weighted effective diameter for each emissions scenario. Considering first the total ice

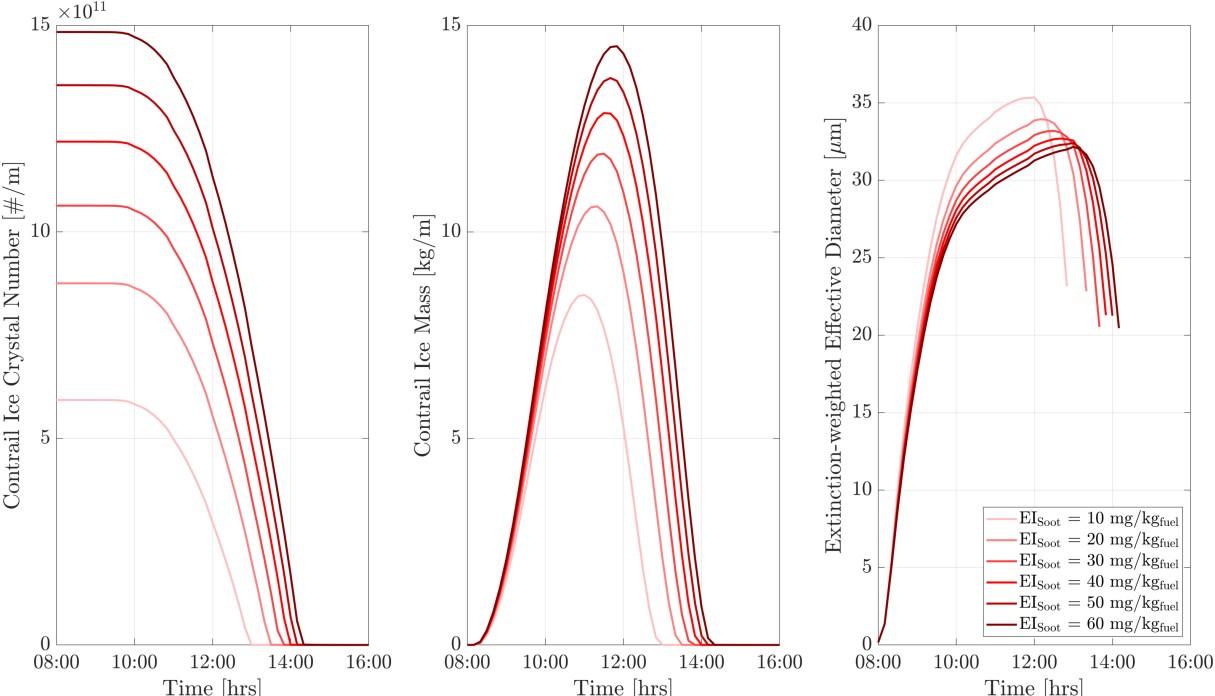

**Figure 12.** Total contrail ice crystal number (left), ice crystal mass (middle) and extinction-weighted effective diameter (right) for different black carbon emission indices. A background relative humidity of 110%, a temperature of 217 K at flight level are assumed.

mass, two distinct regimes are visible. The first regime occurs over the first three hours after formation. In this regime, all cases have identical ice masses. The contrail ice mass is controlled only by the ambient relative humidity, as all available water mass is taken up by whatever particles are available, with lower emission indices result in larger crystals.

After three hours, the simulations with the lowest emission indices start to lose ice mass. This is because of the discrepancy in ice crystal radius. Since the same ice mass is taken up on a smaller number of particles, they become larger and fall more rapidly to warmer altitudes. Lower emission indices result in smaller particles, extending the lifetime of the contrail. As the largest crystals are removed, only a core of small crystals remains.

These changes in contrail ice also affect the optical thickness of the plume. We calculate optical thickness by integrating the extinction $\chi$, as defined in Ebert and Curry (1992). Figure 13 shows the temporal evolution of the optical thickness, integrated over the vertical and horizontal (perpendicular to the flight path) axes. As seen previously, the ice water content is identical in the first few hours for all simulations, meaning that a scenario with a large particle number has a larger total crystal surface area. This means that reducing the black carbon emission index decreases both the optical depth

**Table 7.** Predominant optical depth for different black carbon mass emission scenarios

| $\mathrm{EI_{BC}}$ $(\mathrm{mg/kg_{fuel}})$ | 30 min | 1 hour | 2 hours | 5 hours |
|---|---|---|---|---|
| 10 | 0.11 | 0.22 | 0.32 | 0 |
|    | -20%[*] | -16%[*] | -15%[*] | -100%[*] |
| 20 | 0.13 | 0.26 | 0.38 | 0.15 |
|    | - | - | - | - |
| 40 | 0.16 | 0.30 | 0.43 | 0.30 |
|    | +20%[*] | +15%[*] | +14%[*] | +105%[*] |
| 60 | 0.18 | 0.32 | 0.47 | 0.39 |
|    | +32%[*] | +26%[*] | +24%[*] | +165%[*] |

[*] Relative changes with respect to the baseline values corresponding to a black carbon mass emission index of 20 $\mathrm{mg/kg_{fuel}}$

of the contrail and its lifetime. This is quantified in Table 7, which quantifies the "predominant" optical depth as

$$\tau_{\mathrm{pre}}(t) = \frac{\int \tau^2(x,t)dx}{\int \tau(x,t)dx}, \qquad (9)$$

following the formulation of Unterstrasser and Gierens (2010a). Table 7 shows how this optical depth varies over time for each scenario. In all cases, the predominant optical depth increases with the black carbon emission index,

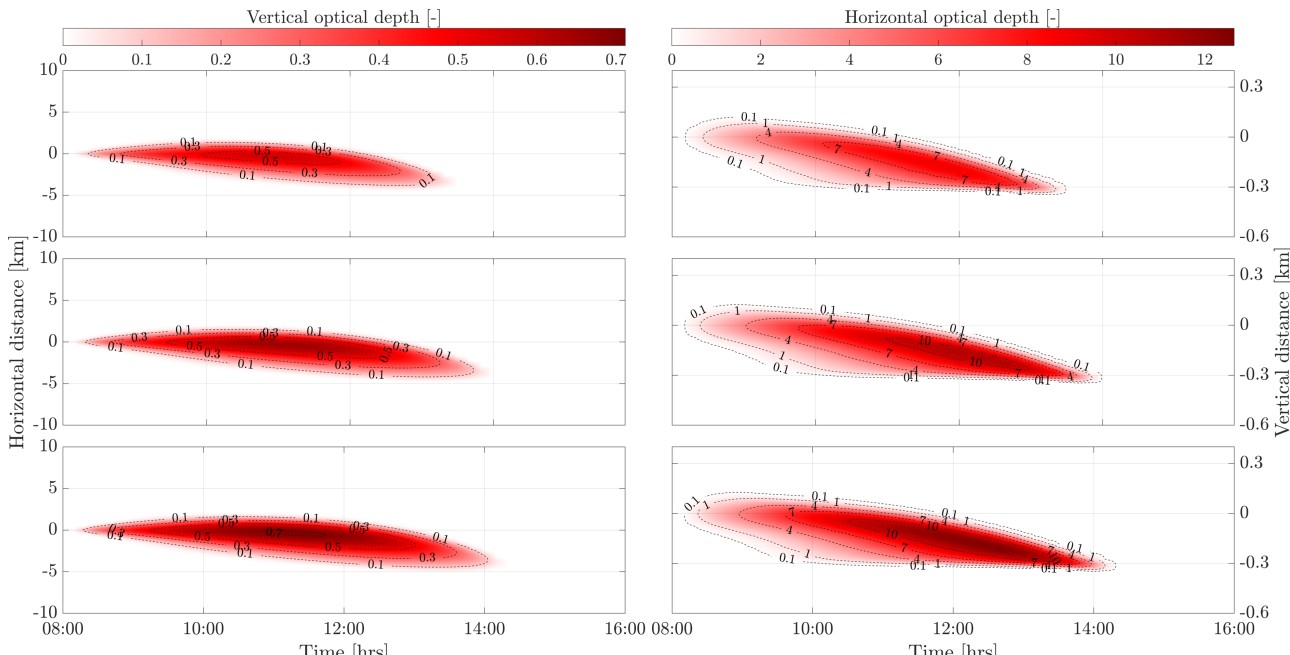

**Figure 13.** Contours of optical thickness along the vertical (left) and horizontal (right) direction for different black carbon mass emission indices: 20 (top), 40 (center), 60 mg/kg$_{\text{fuel}}$ (bottom) respectively. The horizontal axis is centered on the flight location. The vertical axis represents the distance with respect to the altitude after vortex sinking. The flight altitude before vortex sinking is at a pressure altitude of 10.6 km. Vortex sinking caused the plume to settle at a new altitude, 112 m lower, corresponding to the origin of the vertical axis. We apply a wind shear of 0.002 s$^{-1}$, causing the contrail to move left of the initial center line while sinking. An optical depth of 0.1-0.2 is usually given as a contrail detection threshold through satellite sensing (Kärcher et al., 2009).

with the relative difference between scenarios increasing over time until 4 hours after emissions. After this time, the cases with the lowest black carbon emission indices have almost fully sublimated. Scaling the emitted black carbon by a factor 2 with respect to the baseline case (set to 20 mg/kg$_{\text{fuel}}$) increases the 2-hour predominant optical depth by 14%. Further enhancement of the black carbon emission index yields smaller increases in $\tau_{\text{pre}}$, although we find no reversal in trend for EI$_{\text{BC}}$ up to 3 times the baseline value.

Aggregating these results, we find that doubling the black carbon mass emission index from the baseline case increases the peak ice mass by approximately 24% and delays the time at which the peak occurs by up to 1 hour. This translates to a larger climate impact. Reducing the amount of released black carbon particles could instead cut down the contrail-cirrus radiative forcing; halving the black carbon emission index decreases the optical depth after two hours by ~15%.

### 3.6.3 Influence of engine black carbon emissions on chemical composition

In addition to changing the optical properties of the contrail, black carbon emissions affects the chemical impact of the plume. Table 8 shows how the 24-hour emissions conversion factors for ozone, NO$_x$, and NO$_y$ reservoir species change between the scenarios described in Section 3.6.2. The com-

parison is here made with respect to a case where no contrail forms, thus yielding the contrail-related chemical perturbation.

Contrail-induced impacts on ozone production are small, with an overall difference of ~10% with respect to the pure gas-phase response. The NO$_y$ partitioning is affected to a greater extent. The ice crystal surface area in the plume provides a surface for rapid, heterogeneous conversion of N$_2$O$_5$ to HNO$_3$. The descending, crystal-dense contrail therefore rapidly converts NO$_x$ into reservoir forms, with the overall rate and extent of conversion increasing with black carbon emissions. At a black carbon mass emission index of 20 mg/kg$_{\text{fuel}}$, HNO$_3$ concentrations are 170% greater after 24 hours than for the baseline case. We also find that concentrations of NO$_x$, corresponding to the overall NO$_x$ "survival fraction", are lower after 24 hours with higher black carbon emissions owing to the heterogeneous reactions on the surface of ice crystals.

Chemical species have an asymmetric profile across the contrail height. A greater aerosol surface area in the lower side of the plume leads to larger chemical rates through heterogeneous chemistry. The extent of the asymmetry depends on ice crystal microphysical parameters and therefore on meteorological conditions as well as aircraft parameters. Horizontally-integrated chemical perturbations of O$_3$, HNO$_3$

**Table 8.** Effect of changing the black carbon mass emission index on emission conversion factors in the upper troposphere

| $EI_{BC}$ $(mg/kg_{fuel})$ | $ECF_{O_3}$ | $ECF_{NO_x}$ | $ECF_{HNO_3}$ | $ECF_{N_2O_5}$ |
|---|---|---|---|---|
| 0[1] | 0.68 | 0.78 | 0.12 | 0.041 |
| | - | - | - | - |
| 10 | 0.62 | 0.73 | 0.32 | -0.042 |
| | -8.8%[2] | -6.4%[2] | +170%[2] | -200%[2] |
| 20 | 0.60 | 0.72 | 0.38 | -0.063 |
| | -12%[2] | -7.7%[2] | +220%[2] | -250%[2] |
| 40 | 0.58 | 0.71 | 0.45 | -0.093 |
| | -15%[2] | -9.0%[2] | +280%[2] | -330%[2] |
| 60 | 0.56 | 0.70 | 0.50 | -0.11 |
| | -17%[2] | -10%[2] | +320%[2] | -370%[2] |

Values are computed 24 hours after emissions. The emission time is set to 8:00 AM.
[1] A black carbon mass emission index of 0 mg/kg$_{fuel}$ denotes a case in which no contrail forms.
[2] Relative changes with respect to the baseline case where no contrail forms.

and $N_2O_5$ in the upper troposphere can be found in Figure S5 of the Supplementary Information.

### 3.6.4 Effects of stratospheric contrails

We carry out an additional simulation to quantify how the radiative and chemical impact of contrails is different when forming in the stratosphere. Although unlikely due to the dry conditions of the stratosphere, any contrails that survive the initial formation stage would be likely to persist for significant periods due to the low mixing rates in this region of the atmosphere.

The relative humidity profile is kept identical. We find that the lower temperature lapse rate in the stratosphere leads to a smaller contrail ice mass, with optically thinner contrails. This is explained by lower temperatures in the lower part of the contrail core and in the fallstreak. As explained in Unterstrasser and Gierens (2010a), a lower temperature reduces the effective crystal radius and optical thickness of the contrail. In the stratosphere, we find that contrails reach horizontal dimensions of 8 to 15 km compared to 5 to 10 km in the upper troposphere after 4 hours.

The nitric acid emission conversion factors are also greater compared to the tropospheric case. In some cases, the $HNO_3$ ECF reaches values above one as ambient, short-lived $N_2O_5$ is converted to long-lived $HNO_3$. At a pressure of 100 hPa, a black carbon mass emission index of 10 mg/kg$_{fuel}$ leads to greater absolute contrail-induced impacts compared to typical subsonic altitudes. The relative changes are however smaller, corresponding to a 73% and -69% change in the $HNO_3$ and $N_2O_5$ perturbations respectively compared to the baseline case. The same change in emission causes a decrease of the ozone perturbation per unit of $NO_x$ emitted by 21% compared to 12% for the tropospheric case. Figure

S6 of the Supplementary Information displays horizontally-integrated chemical perturbations of $O_3$, $HNO_3$ and $N_2O_5$ under stratospheric background conditions.

Aircraft-induced stratospheric cirrus clouds are found to have shorter lifetimes and lead to a smaller optical thickness compared to tropospheric altitudes assuming similar relative humidity profiles. Heterogeneous chemistry on ice crystals gains greater importance at lower pressures and shifts the $N_2O_5$-$HNO_3$ local equilibrium. Further work is needed to quantify how stratospheric contrails might differ from those that form in the troposphere.

## 4 Limitations and further work

Although our approach gives a significant improvement in terms of numerical fidelity compared to the instant dilution approach, additional work is needed to account for additional physical phenomena. For example, APCEMM accounts for mixing with ambient air and the effect of wind shear on the plume. However, it does not capture the enhanced diffusion from the small-scale vortices generated by the wind shear. We also consider a highly simplified approach to simulate the aircraft-induced vortex dynamics and the vertical motion of the plume due to radiative heating of the contrail. APCEMM does not explicitly model turbulence in the vicinity of the contrail, which induces fluctuations in the relative humidity field of the order of $\pm 5\%$ around saturation (Unterstrasser and Gierens, 2010a; Gierens and Bretl, 2009). Contrail regions with greater ice water content experience more heating, thus leading to a non-uniform adiabatic contrail uplift and a vertical stretching of the contrail. Additionally, we use a simplistic representation to model the role of organic species on the early-plume microphysics. Future work is planned to explicitly model these processes in APCEMM.

These results are also isolated to the 24-hour period immediately following passage of an aircraft. In order to translate these results into an assessment of the net global impact of aviation, we aim to implement the results of the plume-scale processing of aircraft emissions into a global atmospheric chemistry-transport model in a consistent manner with the chemistry and microphysical processes in place. However, doing so in a fashion that can be easily maintained will constitute a non-trivial software engineering challenge.

## 5 Conclusions

We develop a parameterized aircraft plume model to simulate chemistry in an aircraft wake. This study shows that neglecting the non-linear plume-scale processes leads to inaccuracies in the assessment of $O_3$ perturbations and of the conversion of $NO_x$ to reservoir species.

We use APCEMM to quantify the 24-hour atmospheric chemical response to aircraft emissions, and how it differs from the results simulated under the "instant-dilution" ap-

proach typically used in global models. We also assess how this depends on ambient conditions, aircraft parameters and fuel properties. Based on a parameter sweep for typical cruise conditions, we find that the instant dilution assumption leads to greater ozone production compared to the plume model, with errors of up to ~200% at cruise altitude. This is due to plume-scale effects not resolved in the instant-dilution approach. In the plume model, the release of emissions into a small volume leads to $O_3$ depletion (through NO titration) and a $HO_x$-limited regime that last up to 5 hours after emissions. The lack of $HO_x$, which never occurs in the instant dilution approach, causes enhanced ozone production and a reduced fraction of $NO_x$ remaining in the plume. We also quantify the role of wind shear and atmospheric diffusion.

Our approach also permits us to explicitly model the formation and effects of condensation trails. These affect the in-plume chemical response through heterogeneous reactions on the surface of ice crystals. Such plume-scale processes are not accounted for in almost all global-scale modeling approaches, but are needed in order to fully understand the role of aircraft particulate emissions in upper tropospheric chemistry. We find that a 50% reduction of the mass of emitted black carbon (assuming a baseline case of 20 mg/kg$_{fuel}$) leads to a decrease of 16% in the aircraft-attributable $HNO_3$ perturbation, while ozone and nitrogen oxides increase by ~3% and 1.5% respectively. This is accompanied by a 15% decrease in the 2-hour optical depth of the contrail, such changes would not be captured in models that lack explicit plume modeling.

Previous studies assessing the impacts of aviation emissions have released emissions at the grid-scale level. We recommend that atmospheric models include a plume-scale treatment (or parameterization thereof) of aircraft emissions to compensate for these errors. Parameters should include meteorological conditions, local atmospheric composition, flight properties as well as engine and fuel characteristics. This is expected to significantly affect the estimated contribution of aircraft emissions to atmospheric $NO_x$ and ozone.

## Appendix A: Growth during the early plume phase

### A1   Growth of ice particles during the early phase

As previously stated, the rate of change in the ice mass of a particle, $m_p$, is given by

$$\frac{dm_p}{dt} = H_p^{act}(m_p) \times 4\pi C_p D_{v,eff} \left( P_{H_2O} - P_{H_2O}^{sat} \right), \qquad (A1)$$

where $H_p^{act}$ is a function accounting for nucleus activation (equation A3), $C_p$ is the ice crystal capacitance (equal to the particle radius $r_p$ for spherical nuclei), $D_{v,eff}$ is the effective water vapor diffusion coefficient in air (equation A4), and $P_{H_2O}$ the water partial pressure.

The saturation vapor pressure of water $P_{H_2O}^{sat}$ is calculated as

$$P_{H_2O}^{sat} = \exp\left( r_K/r_p \right) \times P_{H_2O}^{flat} \qquad (A2)$$

where $r_p$ is the particle radius (in nm), $r_K$ is the Kelvin radius (set here to 2.3 nm), and $P_{H_2O}^{flat}$ is the saturation vapor pressure over a flat surface. The factor $\exp(r_K/r_p)$ represents the effect of particle curvature (the Kelvin effect), increasing the apparent vapor pressure over a convex surface relative to a flat surface (Lewellen et al., 2014). $p_{H_2O}$ is calculated as a function of temperature only (Pruppacher et al., 1998).

Growth is only permitted if particles are activated, meaning that they either already have an ice coating or in air that is locally supersaturated. This is characterized through the variable $H_p^{act}$, as

$$H_p^{act} = \begin{cases} 1 & \text{if } m_p > 0 \text{ or } RH_{w,loc} \geq 1 \\ 0 & \text{otherwise.} \end{cases} \qquad (A3)$$

where $RH_{w,loc}$ is the local relative humidity with respect to liquid water (Picot et al., 2015). $D_{v,eff}$ accounts for latent heat effects, and is calculated as

$$D_{v,eff} = \frac{D_v \times \beta\left( r_p \right)}{\frac{D_v L_s P_{H_2O}^{sat}}{\kappa_d T}\left( \frac{L_s}{R_v T} \right) + R_v T}, \qquad (A4)$$

where $L_s$ is the latent heat of sublimation and $\kappa_d$ the thermal conductivity of air. The function $\beta$ accounts for the transition in uptake behavior between the gas kinetic (Kn $\gg$ 1) to the diffusional regime (Kn $\rightarrow$ 0), and is calculated as

$$\beta(r_p) = \left( \frac{1}{1+r_p/\lambda_v} + \frac{2D_v C_p}{\alpha v_{th,v} r_p^2} \right)^{-1} \qquad (A5)$$

with $\alpha$ being the deposition coefficient. For deposition of water molecules on ice particles, we take $\alpha = 0.5$, which is in agreement with laboratory and field studies from Haag et al. (2003). The mean free-path of vapor molecules $\lambda_v$ and the diffusion coefficient of vapor in air $D_v$ are functions of the local temperature and pressure and are computed according to relations from Pruppacher et al. (1998).

### A2   Growth of liquid particles during the early phase

In the early plume phase, liquid particles form as the plume undergoes rapid cooling. As stated previously, we use the parameterization from Vehkamäki et al. (2002) to compute nucleation rates, cluster size and composition.

Throughout the plume lifetime, the growth of liquid particles is affected by coagulation, which dominates the early-plume phase because of the initially high aerosol particle concentrations. Coagulation is a volume-conserving process that decreases the aerosol number concentration.

Coagulation of newly-formed aerosols and scavenging by black carbon and ice particles take place on different

timescales. The coagulation timescale of particles of radius $r_1$ can be evaluated from $t_{\text{coa}} = 1/(K(r_1, r_2)n_2)$, where $n_2$ and $r_2$ are the number density and the radius of the scavenging particles. As shown in the Supplementary Information, self-coagulation of liquid aerosols ($r_1 = 0.5$ - $10$ nm) occurs on a timescale of a few seconds to minutes, assuming a typical number density between $10^6$ and $10^{10}$ particles/cm$^3$. Similarly, scavenging by black carbon and ice particles happens over timescales that are of the same order of magnitude, assuming a radius of 50 nm (1 μm) for black carbon particles (for ice crystals, respectively) and a number density of $10^3$ molecules/cm$^3$. On the other hand, self-aggregation of black carbon particles and ice crystals occur on timescales that are much longer, of the order of several hours.

## Appendix B: Coagulation kernel

The coagulation kernel represents Brownian diffusion, convective Brownian diffusion enhancement, sedimentation-induced aggregation, and turbulent inertial motion, as well as enhancement due to turbulent shear. The kernel is described by the following equations, taken from Jacobson (2005). We here consider a particle of size $i$ coagulating with particles of size $j$.

The Brownian collision kernel is described by:

$$K_{i,j}^{\text{B}} = \frac{4\pi(r_i + r_j)(D_{p,i} + D_{p,j})}{\frac{r_i + r_j}{r_i + r_j + \sqrt{\delta_i^2 + \delta_j^2}} + \frac{4(D_{p,i} + D_{p,j})}{\sqrt{\bar{v}_{p,i}^2 + \bar{v}_{p,j}^2}(r_i + r_j)}}, \quad \text{(B1)}$$

where $r$ is the particle radius, $D_p$ the particle diffusion coefficient, $\delta$ the mean distance from the center of a sphere reached by particles leaving the sphere's surface and traveling a distance equal to the particle mean free path and $\bar{v}_p$ the thermal speed of a particle in air.

The convective Brownian diffusion enhancement kernel is defined by:

$$K_{i,j}^{\text{DE}} = \begin{cases} K_{i,j}^{\text{B}}0.45\text{Re}_j^{1/3}\text{Sc}_{p,i}^{1/3} & \text{if } \text{Re}_j \leq 1, r_j \geq r_i \\ K_{i,j}^{\text{B}}0.45\text{Re}_j^{1/2}\text{Sc}_{p,i}^{1/3} & \text{if } \text{Re}_j > 1, r_j \geq r_i, \end{cases} \quad \text{(B2)}$$

where Re and Sc are the particle Reynolds and Schmidt numbers.

The sedimentation-induced aggregation kernel is described by:

$$K_{i,j}^{\text{SI}} = E_{\text{agg}}\pi(r_i + r_j)^2 \mid V_{\text{f},i} - V_{\text{f},j} \mid, \quad \text{(B3)}$$

where $E_{\text{agg}}$ is a collision efficiency.

The turbulent inertial motion and turbulent shear kernel are defined by:

$$K_{i,j}^{\text{TI}} = \frac{\pi\varepsilon_d^{3/4}}{g\nu_a^{1/4}}(r_i + r_j)^2 \mid V_{\text{f},i} - V_{\text{f},j} \mid \quad \text{(B4)}$$

$$K_{i,j}^{\text{TS}} = \left(\frac{8\pi\varepsilon_d}{15\nu_a}\right)^{1/2}(r_i + r_j)^3, \quad \text{(B5)}$$

where $\varepsilon_d$ is the rate of dissipation of turbulent kinetic energy, $g$ the acceleration due to gravity and $\nu_a$ the kinematic viscosity.

The total coagulation kernel is equal to the sum of each individual kernel:

$$K_{i,j} = K_{i,j}^{\text{B}} + K_{i,j}^{\text{DE}} + K_{i,j}^{\text{SI}} + K_{i,j}^{\text{TI}} + K_{i,j}^{\text{TS}}. \quad \text{(B6)}$$

## Appendix C: Plume-averaged NO$_x$ chemical rate

We assume that the conversion of NO$_x$ to reservoir species is dictated by the daytime conversion pathway through the following reactions:

$$\text{NO}_2 + \text{OH} \xrightarrow{\text{M}} \text{HNO}_3 \quad \text{(CR1)}$$

$$\text{NO}_2 + \text{HO}_2 \xrightarrow{\text{M}} \text{HO}_2\text{NO}_2 \quad \text{(CR2)}$$

The chemical reaction rate can be written as:

$$\frac{d[\text{NO}_2]}{dt} = -(k_1[\text{OH}] + k_2[\text{HO}_2])[\text{NO}_2]$$

$$\frac{d[\text{NO}_2]}{dt} = -k_{\text{eff}}[\text{HO}_x][\text{NO}_2]$$

where HO$_x$ has been defined such that $[\text{HO}_x] = \frac{k_1[\text{OH}] + k_2[\text{HO}_2]}{k_1 + k_2}$.

We assume that the concentration field at a fixed point can be expressed as the sum of spatially-averaged quantity and the instantaneous fluctuation, such that:

$$[\text{NO}_2] = \overline{[\text{NO}_2]} + [\text{NO}_2]'$$

$$[\text{HO}_x] = \overline{[\text{HO}_x]} + [\text{HO}_x]'$$

The chemical conversion rate of NO$_x$ can therefore be written as:

$$\frac{d\overline{[\text{NO}_2]}}{dt} = -k_{\text{eff}} \times \overline{[\text{HO}_x][\text{NO}_2]}$$

$$= -k_{\text{eff}} \times \left(\overline{[\text{HO}_x]} \times \overline{[\text{NO}_2]} + \overline{[\text{HO}_x]' \times [\text{NO}_2]'}\right)$$

The first term on the right hand side leads to a net depletion. NO$_2$ is an emitted species. Therefore, the NO$_2$ fluctuation is positive in the core of the plume, while it is negative far away. HO$_x$, however, gets depleted to form HNO$_3$ and HO$_2$NO$_2$. Therefore, $[\text{HO}_x]' \leq 0$ in the inner plume and $[\text{HO}_x]' \geq 0$ outside of the core. Thus, the second term reduces NO$_x$ conversion and is proportional to the correlation of the fluctuations. If both fluctuations are negatively correlated, the depletion is reduced compared the case where the fields are uniform.

This explains why, for the same emission quantity, a small plume with large spatial fluctuations leads to a lower conversion compared to a large plume with smaller gradients.

## Appendix D: Validation of the contrail dynamics and microphysics

In this section, we compare the results of APCEMM's microphysical module to existing data from numerical simulations. Although observational data is sparse, direct comparison to in-situ measurements of both plume chemistry and ice would be a valuable future step.

We first compare the results of our simulation to Unterstrasser and Gierens (2010a) for different case studies. In all the following cases, the same meteorological background conditions (1-km deep layer) and emission characteristics as Unterstrasser and Gierens (2010a) are used. Using the same configuration, we then study the vertical distribution of the ice crystal number, surface and mass densities.

### D1 Comparison

Figures D1 and D2 show the ice crystal concentration, ice mass concentration, extinction, and relative humidity profiles at three points in time according to the simulations from Unterstrasser and Gierens (2010a) and APCEMM respectively. We perform all comparisons using definitions from Unterstrasser and Gierens (2010a). The spatial distributions of ice crystal number and mass densities at t = 2,000 s are in qualitative agreement, with a maximum horizontal discrepancy less than 500 m, although the small scale discrepancies resolved by large eddy simulations are not reproduced by APCEMM. APCEMM also predicts a reduced vertical extent, likely because of a more uniform, smaller ice crystal size distribution. At later times, APCEMM fails to capture the greater horizontal extent of the contrail seen in the results from Unterstrasser and Gierens (2010a). We believe that this is due to the faster initial settling that occurs in the large eddy simulations. This discrepancy is visible on the ice water content row in Figures D2 and D1 where APCEMM predicts a horizontal extent of 6 and 12 km at t = 8,000 s and t = 17,000 s respectively compared to 10 and 22 km according to the data from Unterstrasser and Gierens (2010a). In spite of this, the integrated contrail ice mass and number both differ by less than 10% at the three time instants.

### D2 Discussion

The time evolution of the ice number density, mass density and extinction shows similar trends between APCEMM and the high-fidelity simulations from Unterstrasser and Gierens (2010a), but there are some significant differences. Overall, APCEMM predicts a longer contrail lifetime with greater ice mass, particle number and extinction.

To fully understand the differences between these two models, a more detailed modeling study would be required. Qualitatively, we consider that there are a number of plausible explanations. One such explanation is that APCEMM does not represent the effect of turbulent motion, which causes fluctuations around supersaturation as described in Gierens and Bretl (2009). Such local fluctuations in a supersaturated region would induce changes in the total ice mass and cause the local sublimation of ice crystals, which would tend to reduce the contrail lifetime.

The different numerical schemes used by the two models will also drive differences. We apply a simplified model for the movement of species in APCEMM, using a spectral method to simulate diffusion with spatially-uniform coefficients, albeit time-varying and differing between the horizontal and vertical dimensions. APCEMM neglects 3-D effects, performing all calculations on a 2-D cross-section of the exhaust. By comparison, the model used by Unterstrasser and Gierens (2010a) performs 3-D large eddy simulations which will capture small and large scale dynamical features not resolved by APCEMM. They also use a different approach to represent ice microphysics, applying a two-moment scheme with an explicit correction for non-spherical ice particles.

Each of these factors will contribute to the differences in our comparison. Although APCEMM is designed to provide an intermediate-complexity approach which bridges the gap between (e.g.) Gaussian plume approaches and large eddy simulations, it would be helpful to understand exactly why and where the results of APCEMM differ from those of both the higher- and lower- fidelity approaches. Future efforts to improve APCEMM's simulation of contrail dynamics would benefit from direct comparison to LES simulations in which different components of the simulation could be modified to identify the role of (e.g.) turbulent fluctuations as opposed to differences in ice microphysics. It would also be useful to directly simulate a simulated 2-D flow field in APCEMM and thereby understand the degree of difference introduced by the simplified approach to transport.

### D3 Contrail phases

This section aims to describe the different phases of the contrail as described by the simulations from APCEMM.

The peak particle number concentration decreases over time from an initial value of ~10 #/cm$^3$. The fallstreak, particularly noticeable at t = 17,000 seconds in Figure D1 and Figure D2, represents a significant fraction of the total ice mass, especially considering that it is characterized by a low number of ice crystals (~20 times lower than the peak particle number). Moist air is entrained at the periphery of the contrail leading to large spatial heterogeneities in the ice mass distribution. Given that the fallstreak is predominantly composed of large crystals, it can survive below the saturation depth until these crystals fully sublimate.

The extinction, as plotted on Figure D2, represents the lidar-detectable region of the contrail (with a detection threshold of $\chi_0 = 10^{-5}$ m$^{-1}$). As pointed out in Unterstrasser and Gierens (2010a), non-negligible regions of the contrail cannot be detected through lidar measurements, as large ice crystals have a low extinction parameter.

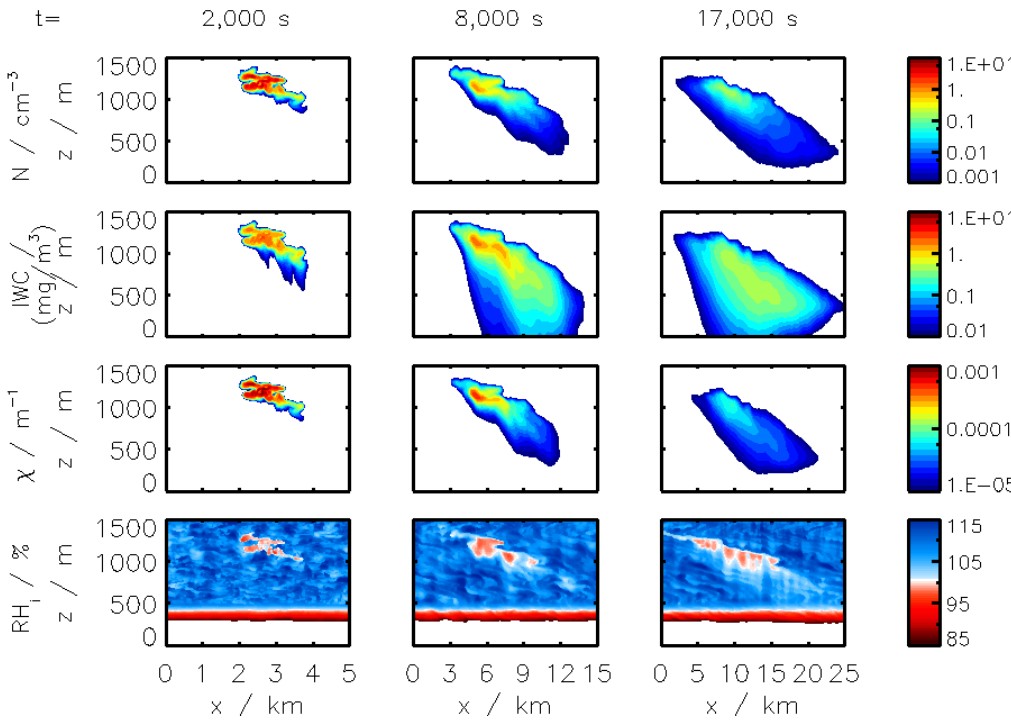

**Figure D1.** Results from the simulations of Unterstrasser and Gierens (2010a) showing snapshots of ice crystal number concentration, ice water content, extinction and relative humidity with respect to ice. Figure adapted from Unterstrasser and Gierens (2010a), courtesy of S. Unterstraßer.

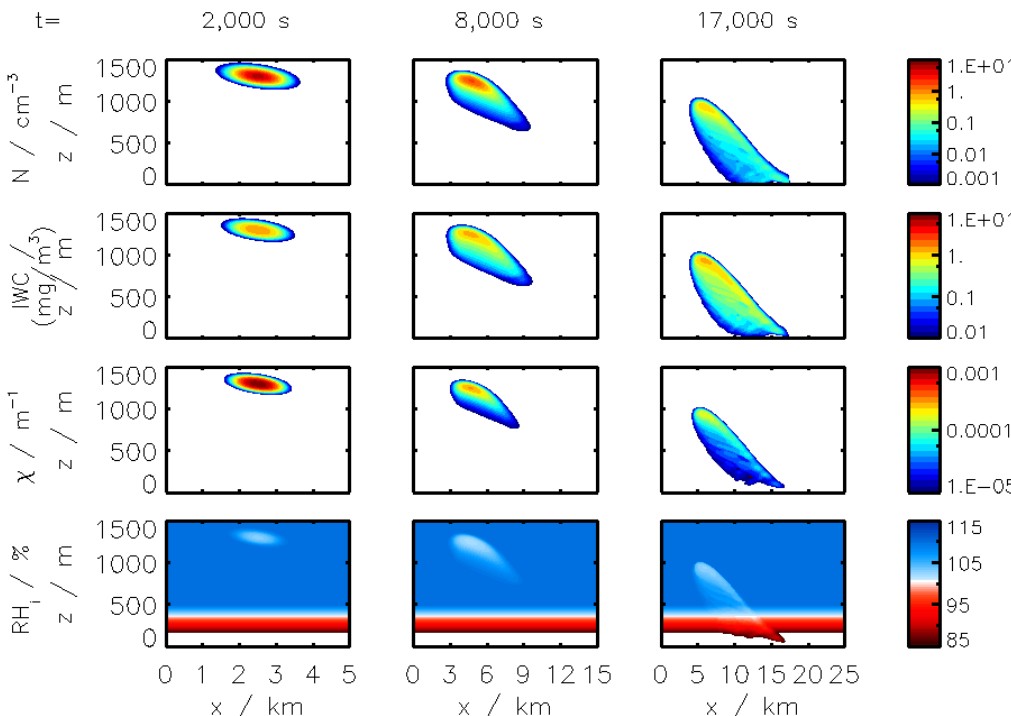

**Figure D2.** Ice crystal concentration N, ice mass concentration IWC, extinction $\chi$ and relative humidity $RH_i$ for the same conditions as described in Unterstrasser and Gierens (2010a). The flight level is at a vertical distance of 1,300 m.

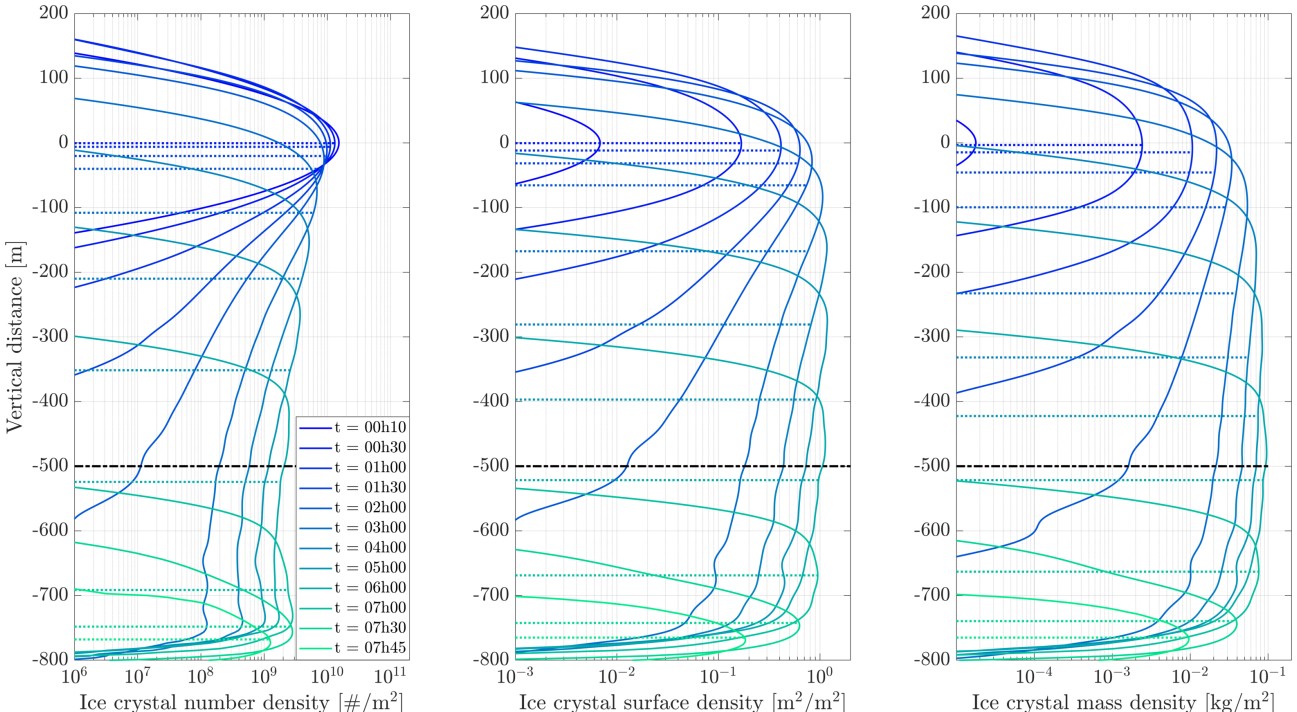

**Figure D3.** Time evolution of horizontally-integrated ice crystal number, surface and mass densities. The color gradient refers to the contrail age. The supersaturation region extends from ±500 m around the flight altitude (represented as a dash-dotted black line). The median of each distribution is plotted as a dotted line.

The relative humidity profiles in Figure D2 show that the core of the contrail is characterized with a uniform region, close to ice saturation. As the fallstreak sublimates under the saturation depth, it releases its water content at lower altitudes than the flight-level contributing to the dehydration of the upper-troposphere/lower-stratosphere.

Figure D3 displays the time evolution of horizontally-integrated ice crystal densities assuming a 1-km deep saturation layer. In the first hour, the particle densities are approximately distributed symmetrically around the flight altitude as settling has not contributed significantly. The ice mass continually increases until 5 hours after emissions, leading to a vertical stretching of the contrail through gravitational settling of large particles. At that point, the ice mass density is concentrated at, or slightly below, the saturation depth (where the saturation reaches 100%) and starts to decrease until the contrail has fully sublimated at 7h45.

The colored dashed lines on Figure D3 show the median altitude of the ice crystal number, surface and mass density distributions respectively, at each time instant. Initially, all median values lie around the flight altitude. After ~3 hours, the median values start to "settle" at different speeds with the ice mass median altitude dropping the fastest, followed with the surface density and number density. This is due to the presence of a contrail core rich in small ice crystal particles (~1 μm) and a fallstreak composed of larger and fewer nuclei contributing to a large fraction of the ice mass.

Figure S7 of the Supplementary Information displays the same content as Figure D3 assuming a supersaturated region extending between ±200 m. The same conclusions as in Figure D3 can be drawn. After 6 hours, the contrail has fully sublimated and it appears from Figure S7 of the SI that the median altitudes converge towards the saturation depth.

*Code availability.* APCEMM is openly available since April 2020 at https://github.com/fritzt/APCEMM. For this work, APCEMM v5 was used, a copy of which has been permanently archived (DOI: 10.5281/zenodo.3755701).

*Author contributions.* T.M.F. implemented and ran the model, and wrote the manuscript. All authors were involved in study design, model validation and improvement, and manuscript review, editing, and finalization.

*Competing interests.* The authors declare no competing interests.

*Acknowledgements.* This work was supported by NASA grant number NNX14AT22A. We would like to thank S. Unterstraßer for providing data for comparison, as well as for the useful discussion regarding the simulation of the vortex sinking process.

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
