# Peer review of "The role of plume-scale processes in long-term impacts of aircraft emissions"

_Atmospheric Chemistry and Physics, 2019_

## Referee Comment (RC1) · Anonymous Referee #1 · 13 Aug 2019

This is an interesting and important analysis that emphasizes the potential impacts of plume processing of aircraft emissions prior to their incorporation into climate models at their grid scales. This is an important point that has been made previously on a number of occasions but is often ignored and not included in analyses.

I find some parts of the modeling results to be useful, and worthy of publication. However, there are a number of issues that should be addressed before this manuscript be accepted for publication, in my opinion. The gaseous chemistry regarding ozone formation is compelling, useful, and is quoted as agreeing with prior analyses. This is useful confirmation of the importance of plume processing for ozone impacts. The contrail impacts are also of interest in how particle properties are affected by plume processing. However, the details of the way the contrail modeling has been done need

to be qualified to a greater degree, due to assumptions that are made (mono-modal soot distribution) and implied (dependence, or lack thereof of, ofwater uptake on particle surface composition).

1) The approach shown for volatile PM (nucleation and growth of new particles, and uptake on soot particle coatings), seems incomplete and thus potentially flawed. No specific results are shown in plots nor discussed, and it is not clear that such results impact the chemistry nor contrails results that ARE shown. This perspective will be discussed further below, where I suggest removing or discussing in a much different way. 2) The contrail modeling has made some simplifications that may impact the results that the authors claim to be important. They need to discuss in more detail how the assumptions might qualify their results and the claimed quantification of the effects that they observe. 3) There are a number of more minor wording or presentation issues that I will identify below, along with some suggestions for how they might be addressed.

1. In section 2.2.3, page 6, line 15, "Soot and ice particles can also grow by condensation of water vapor, sulfuric acid, and nitric acid . . .". Experimental results show that the growth of particle mass in aircraft exhaust plumes is dominated by organic species (and nitric acid is not usually observed in the initial plume regions). Leaving out organic species is leaving out a primary contributor to the mass of these newly formed particles (prior to water deposition in contrail formation) as well as the coatings on soot particles. Thus, the presented microphysical approach is missing the major contribution to mass. (Yet the authors do note that volatile organics are in the exhaust, section 3.1, page 10, line18.)

However, there are no results presented in the paper that show the importance of this microphysical processing. Neither results showing newly nucleated sulfate aerosol nor the coatings on the soot particles (and their composition) are presented in the paper. It is not clear from the material presented how the eventual uptake of water is dependent on the condensed matter due to these species. Is the later water uptake affected by the surface composition? If not, there seems to be no impact of this analysis on the

contrail results presented later in the paper. No size distribution results are shown, so it is not clear how the soot distribution and newly nucleated particles make up the input to the downstream mature plume modeling, and how they affect the subsequent analysis.

It is worth noting that many other modeling studies suggest that the "nucleation mode" is not important for contrail processes when the soot mode is present, due to the larger size of the soot mode. Thus, there is a basis for questioning the importance of this smaller mode. The question of the compositional changes of the soot surface due to condensation seems open, but unaddressed by the present study.

Unless more information is provided, it seems that this is an incomplete analysis that has limited bearing on the problem at hand, and it does not appear that the model has a means to include the effects of this analysis on the key results presented. I suggest this part of the analysis be removed or completely re-described.

2. In a related issue, the modeling assumes (section 2.2.3, page 6, line 5) that the soot distribution is a mono-modal distribution ("a single representative particle"). While that may make sense to define a more computationally tractable problem, the microphysical modeling discussed in 1. above seems to require a binned size distribution approach (page 7 line 3), so why is it necessary to force the soot to be mono-modal? But if the response to issue 1. above is to remove the volatile particle modeling, then perhaps the mono-modal soot distribution may be justifiable to simplify the computations.

However, if the approach is to accept a more limited modeling approach, based on a mono-modal soot distribution as has been done before (as referenced by the authors), then another separate question arises. The results show important differences due to differences in the fate of large particles versus small particles in the later plume processing (section 3.5.1, line 12 et seq.). If the initial soot distribution is mono-modal, the contrail particles will also be mono-modal for those particles that have had the same history (i.e., in the same ring). There needs to be more discussion of how the history

of particles might generate a size distribution that differs from the initial mono-modal soot size distribution, if this is, indeed, what generates a polydisperse contrail particle size distribution.

3. Presentation issues and typos: a. In the introduction (page 2, line 29 - 30), "aviation is . . . the only direct, significant source . . .", what about rockets? May not be as large, but rockets may still be significant.

b. Also, in the introduction: this is not meant to be a review article, but it might be worth mentioning that the importance of plume processing has a history that goes back to CIAP (CIAP monograph 3, 1975, DOT-TST-75-53, chapter 2 and references therein) and, {especially since the manuscript is a NASA sponsored study}, to NASA (Atmospheric Effect of Aviation: First Report of the Subsonic Assessment Project, 1996, NASA Ref. Pub. 1385, chapter 4 and references therein)

c. Figure 1. As a schematic, this figure seems to address only the 2.3 mature plume modeling part. There is an inset box that discusses the plume box model processing, but there is no schematic representation of the box model in the artwork. And there is no equivalent inset box that describes the discretized rings in the figure.

If the box model inset box were removed from the existing figure, I would suggest this figure would sit better in section 2.3, where the mature plume modeling is discussed. It has little schematic value for the box model as drawn so doesn't provide much benefit as placed in section 1. If, on the other hand, the figure was adjusted so that there were schematic aspects and inset boxes for both parts of the model, then perhaps a redrawn version might have reason to remain in section 1.

d. Section 2.1, page 4, line 13. "The output of this box model . . ." This sentence is confusing. The antecedent to "this box model" doesn't exist, since the model hasn't been mentioned yet, and is described in the subsequent section.

The preceding paragraph describes the physical phenomena to be addressed, but

there is no mention of the box model that will be used. One solution would be to briefly mention how a box model (to be described in detail later) will be formulated to capture the elements described.

Another solution would be to drop that sentence and pick it up later after the two models are discussed. If this approach is taken, then the material in these two paragraphs (last two paragraphs of section 2.1) would just be discussing the physical phenomena in the two regimes and leave the box and ring models' discussion for the later sections. (If this approach is taken, the title of 2.1 might need to be adjusted.)

As written, the sentence is confusing, referencing models that haven't been introduced yet.

e. Section 2.2.2, page 5, line 22, Tremmel, (and by Lukachko et al., 1998 JGR 103, and in 2008, J. Eng. Gas Turbines and Power, 130, 2008) found that the conversion of S(IV) to S(VI) occurred primarily in the engine's turbine, and not in the plume to a significant degree. Later processing in the atmosphere happens also, but at time-scales much longer than the initial plume being addressed by this study.

f. Section 2.4.1, page 9, line 21. "equipped with GEnx engines". In what sense is this engine represented in the model? In section 3.5.3, page 19, line 8, a soot emission index (EI) is given as 0.06 g/gfuel. (I assume this is a typo, and it is meant to be 0.06 g/kgfuel, or 60 mg/kgfuel). This seems very high for the GEnx engine, especially at cruise at altitude. This (even after correcting for the typo!) is 1.5 times the {high} value used in the 1999 IPCC report of 0.04 g/ kgfuel. And where was the EI soot data obtained?

In addition, is the NOx EI chosen to be representative of the GEnx from the ICAO Databank?

g. Section 3.5.1, page 16, line 1. Supersaturations of 102% to 108% are quoted, but is this respect to water or ice? (And is 108% observed in the natural atmosphere?)

---

## Referee Comment (RC2) · Anonymous Referee #2 · 2 Jan 2020

The study introduces a new model called APCEMM, which is designed for simulating chemical processes in aircraft plumes and also considers the effect of contrails on plume chemistry. My impression is that authors do not have a strong background in atmospheric physics. This becomes apparent in quite a few passages of the manuscript. I recommend that the author team strengthens their expertise in atmospheric physics before revising the manuscript and redesigning the APCEMM. The manuscript could become publishable only after major revisions.

**General comments**

As already stated in the summary above, I doubt that the implemention of the various

atmospheric processes is done correctly. Moreover, sometimes processes or phenomena are included, that are irrelevant and only pretend to increase the level of detail in the model. Comments on the physical soundness of your apprach are listed in the section "Specific comments".

Here, only several general comments on terminology and language are made.

- Even though often written and read, it is wrong: Temperature is not cold or warm. It is low or high and tells us if something is hot or warm. Please check the whole manuscript.

- E.g. formula (2), (4)
  I find it awkward to provide units for each quantity. This somehow pretends that the formulae are only valid in conjunction with exactly those units. This is certainly not the case. I understand that supplying units helps the reader to make a first check of the correctness of the formulae. But the way it is presented, it is misleading.

- Aerosol is a gas with suspended particles. If you refer to the particles only, better use the term aerosol particle. Sometimes you use the term aerosol even for ice crystals (in particular, last paragraph of section 2.2.3). I would make a clear distinction between aerosol particles and ice crystals.
  http://glossary.ametsoc.org/wiki/Aerosol

- Please use the terms deposition, sublimation, condensation and evaporation consistently.

- p.4. l.31: those THAT were emitted
  See https://www.wisegeek.com/what-is-the-difference-between-that-and-which.htm
  Please check the whole manuscript.

- Concerning statements in the abstract like "evaporate ˜9% faster and are 14% optically thinner"
  Given the accuracy of the (still) simplified treatment, I would prefer to leave out such precise numbers (in the abstract). How much are they worth? If you use another definition of optical thickness or define the time of evaporation slightly differently, I am sure you can get anything between $5\%$ and $20\%$.
  Table 7 and 8
  Given the uncertainties, it is not meaningful to provide numbers like $-5.35\%$ with two decimal places. Please round them to a reasonable precision. Similarly the value 1.2581 in Table 4 it "too" precise. Please go through the whole manuscript.

**Specific comments**

p.2, l.39:

what are "local aerosol clouds"?

p.4, l.5:

The ambient temperature at cruise altitude is not 280K. Climate change is not that fast :-)

p.4, l.14:

Your statement implies that the coherent vortex flow field is just turbulence which is not the case. Please better describe how the vortices break up. Paoli and Shariff (2016) is a good source of information for contrail-specific processes and phenomena.

p.4., l.16:

Schumann, 2012 is a long paper. Which formula do you use? Do assume that the vertical motion is constant over time during the vortex phase?

Note that in a stably stratified atmosphere, large parts of the vertically displaced plume rise back to the original emission altitude after vortex breakup due to buoyancy. As some portion of the ice crystals (or some other tracer) remains at lower levels, the vortex sinking causes a strong and fast vertical plume expansion (compared to time scales of natural processes). It seems that this effect is not considered in your model.

p.7, l.45:

Who is the user in this case and is supposed to choose a value for $D_h$?

p.8, first paragraph:

The way you include the effect of radiation is not correct. Contrail parts with the highest IWC are usually heated the most. This heating causes an uplift of those contrail parts during which the air cools adiabatically (again proportional to $\Gamma_d$). Assuming the atmosphere is stably stratified, the local uplift is sustained as long the ambient temperature is below the temperature of the contrail patch. So for typical stratification values, the initial heating actually translates in a cooling of the contrail! As the heating in the contrail fall streak is usually not that strong, radiation leads to a contrail vertical stretching.

See introductory textbooks on lifting condensation level for the general physics (unlike to warm clouds however, the latent heating effects are not that important in ice clouds and the moist-adiabatic lapse rate $\Gamma_m$ is roughly the same as $\Gamma_d$).

p.8, l.11:

Could you describe in a few words what KPP is.

p.8:

The inclusion of ice aggregation seems very sophisticated (iterative determination of coalescene efficiency) compared to the treatment of other processes in the model. But more aggravating is the fact that the cited Beard & Ochs paper deals with precipitation drops and not ice crystals. Please refer to literature refering to aggregation, not coalescene of liquid cloud droplets. Section 4 of Sölch and Kärcher (2010) could be a good starting point to dive into the physics behind the aggregation efficiency.

p.9., l.21:

What specifications of the chosen aircraft and engine type appear in the model, and which ones matter in the end? Are your results only valid for this specific aircraft/engine combination or can your findings be generalised?

p.9:

Why is it important to evaluate the error $\epsilon_X$ after 24 hours, hence at the same time of the day as the initialisation was done? Wouldn't it be better to evaluate the error at a time where the spatial dimensions of the APCEMM-modelled plume and the BOX-modelled plume are similar? Refering to Figure 2, you state in the text that 1.2 kg or 0.2 kg ozone is produced, giving a fatcor 6 difference between the two modelling approaches. A variation of the evaluation time would dramatically change this factor. Using an earlier point in time (e.g. 2 hours earlier at time 6:00) the factor would be much higher. Could you make clearer the strategy behind your evaluation effort.

Remarks on section 3.5.1:

- Several choices of the background conditions are not reasonable at all. Section 2.2 of Kärcher et al. (2009) may help to choose more realistic cases.

- A 10 cm/s cooling over 24 hours translates into a lifting by 8 km and an adiabatic cooling by 80K. This is not realistic. Compared to this, the 0.1 K diurnal

temperature variation can be safely neglected.

- Persistent contrail formation is likely to occur in a $RH_i$-range of $100\% - -140\%$ (above the upper limit, natural cirrus formation could not be neglected)

- Does the depth of supersatured layer remain constant over time? Given the prescribed uplift, the initially subsatured layer above/below the supersatured layer would eventually become supersatured as well and the supersaturated layer would grow in size. If you used a time-constant 200m thick layer and included the radiation effect correctly, the contrail would move out of supersaturated layer into the drier air above. This leads to entrainment of dry air into the contrail which would then start to vanish.

- Given the quite thin supersaturated layer, the simulated contrail lifetimes of $> 10$h appear to be too large (in particular for $v_{UP} = 0$).

- You first make a link between in-situ loss and aggregation and few lines later you say in-situ loss is due to Ostwald ripening.

- line 21: Yes, it is usually warmer further down, but this is irrelevant here. Or do really want to say the ice crystal melt and become water droplets? What matters is that it is dry and the ice crystals sublimate and are lost eventually.

- If your contrail model produces reasonable results, could be checked by a comparison with the higher resolution model used in Unterstrasser & Gierens, 2010a,b. This comparion should be feasible to achieve with small extra effort as you anyway use mainly their definitions of contrail properties. It would only require to specify the same background conditions. This could show if your modeled contrail lifetimes and response to variations of $RH_i$ or $EI_{soot}$ are reasonable.

Table 8:

Wouldn't it be interesting and more insightful to compare all four simulations to the non-contrail simulation?

**Technical corrections**

p.3, l.14: the effects OF these

p.7, l.21/22: No complete sentence.

p.7, l.42: repetition of "a measure of local .."

p.11, l.24: please reformulate the sentence.

**References**

B. Kärcher, U. Burkhardt, S. Unterstrasser, and P. Minnis. Factors controlling contrail cirrus optical depth. *Atmos. Chem. Phys.*, 9(16):6229–6254, 2009. ISSN 1680-7316. URL http://www.atmos-chem-phys.net/9/6229/2009/.

R. Paoli and K. Shariff. Contrail modeling and simulation. *Annual Reviews in Fluid Mechanics*, 48(1):393–427, 2016. doi: 10.1146/annurev-fluid-010814-013619. URL http://dx.doi.org/10.1146/annurev-fluid-010814-0136190.

I. Sölch and B. Kärcher. A large-eddy model for cirrus clouds with explicit aerosol and ice microphysics and Lagrangian ice particle tracking. *Q. J. R. Meteorol. Soc.*, 136:2074–2093, 2010.

---

## Referee Comment (RC3) · Anonymous Referee #3 · 12 Jan 2020

This paper describes a detailed chemical and microphysical model that calculates the composition of aircraft plumes and its interaction with the atmosphere. This is certainly one of the most comprehensive studies that accounts for many chemical reactions and interactions between chemistry and microphysics of particles. The results obtained confirm previous studies, like for example the overestimation of ozone production due to NOx emissions when instantaneous dilution is adopted in CTMs and the role of heterogeneous processes that convert part of the emitted NOx into HNO3. All those results are interesting but in my opinion there are several issues that should be further discussed. Follows my major points.

1/ In the introduction it is claimed that almost all the CTM use the instantaneous dilution (ID) approximation to account for aircraft emissions. The authors seem to ignore the

many attempts of modelers to introduce plume processing in their large-scale model. The authors should refer to the review paper by Paoli et al. (2011) that gives a comprehensive review of the different approaches that have been followed to account for plume effects: Effective Emission Indexes, Emission Conversion Factors or Emission Reaction Rates. In the same paper are listed the CTM that use those parameterizations and the limitations of each approach.

2/ In contrast to the detailed chemistry and microphysics introduced in their model the authors have chosen a very simplified representation of the contrail and plume dynamics based on a simple diffusion model. This is a very crude approach that for instance ignores the details of contrail dynamics with the role of the crow instability, the formation of secondary vortices that maintain a significant fraction of the emissions at flight level with often persisting ice particles, and the complex nature of atmospheric turbulence and its interaction with radiation (e.g. Paoli et al., 2017). For instance wind shear and diffusion are considered as separate processes, although depending on the scale considered wind shear and diffusions are both the results of turbulence in a stratified atmosphere. Thus, it is very difficult to evaluate how the approximations made can influence the results of the model. Is it a balanced approach to introduce a detailed chemical and microphysics schemes with such a simplified dynamical scheme?

3/ Little is said on the validation of the model. Do we have measurements to confront to the model outputs? Can we constraint ozone formation rates and the conversion fraction of the emitted NOx to nitrogen reservoirs?

4/ In the conclusion it is concluded that plume effects are important and should be included in CTM. This is not new (see all the articles referenced by Paoli et al. 2011) and leaves open the difficulty to do that in a consistent manner with the chemistry and microphysics in place in those CTMs or GCMs. It is often because this consistency is difficult to preserve that the CTM and MCG modelers keep the ID approach despite its limitation.

[Figure]

References :

Paoli, R., O. Thouron, D. Cariolle, M. Garcia and J. Escobar. Three-dimensional large-eddy simulations of the early phase of contrail-to-cirrus transition: effects of atmospheric turbulence and radiative transfert. Meteorologische Zeitschrift, Vol. 26, 6, 597-620, 2017. Paoli, R., D. Cariolle, R. Sausen. Review of effective emissions modeling and computation. Geosci. Mod. Dev., 4, 643-667, 2011.

---

## Author Comment (AC1) · 23 Feb 2020

**Sebastian D. Eastham**
**Research Scientist,**
**Laboratory for Aviation and the Environment**
**Department of Aeronautics and Astronautics**

[Figure]

**Massachusetts Institute of Technology**
77 Massachusetts Avenue, Building 33-322A
Cambridge, Massachusetts 02139–4307, USA
http://lae.mit.edu

Atmospheric Chemistry and Physics Editorial Office

23rd February 2020

Dear Editor,

**Re: Revisions for "The role of plume-scale processes in long-term impacts of aircraft emissions" after submission to *Atmospheric Chemistry and Physics**

Thank you for considering our submission and for arranging the detailed and careful reviews. We appreciate the time and effort taken to provide feedback on this manuscript. We are grateful for the detailed comments received which have guided us to make significant improvements to the paper.

Please find below our responses (in **bold**) to each of the reviewers' comments (in *italics*). We have attached two copies of the manuscript: one "clean" version incorporating all changes, and a "markup" version in which all changes are highlighted. Line numbers below correspond to the "markup" version.

All simulations were re-run to account for changes suggested by the reviewers. The inclusion of a constant aggregation efficiency changed the total number of particles in bins with radii greater than 30 μm by at most 0.3%, as our previous estimate of aggregation efficiencies was close to the constant value prescribed in Sölch and Kärcher (2010). Furthermore, based on review of Kärcher et al. (2009), we have updated our contrail simulations to use a more realistic background condition. We now use the background meteorological conditions described in Unterstrasser et al. (2010), in which a background 50% relative humidity and a single supersaturated band (1 km in thickness) are prescribed. This change also enabled us to perform direct comparison of our results to those from Unterstrasser et al. (2010) in Appendix D of the manuscript. Neither modification has changed the qualitative conclusions of the paper.

**References:**

**Kärcher, Bernd et al. "Factors controlling contrail cirrus optical depth".** *Atmospheric Chemistry and Physics* **9 (2009): 6229-6254.**

**Sölch, Ingo, and Bernd Kärcher. "A large-eddy model for cirrus clouds with explicit aerosol and ice microphysics and Lagrangian ice particle tracking."** *Quarterly Journal of the Royal Meteorological Society* **136.653 (2010): 2074-2093.**

**Unterstrasser, Simon, and Klaus Gierens. "Numerical simulations of contrail-to-cirrus transition – Part 1: An extensive parametric study."** *Atmospheric Chemistry and Physics* **10.4 (2010): 2017-2036.**

*This is an interesting and important analysis that emphasizes the potential impacts of plume processing of aircraft emissions prior to their incorporation into climate models at their grid scales. This is an important point that has been made previously on a number of occasions but is often ignored and not included in analyses.*

*I find some parts of the modeling results to be useful, and worthy of publication. However, there are a number of issues that should be addressed before this manuscript be accepted for publication, in my opinion. The gaseous chemistry regarding ozone formation is compelling, useful, and is quoted as agreeing with prior analyses. This is useful confirmation of the importance of plume processing for ozone impacts. The contrail impacts are also of interest in how particle properties are affected by plume processing. However, the details of the way the contrail modeling has been done need to be qualified to a greater degree, due to assumptions that are made (mono-modal soot distribution) and implied (dependence, or lack thereof of, of water uptake on particle surface composition).*

**We thank the reviewer for their interest in our manuscript and are grateful for their detailed comments and in-depth analysis. We have modified the paper substantially in order to give more detail regarding the approach, and the justification of said approach. Please find our responses to each individual comment below.**

*1) The approach shown for volatile PM (nucleation and growth of new particles, and uptake on soot particle coatings), seems incomplete and thus potentially flawed. No specific results are shown in plots nor discussed, and it is not clear that such results impact the chemistry nor contrails results that ARE shown. This perspective will be discussed further below, where I suggest removing or discussing in a much different way. 2) The contrail modeling has made some simplifications that may impact the results that the authors claim to be important. They need to discuss in more detail how the assumptions might qualify their results and the claimed quantification of the effects that they observe. 3) There are a number of more minor wording or presentation issues that I will identify below, along with some suggestions for how they might be addressed.*

*1. In section 2.2.3, page 6, line 15, "Soot and ice particles can also grow by condensation of water vapor, sulfuric acid, and nitric acid...". Experimental results show that the growth of particle mass in aircraft exhaust plumes is dominated by organic species (and nitric acid is not usually observed in the initial plume regions). Leaving out organic species is leaving out a primary contributor to the mass of these newly formed particles (prior to water deposition in contrail formation) as well as the coatings on soot particles.Thus, the presented microphysical approach is missing the major contribution to mass. (Yet the authors do note that volatile organics are in the exhaust, section 3.1, page 10, line18.)*

**We share the reviewer's concerns regarding the role played by organic species in the initial plume phase. We agree that organic species may alter the freezing behavior of aerosols, as has been shown in previous studies (Cziczo et al. (2004); Kärcher et al., (2005); Murray et al., (2010)). Kärcher et al. (2015) in particular describes the role of organics in great detail and has been a valuable source of knowledge. However, the theory behind particle growth enhanced by organic species or nucleation is still limited and it is not clear how best to incorporate this information. In addition, a large number of soluble organic species involved are typically involved and the processes by which the organic compounds contribute to the aerosol growth are complex and poorly understood. As a consequence, we do not explicitly consider the effects of organic species in these calculations.**

**However, we have now performed additional sensitivity calculations to try and bound the possible roles of organic aerosol. Specifically, we consider two possibilities: that organic aerosol could act like sulfate aerosol (i.e. as a coating material on soot), and that organic aerosol might act as additional ice nuclei. In the**

first case, we simulate additional cases in which the soot coating fraction is assumed to be increased from the baseline value (0%) to between 0 and 25% due to the role of organic matter prior to water deposition. This is presented as a sensitivity analysis in Section 3.1.1. This shows that the inclusion of organic species leads to a faster growth rate through the deposition of gaseous water. However, the particle radius after ~1 second was unchanged in all cases (whether the background was subsaturated or supersaturated). Similarly, the gaseous chemical composition is unaffected by the condensation of organic compounds onto soot particles, even under supersaturated conditions. This suggests that the contrail evolution and chemical consequences of an aircraft exhaust plume are not sensitive to this assumption (page 10, lines 90-92). We also simulate the effect of changing the black carbon emissions index in Section 3.1.3. If organic aerosol acts as a nucleus rather than as a coating, then the result of doubling the black carbon emissions index can be considered as an analog for the effect of including organic aerosols. We now make this comparison explicit (page 13, lines 11-15) but recognize that this is, at best, only a crude approximation of the role of organic aerosol.

We have added a paragraph discussing this limitation in Section 2.2.3, reviewing the literature discussed above (page 5, lines 64-88). We have also added as "future work" the goal of explicitly modeling condensation of organic species to enable a more robust investigation of the potential effect of organic aerosols with respect to contrail formation and plume chemistry (page 22, lines 36-38).

*However, there are no results presented in the paper that show the importance of this microphysical processing. Neither results showing newly nucleated sulfate aerosol nor the coatings on the soot particles (and their composition) are presented in the paper. It is not clear from the material presented how the eventual uptake of water is dependent on the condensed matter due to these species. Is the later water uptake affected by the surface composition? If not, there seems to be no impact of this analysis on the contrail results presented later in the paper. No size distribution results are shown, so it is not clear how the soot distribution and newly nucleated particles make up the input to the downstream mature plume modeling, and how they affect the subsequent analysis.*

The later water uptake is influenced by the soot coating fraction as the bare soot particles are assumed to be hydrophobic. In our current modeling approach, this coating is made up of sulfuric acid and water. We now include an analysis in Section 3.1 which demonstrates the microphysical evolution of the plume, including the partitioning between gaseous and liquid sulfur, including the aerosol distribution, and how this changes over the first 15 minutes (section 3.1).

After formation, the sulfate aerosol distribution and ice particle mean radius are used to initialize the mature plume module. The contrail ice particles are initialized with a log-normal distribution. We have added the following in Section 2.3.3 to clarify how the results from the early plume module propagate to the later simulation stages: "The aerosol distributions in the mature plume phase are initialized based on the output from the early-plume module. The distribution of sulfate aerosols is unchanged while ice particles are distributed assuming a log-normal distribution, using the mean ice particle radius and a geometric standard deviation of 1.6 (Goodman et al., 1998; Jensen et al., 1998a). The use of a log-normal distribution is based on *in situ* measurements (Schröder et al., 2018) and this assumption has been used in previous work to initialize the contrail ice particle size distribution (Jensen et al., 1998b; Picot et al., 2015)." (page 7, lines 59-62).

*It is worth noting that many other modeling studies suggest that the "nucleation mode" is not important for contrail processes when the soot mode is present, due to the larger size of the soot mode. Thus, there is a basis for questioning the importance of this smaller mode. The question of the compositional changes of the soot surface due to condensation seems open, but unaddressed by the present study.*

**Our results support the finding that the nucleation mode plays a negligible role compared to the soot mode, due to its larger size. We include a nucleation mode to enable the model to capture the behavior of contrails for engines with very low soot emissions indices. Under these circumstances, it has been hypothesized that liquid plume and ambient particles could play a significant role in contrail formation. Kärcher et al. (2009) describes three regimes for the origin of ice crystals based on the soot number emission index:**

- **In a soot-rich regime ($EI_N > 10^{15}$ particles / kg fuel), contrail ice is formed when water freezes onto soot (black carbon) particles.**
- **In a soot-poor regime ($EI_N < 10^{13}$ particles / kg fuel), contrail ice is formed when water freezes onto/with liquid particles.**
- **In the intermediate regime, ice can form on both soot particles and liquid particles. The crossover point varies as a function of ambient parameters, but is consistently estimated to be below soot particle number emissions indices ($EI_N$) of $10^{14}$ particles / kg fuel.**

***In situ* observations showed that, for current engines, the soot number emission indices vary between 3.5× $10^{14}$ and $1.7×10^{15}$ particles / kg fuel (Petzold et al., 1999). Those measurements suggest that ice particles currently form by freezing of water around soot cores. However, it is possible that alternative fuels or new combustor technologies may reduce the emissions index to the point that this nucleation mode is significant. We have included a sentence in Section 2.2.3 which clarifies this (page 5, lines 15-20).**

*Unless more information is provided, it seems that this is an incomplete analysis that has limited bearing on the problem at hand, and it does not appear that the model has a means to include the effects of this analysis on the key results presented. I suggest this part of the analysis be removed or completely re-described.*

**We have made the changes above in an effort to make clearer the strengths of our approach, and to clarify where there are limitations or opportunities for further research (Sections 2.2, 3.1, 4).**

*2. In a related issue, the modeling assumes (section 2.2.3, page 6, line 5) that the soot distribution is a mono-modal distribution ("a single representative particle"). While that may make sense to define a more computationally tractable problem, the microphysical modeling discussed in 1. above seems to require a binned size distribution approach (page 7 line 3), so why is it necessary to force the soot to be mono-modal? But if the response to issue 1. above is to remove the volatile particle modeling, then perhaps the mono-modal soot distribution may be justifiable to simplify the computations.*

*However, if the approach is to accept a more limited modeling approach, based on a mono-modal soot distribution as has been done before (as referenced by the authors), then another separate question arises. The results show important differences due to differences in the fate of large particles versus small particles in the later plume processing (section 3.5.1, line 12 et seq.). If the initial soot distribution is mono-modal, the contrail particles will also be mono-modal for those particles that have had the same history (i.e., in the same ring). There needs to be more discussion of how the history of particles might generate a size distribution that differs from the initial mono-modal soot size distribution, if this is, indeed, what generates a polydisperse contrail particle size distribution.*

**The soot particles are represented as a monodisperse distribution, with the assumption that (if a contrail forms) they will act as "seeds" for ice crystals. At this point, the contrail ice particles are initialized with a log-normal distribution whose median radius is computed using the initial box model and with a geometric standard deviation of 1.6 (as now described in Section 2.3.3, page 7, lines 53-62). Even without the application of a log-normal distribution to initialize the model, a polydisperse distribution would arise due to coagulation. This results in different settling velocities, such that the larger particles are exposed to "fresher" air, further changing the overall particle size distribution. Shear of the plume also results in particles in different parts of the plume being exposed to different conditions. The soot distribution is only**

**used to make the problem more computationally tractable initially. The above description is now included in Section 2.3.3 (page 7, line 65 - page 8, line 3).**

**Regarding the ring-shaped mesh, this is only used to reduce the computational cost associated with the chemistry. Both transport and microphysical processes are performed on the fine cartesian grid as described in Section 2.3 (page 6, lines 68-70).**

*3. Presentation issues and typos:*

*a. In the introduction (page 2, line 29 - 30), "aviation is . . . the only direct, significant source . . .", what about rockets? May not be as large, but rockets may still be significant.*

**We agree that the rocket industry is an expanding sector and the number of launches is growing each year. However, there were a total of 90 rocket launches in 2017, while ICAO registered 37 million flights for the same year. In terms of fuel burn, the first stage of the Falcon Heavy (which burns for the first 70 km or so of altitude gain) has a tank capacity of 123 tonnes of kerosene, meaning that the total kerosene burn below 70 km would be ~11 Gg if we use the Falcon Heavy first stage as a generic proxy. In 2015 there was, for comparison, approximately 240 Tg of fuel burn from commercial aviation (from FAA's AEDT). Rocket launches thus correspond to at most 0.005% of the total aviation fuel burn.**

**We replaced the sentence with: "aviation is a unique sector in terms of its environmental challenges as it is the most significant anthropogenic source of pollution at high altitude (8-12 km). In 2015, an estimated 240 Tg of jet fuel were burned for commercial aviation according to the global inventory from the FAA Aviation Environmental Design Tool (AEDT). For comparison, even under a very conservative assumption - that every rocket launch in 2015 was performed with the high-capacity, kerosene-burning "Falcon Heavy" - we estimate that rockets burned at most 11 Gg of fuel below the stratopause in that year." (page 1, line 61 - page 2, line 3).**

*b. Also, in the introduction: this is not meant to be a review article, but it might be worth mentioning that the importance of plume processing has a history that goes back to CIAP (CIAP monograph 3, 1975, DOT-TST-75-53, chapter 2 and references therein) and, {especially since the manuscript is a NASA sponsored study}, to NASA (Atmospheric Effect of Aviation: First Report of the Subsonic Assessment Project, 1996, NASA Ref. Pub. 1385, chapter 4 and references therein).*

**We have added the following sentence "The impact of plume-scale modeling of aircraft wakes has been investigated over the past few decades mostly for its relevance to the environmental impact of aviation." and we cite the CIAP Monograph 3 and NASA report (page 2, lines 29-36).**

*c. Figure 1. As a schematic, this figure seems to address only the 2.3 mature plume modeling part. There is an inset box that discusses the plume box model processing, but there is no schematic representation of the box model in the artwork. And there is no equivalent inset box that describes the discretized rings in the figure.*

*If the box model inset box were removed from the existing figure, I would suggest this figure would sit better in section 2.3, where the mature plume modeling is discussed. It has little schematic value for the box model as drawn so doesn't provide much benefit as placed in section 1. If, on the other hand, the figure was adjusted so that there were schematic aspects and inset boxes for both parts of the model, then perhaps a redrawn version might have reason to remain in section 1.*

**We agree, Figure 1 only represents the evolution of the plume in the mature phase. We have moved Figure 1 accordingly to Section 2.3.**

*d. Section 2.1, page 4, line 13. "The output of this box model..." This sentence is confusing. The antecedent to "this box model" doesn't exist, since the model hasn't been mentioned yet, and is described in the subsequent section.*

*The preceding paragraph describes the physical phenomena to be addressed, but there is no mention of the box model that will be used. One solution would be to briefly mention how a box model (to be described in detail later) will be formulated to capture the elements described.*

*Another solution would be to drop that sentence and pick it up later after the two models are discussed. If this approach is taken, then the material in these two paragraphs (last two paragraphs of section 2.1) would just be discussing the physical phenomena in the two regimes and leave the box and ring models' discussion for the later sections. (If this approach is taken, the title of 2.1 might need to be adjusted.)*

*As written, the sentence is confusing, referencing models that haven't been introduced yet.*

**Thank you for catching this oversight. As suggested, we now provide a brief description of the box model at first mention (page 3, lines 36-39).**

*e. Section 2.2.2, page 5, line 22, Tremmel, (and by Lukachko et al., 1998 JGR 103, and in 2008, J. Eng. Gas Turbines and Power, 130, 2008) found that the conversion of S(IV) to S(VI) occurred primarily in the engine's turbine, and not in the plume to a significant degree. Later processing in the atmosphere happens also, but at time-scales much longer than the initial plume being addressed by this study.*

**The studies from Lukachko et al. (1998) and Tremmel et al. indicate that a significant fraction of S(VI) production occurs between the combustor and the nozzle of the engine. We model this by prescribing a fraction of S(IV) that gets converted to S(VI) at the exit plane of the engine. Brown et al. (1996) also describes that the gas phase oxidation is a much slower process and thus acts as a weaker source of S(VI) than the initial conversion in the aircraft engine. Thus, a large fraction of the sulfur present in the plume 24 hours after emission was already present in the plume at the engine exit plane. To make this clearer to the reader we now state that "Oxidation of S(IV) to gaseous S(VI) is not simulated during this period. This process mostly occurs in the engine's turbines and only a negligible fraction is converted in the young aircraft plume." in Section 2.2.2 (page 4, lines 81-85).**

*f. Section 2.4.1, page 9, line 21. "equipped with GEnx engines". In what sense is this engine represented in the model? In section 3.5.3, page 19, line 8, a soot emission index (EI) is given as 0.06 g/gfuel. (I assume this is a typo, and it is meant to be 0.06 g/kgfuel, or 60 mg/kgfuel).*

**The use of "g/g fuel" was a typo. This has been fixed and changed to mg/kg fuel (page 21, lines 30-31). We answer the broader question of representation of the GEnx engine below.**

*This seems very high for the GEnx engine, especially at cruise at altitude. This (even after correcting for the typo!) is 1.5 times the {high} value used in the 1999 IPCC report of 0.04 g/ kgfuel. And where was the EI soot data obtained? In addition, is the NOx EI chosen to be representative of the GEnx from the ICAO Databank?*

**The representation of the GEnx engines in APCEMM at cruise altitude uses the equations of the Boeing Fuel Flow Method 2 (Dubois et al., 2006) to compute a cruise $NO_x$ emission index from the ones provided by the ICAO Engine Emissions Databank.**

**The value of 60 mg soot/kg fuel is not intended to specifically characterize the GEnx, but is rather part of a sensitivity analysis for that section only. In all other areas, we use the SN-$C_{BC}$ method described in Stettler et al. (2013), equation (5) to estimate the mass soot emission index. As a consequence, the soot emissions**

index used (unless otherwise specified) is 10 - 14 mg/kg fuel (depending on local conditions based on the SN-$C_{BC}$ method). We now describe this approach on page 9, lines 17-38, and specify the calculated soot and $NO_x$ EIs where relevant (e.g. page 14, lines 11-12).

In Sections 3.5.2 and 3.5.3, we explicitly varied the soot emission index from 10 mg/kg fuel to 60 mg/kg fuel to estimate the sensitivity of our results to this parameter. Stettler et al (2013) estimate that the fleet-wide average soot emissions index is 28 mg/kg fuel. Previous estimates have found that different engines in the fleet can have emissions indices which vary by an order of magnitude (e.g. 11 - 100 mg/kg fuel as estimated by Petzold et al., 1999). This clarification has been included in page 19, lines 50-60.

By varying the soot emission index, we do not intend to model variation in the GEnx engine but rather to focus on the impact of different soot emission indices on contrail and chemical properties. We thus keep everything unchanged from the GEnx emission characteristics except from the soot emissions. To clarify, we added the following to the manuscript (changes underlined):
"We next model how changes in soot emissions affect the properties of the contrail. We simulate an aircraft plume in which the soot mass emission indices are varied between 10 and 60 mg/kg fuel, compared to 10 to 14 mg/kg estimated using the SN-$C_{BC}$ method for the GEnx engine. All other aircraft and engine emissions parameters are fixed for this sensitivity analysis." (page 19, line 50-60).

*g. Section 3.5.1, page 16, line 1. Supersaturations of 102% to 108% are quoted, but is this respect to water or ice? (And is 108% observed in the natural atmosphere?)*

The saturations quoted in the paper are expressed with respect to the ice saturation pressure. Gierens et al. (1999) estimate that, in supersaturated regions of the upper troposphere, the mean supersaturation is 15% (corresponding to a relative humidity with respect to ice of 115%). This clarification is now given in the main text on page 18, line 22-25.

**References:**

Brown, R. C., et al. "Aircraft exhaust sulfur emissions." *Geophysical research letters* 23.24 (1996): 3603-3606.

DuBois, Doug, and Gerald C. Paynter. "" Fuel Flow Method2" for Estimating Aircraft Emissions." *SAE Transactions* (2006): 1-14.

Cziczo, D. J., et al. "Observations of organic species and atmospheric ice formation." *Geophysical research letters* 31.12 (2004).

Gierens, Klaus, et al. "A distribution law for relative humidity in the upper troposphere and lower stratosphere derived from three years of MOZAIC measurements." *Annales Geophysicae*. Vol. 17. No. 9. Springer-Verlag, 1999.

Goodman, J., et al. "Shape and size of contrails ice particles." *Geophysical research letters* 25.9 (1998): 1327-1330.

Jensen, E. J., et al. "Ice crystal nucleation and growth in contrails forming at low ambient temperatures." *Geophysical research letters* 25.9 (1998a): 1371-1374.

Jensen, Eric J., et al. "Spreading and growth of contrails in a sheared environment." *Journal of Geophysical Research: Atmospheres* 103.D24 (1998b): 31557-31567.

Kärcher, B., et al. "The microphysical pathway to contrail formation." *Journal of Geophysical Research: Atmospheres* 120.15 (2015): 7893-7927.

Kärcher, B., and Thomas Koop. "The role of organic aerosols in homogeneous ice formation." *Atmospheric Chemistry and Physics* 5.3 (2005).

Kärcher, B., and F. Yu. "Role of aircraft soot emissions in contrail formation." *Geophysical Research Letters* 36.1 (2009).

Lukachko, S. P., et al. "Production of sulfate aerosol precursors in the turbine and exhaust nozzle of an aircraft engine." *Journal of Geophysical Research: Atmospheres* 103.D13 (1998): 16159-16174.

Murray, Benjamin J., et al. "Heterogeneous nucleation of ice particles on glassy aerosols under cirrus conditions." *Nature Geoscience* 3.4 (2010): 233-237.

Petzold, Andreas, et al. "In situ observations and model calculations of black carbon emission by aircraft at cruise altitude." *Journal of Geophysical Research: Atmospheres* 104.D18 (1999): 22171-22181.

Picot, J., et al. "Large-eddy simulation of contrail evolution in the vortex phase and its interaction with atmospheric turbulence." *Atmospheric Chemistry and Physics* 15.13 (2015): 7369.

Schröder, Franz, et al. "On the transition of contrails into cirrus clouds." *Journal of the atmospheric sciences* 75.2 (2018).

Stettler, Marc EJ, et al. "Global civil aviation black carbon emissions." *Environmental science & technology* 47.18 (2013): 10397-10404.

Tremmel, Hans Georg, and Ulrich Schumann. "Model simulations of fuel sulfur conversion efficiencies in an aircraft engine: Dependence on reaction rate constants and initial species mixing ratios." *Aerospace Science and Technology* 3 (1999): 417-430.

*Referee: 2*

*The study introduces a new model called APCEMM, which is designed for simulating chemical processes in aircraft plumes and also considers the effect of contrails on plume chemistry. My impression is that authors do not have a strong background in atmospheric physics. This becomes apparent in quite a few passages of the manuscript. I recommend that the author team strengthens their expertise in atmospheric physics before revising the manuscript and redesigning the APCEMM. The manuscript could become publishable only after major revisions.*

**We thank the reviewer for their interest in our manuscript and we are grateful for their detailed comments and in-depth analysis. Please find our responses to each individual comment below.**

*General comments*

*As already stated in the summary above, I doubt that the implemention of the various atmospheric processes is done correctly. Moreover, sometimes processes or phenomena are included, that are irrelevant and only pretend to increase the level of detail in the model. Comments on the physical soundness of your apprach are listed in the section "Specific comments".*

**We have given specific responses below. However, we have also made a concerted effort to make both the details of implementation and the rationale (with regards to level of detail) of our modeling choices clearer throughout the entire methods section.**

*Here, only several general comments on terminology and language are made.*

> *• Even though often written and read, it is wrong: Temperature is not cold or warm. It is low or high and tells us if something is hot or warm. Please check the whole manuscript.*

> **Thank you for this correction. We have checked and corrected the entire manuscript accordingly.**

> *• E.g. formula (2), (4)*
> *I find it awkward to provide units for each quantity. This somehow pretends that the formulae are only valid in conjunction with exactly those units. This is certainly not the case. I understand that supplying units helps the reader to make a first check of the correctness of the formulae. But the way it is presented, it is misleading.*

> **We now clarify at relevant locations (e.g. page 4, line 39-42) that units are provided for the purpose of demonstration only, and are not fundamental to the formulae.**

> *• Aerosol is a gas with suspended particles. If you refer to the particles only, better use the term aerosol particle. Sometimes you use the term aerosol even for ice crystals (in particular, last paragraph of section 2.2.3). I would make a clear distinction between aerosol particles and ice crystals.*
> *http://glossary.ametsoc.org/wiki/Aerosol*

> **We now make an explicit statement regarding the distinction between aerosols and aerosol particles on page 6, line 47.**

> *• Please use the terms deposition, sublimation, condensation and evaporation consistently.*

**We have modified the manuscript accordingly.**

*• p.4. l.31: those THAT were emitted*
*See https://www.wisegeek.com/what-is-the-difference-between-that-and-which.htm*
*Please check the whole manuscript.*

**We have modified the manuscript accordingly.**

*• Concerning statements in the abstract like "evaporate ~9% faster and are 14% optically thinner"*
*Given the accuracy of the (still) simplified treatment, I would prefer to leave out such precise numbers (in the abstract). How much are they worth? If you use another definition of optical thickness or define the time of evaporation slightly differently, I am sure you can get anything between 5% and 20%.*

**We agree that the original wording was too precise and may have overstated the achievable accuracy. As such, we have reworded the relevant phrases in the abstract to highlight the qualitative rather than quantitative outcome, and provide the full range of simulated outcomes as an example rather than as a definitive consequence of the use of biofuels. The section in question now reads (page 1, lines 34-37):**

**"Our results suggest that a 50% reduction in black carbon emissions, as may be possible through blending with certain biofuels, may lead to thinner, shorter-lived contrails. For the cases which were modeled, these contrails evaporate ~5 to 15% sooner and are 10 to 22% optically thinner"**

*Table 7 and 8*
*Given the uncertainties, it is not meaningful to provide numbers like −5.35% with two decimal places. Please round them to a reasonable precision. Similarly the value 1.2581 in Table 4 it "too" precise. Please go through the whole manuscript.*

**Given the achievable accuracy of the method, all numbers in the paper have been rounded to two significant figures.**

*Specific comments*

*p.2, l.39:*
*what are "local aerosol clouds"?*

**The sentence has been replaced with:**
**"This approach does not explicitly capture the high initial species concentrations within the plume, including the effects of non-linear chemistry in the early stages or the formation (and chemical effects) of aerosols and ice crystals (i.e. contrails) in the exhaust plumes. " (page 2, lines 20-21)**

*p.4, l.5:*
*The ambient temperature at cruise altitude is not 280K. Climate change is not that fast:-)*

**This typo has now been corrected to the less alarming temperature of 220K.**

*p.4, l.14:*
*Your statement implies that the coherent vortex flow field is just turbulence which is not the case. Please better describe how the vortices break up. Paoli and Shariff (2016) is a good source of information for contrail-specific processes and phenomena.*

**We agree with the reviewer and we realize that the vortex flow field is not purely turbulence. We have used Paoli and Shariff (2016) as a reference to extend our discussion of the flow field, our simplified treatment of it in this model, and possible future extensions of the work.**

**We realize that "viscous dissipation of turbulent energy causes the vortices to break apart" is imprecise and do not mean to claim that the coherent vortex flow field is purely turbulence. As such, we have also added a discussion of relevant vortex dynamics in the first 10 minutes in Section 2.1 (page 3, lines 9-17).**

**In our model, the vortex flow field is not explicitly represented. We assume that the vortex breakup occurs over 5 to 10 minutes. During that period of time, we use the simplifying assumption that the chemical species and ice particles are well-mixed. We acknowledge that this is a limitation of this study, however, and have added a statement to this effect in Section 2.2.1 (page 4, lines 56-61).**
**We also realize that enhancing diffusion in the beginning of the mature plume phase is a simplistic assumption and does not allow us to model the aircraft-induced wake dynamics accurately. We added the following sentence is Section 2.3.1: "Although computationally efficient, our current representation of the aircraft-induced turbulence is simplistic and does not allow us to model the spatial heterogeneity that would arise after the dissipation of the vortex pair."**

*p.4., l.16:*
*Schumann, 2012 is a long paper. Which formula do you use? Do assume that the vertical motion is constant over time during the vortex phase? Note that in a stably stratified atmosphere, large parts of the vertically displaced plume rise back to the original emission altitude after vortex breakup due to buoyancy. As some portion of the ice crystals (or some other tracer) remains at lower levels, the vortex sinking causes a strong and fast vertical plume expansion (compared to timescales of natural processes). It seems that this effect is not considered in your model.*

**We compute the sinking depth according to Equation (13) from Schumann (2012). We do assume in the current model that the sinking velocity is constant over time during the vortex phase.**

**We are also aware that the plume can rise back to its original emissions due to buoyancy. We model this phenomenon as an upward motion applied over the first hour of the plume. We indeed notice that the vortex sinking followed by the updraft leads to a considerable stretching of the contrail, much more than is expected due to vertical diffusion alone. We now note this effect on page 3, lines 49-58. We now also state explicitly that the treatment of vertical motions used in this study is simplified, and an area for potential future improvement (page 7, lines 36-39).**

*p.7, l.45:*
*Who is the user in this case and is supposed to choose a value for Dh?*

**The text has been clarified and now specifies that APCEMM requires the horizontal diffusion coefficient as an input (page 7, line 8). Table S1 in the Supplementary Information lists all the inputs required by APCEMM**

*p.8, first paragraph:*
*The way you include the effect of radiation is not correct. Contrail parts with the highest IWC are usually heated the most. This heating causes an uplift of those contrail parts during which the air cools adiabatically (again proportional to Γd). Assuming the atmosphere is stably stratified, the local uplift is sustained as long the ambient temperature is below the temperature of the contrail patch. So for typical stratification values, the initial heating actually translates in a cooling of the contrail! As the heating in the contrailfall streak is usually not that strong, radiation leads to a contrail vertical stretching.*

*See introductory textbooks on lifting condensation level for the general physics (unlike to warm clouds however, the latent heating effects are not that important in ice clouds and the moist-adiabatic lapse rate Γm is roughly the same as Γd).*

**We are aware of the fact that contrails experience a non-uniform uplift due to heterogeneity in the local ice water content. The current configuration partly reflects the physics described above. Contrail radiative imbalance is modeled as a transient updraft, which results in the contrail cooling (due to the presence of a vertical temperature gradient). We have clarified this in Section 2.3.1 and we have added a sentence in Section 4, discussing the limitations of the current approach (page 22, lines 29-38).**

*p.8, l.11:*
*Could you describe in a few words what KPP is.*

**KPP stands for the Kinetic Pre-Processor. KPP is a software tool which from a set of chemical reactions and rate coefficients generates code to integrate the differential equations and compute the time evolution of chemical species with a suitable numerical integration scheme. This explanation has been added on page 7, lines 48-51.**

*p.8:*
*The inclusion of ice aggregation seems very sophisticated (iterative determination of coalescene efficiency) compared to the treatment of other processes in the model. But more aggravating is the fact that the cited Beard & Ochs paper deals with precipitation drops and not ice crystals. Please refer to literature refering to aggregation, not coalescene of liquid cloud droplets. Section 4 of Sölch and Kärcher (2010) could be a good starting point to dive into the physics behind the aggregation efficiency.*

**We agree with the reviewer. Additional literature review confirmed that Beard & Ochs (1995) refers indeed to liquid cloud droplets. We now adopt the methodology described in Sölch and Kärcher (2010) by setting a constant aggregation efficiency.**

**The appropriate paragraph has been removed from the manuscript and replaced by:**
**"Following the approach from Sölch and Kärcher (2010), we assume a constant aggregation efficiency for ice particles." (page 8, lines 21-23). All analyses in the manuscript have been repeated using the updated methodology. Changes in outcomes due to this modification are included in the summary provided at the start of the reviewer responses, but overall this did not affect the conclusions of the paper. The inclusion of a constant aggregation efficiency for ice particles (equal to 1 as in Sölch and Kärcher (2010)) lead to minor changes as our previous aggregation efficiencies converged towards unity for particles smaller than 50 µm. As an estimate, in a typical experiment (similar to the setup described in Unterstrasser et al. (2010a)),  the integrated number of particles larger than 30 µm changed by less than 0.3% after 6 hours.**

*p.9., l.21:*
*What specifications of the chosen aircraft and engine type appear in the model, and which ones matter in the end? Are your results only valid for this specific aircraft/engine combination or can your findings be generalised?*

**A number of aircraft parameters come into the model. These vary from emission characteristics ($NO_x$, $H_2O$ and soot emission indices, fuel flow), aircraft properties (aircraft mass, wingspan) and flight conditions (flight level, background meteorological and chemical conditions).**

**The main factors influencing our metrics are the $NO_x$ emission index, and the flight-level conditions, as demonstrated in Sections 3.2, 3.3 and 3.4. Microphysical characteristics are more sensitive to the soot emission index, aircraft properties and meteorological conditions as well.**

**To make these dependencies clearer, we now include a table in the SI which lists all of the input parameters used by the model; which are specific to the aircraft type; which are engine parameters; and so on. We also list the sources used for the examples in the paper (Table S1).**

**Only results from a B747-800 equipped with GEnx engines are presented, but all relevant parameters can be modified through the input structure provided to APCEMM. Comparison of results using other aircraft/engine combinations is a future research opportunity and this is now mentioned on page 9, lines 43-45.**

*p.9:*
*Why is it important to evaluate the error X after 24 hours, hence at the same time of the day as the initialisation was done? Wouldn't it be better to evaluate the error at a time where the spatial dimensions of the APCEMM-modelled plume and the BOX-modelled plume are similar? Refering to Figure 2, you state in the text that 1.2 kg or 0.2 kg ozone is produced, giving a fatcor 6 difference between the two modelling approaches. A variation of the evaluation time would dramatically change this factor. Using an earlier point in time (e.g. 2 hours earlier at time 6:00) the factor would be much higher. Could you make clearer the strategy behind your evaluation effort.*

**We use the 24 hour timescale as a convenient approximation, and recognize that it could be further refined. We chose 24 hours for two reasons:**
1. **After 24 hours, the sunlight conditions are identical to the ones after emissions. This is necessary to compare the concentrations of photochemically active species, like NO or $O_3$. Comparing $O_3$ under different photochemical states would not be a fair comparison for this study.**
2. **The plume model allows us to model fine-scale chemical phenomena, like $HO_x$ depletion, early on in the plume. This process cannot be captured with a single box model. After 24 hours, we assume that the plume is sufficiently diluted to conclude that the fine-scale representation is no longer needed, and thus, that the box model and the plume model behave similarly. The plume's dimensions could still be much smaller than that of the box model. Indeed, in most cases we calculate that the "characteristic width" of the plume, based on the extent in which 95% of an emitted tracer is contained, is still only 50 km, compared to the box width (typical of modern global atmospheric simulations) of 100 km.**

**As such, we consider 24 hours to be a convenient stopping point, rather than being necessarily the optimal point at which the plume could be considered "ready" for hand-off to a lower-resolution model. We clarified this in the manuscript by adding the underlined text in Section 2.4.2 (page 9, lines 84-92): "Evaluation after 24 hours ensures that the domain is in the same photochemical state as at initialization. This ensures that we make a fair comparison for photochemically-active species. However, the plume may still be sufficiently concentrated that adding it to a grid cell in a larger simulation may still result in misrepresentation of plume chemistry. Additional work will be needed to quantify the magnitude of this error if plume processing is embedded into a global-scale model. "**

*Remarks on section 3.5.1:*

> *• Several choices of the background conditions are not reasonable at all. Section 2.2 of Kärcher et al. (2009) may help to choose more realistic cases.*

> **Based on review of Kärcher et al. (2009), we have updated our contrail simulations to use a more realistic background condition. Specifically, we use the approach described in Unterstrasser et al. (2010a), in which a background 50% relative humidity and a single supersaturated band (1 km in thickness) are prescribed. The description in Section 3.5 has been updated to reflect this (page 18,**

**lines 18-19), and all results have been updated accordingly. This change also enabled us to perform direct comparison of our results to Unterstrasser's (see later comment).**

*• A 10 cm/s cooling over 24 hours translates into a lifting by 8 km and an adiabatic cooling by 80K. This is not realistic. Compared to this, the 0.1 K diurnal temperature variation can be safely neglected.*

**The simulated updraft is used to model buoyancy and the effect of radiative imbalance, and is only applied during the first hour. As such it results in a lifting of only ~150 m, and a cooling of ~1 K. This is described in Section 2.3.1. (page 7, lines 34-35).**

*• Persistent contrail formation is likely to occur in a RHi-range of 100%−−140% (above the upper limit, natural cirrus formation could not be neglected)*

**We agree. We have restricted our analysis to meteorological conditions with a relative humidity below 140%.**

*• Does the depth of supersaturated layer remain constant over time? Given the prescribed uplift, the initially subsaturated layer above/below the supersaturatedlayer would eventually become supersaturated as well and the supersaturated layer would grow in size. If you used a time-constant 200m thick layer and included the radiation effect correctly, the contrail would move out of supersaturated layer into the drier air above. This leads to entrainment of dry air into the contrail which would then start to vanish.*

**We have assumed in our study that the depth of the supersaturated layer remains constant over time. However, the contrail is able to move relative to the layer, and does so during the first hour ("updraft"). In our current cases, we use a thicker (1 km) layer, consistent with Unterstrasser et al (2010a). Depending on the updraft velocity, the contrail can therefore rise out of the supersaturated band (page 19, lines 27-32).**

*• Given the quite thin supersaturated layer, the simulated contrail lifetimes of >10h appear to be too large (in particular for vUP= 0).*

**Following the implementation of more realistic background conditions, the predicted lifetimes have reduced. We now find lifetimes between 6 and 10 hours, consistent with Lewellen et al (2014a).**

*• You first make a link between in-situ loss and aggregation and few lines later you say in-situ loss is due to Ostwald ripening.*

**This is a mistake in the original manuscript. *In situ* losses are not linked to aggregation of particles. In situ losses correspond to the sublimation of small crystals at a relative humidity close to 100% in favor of larger crystals, as described by Lewellen et al. (2014a). The corresponding clause ("represented in the coagulation kernel applied during the diffusion regime") has been deleted and this has been clarified (page 19, lines 10-16).**

*• line 21: Yes, it is usually warmer further down, but this is irrelevant here. Or do really want to say the ice crystal melt and become water droplets? What matters is that it is dry and the ice crystals sublimate and are lost eventually.*

**We agree that this was worded incorrectly. What was meant is that the ice crystals enter a drier region and sublimate. The manuscript has been updated accordingly (page 19, line 41).**

*• If your contrail model produces reasonable results, could be checked by a comparison with the higher resolution model used in Unterstrasser & Gierens, 2010 a,b. This comparison should be feasible to achieve with small extra effort as you anyway use mainly their definitions of contrail properties. It would only require to specify the same background conditions. This could show if your modeled contrail lifetimes and response to variations of RHi or EIsoot are reasonable.*

**We thank the review for this suggestion. We performed a number of validation tests and now compare our results directly to those presented by Unterstrasser & Gierens (2010a). The inter-comparison is shown in Figure R1 of this document, and a discussion of this comparison (along with the figure itself, in two parts) has been added to the manuscript as Appendix D. The upper 12 panels show results from Unterstrasser & Gierens (2010a), while the lower 12 panels show results from APCEMM.**

**We show total ice particle number concentration, ice water content, extinction and $RH_i$ at three time instances for the same meteorological conditions and emissions as Unterstrasser & Gierens (2010a). Based on visual inspection, we estimate that the integrated ice particle number and ice mass agree to within ~10%. The spatial distributions of ice crystal number and mass densities at t = 2,000 s are in agreement, with a maximum horizontal discrepancy less than 500 m. At later times, APCEMM fails to capture the large horizontal extent observed in the results from Unterstrasser & Gierens (2010a), caused by the fallstreak. This discrepancy is clearly visible on the ice water content row where APCEMM predicts a contrail spreading of 6 and 12 km at t = 8,000 s and t = 17,000 s respectively. By comparison, the LES simulation predicts extents of 10 and 22 km. However, vertical extents appear to be well represented.**

**Overall, APCEMM seems to predict a longer contrail lifetime with greater contrail ice mass, particle number and extinction. This could be explained by the inherent turbulent motions around supersaturation as described in Gierens et al. (2009) and Unterstrasser et al. (2010a). Such local fluctuations in an overall supersaturated region would induce changes in the total ice mass and cause the local sublimation of ice crystals which would tend to reduce the contrail lifetime. Efforts to parameterize this effect, if it is indeed the cause, are now suggested as future work in Sections 3.6 and 4.**

**In spite of this, we believe that the bulk features of the simulated contrails are in good agreement, with a maximum error on the integrated ice mass of less than 10%. The largest discrepancy lies in the horizontal spreading of the profiles but we think that APCEMM produces useful results given the simplicity of the approach taken to model contrail dynamics.**

*Table 8:*
*Wouldn't it be interesting and more insightful to compare all four simulations to the non-contrail simulation?*

**We now present the results of the contrail-induced chemical impact against the baseline case where no contrail is present (assuming the same background meteorological conditions). This comparison is shown and discussed in Section 3.6.3, which has been reframed accordingly (page 21, line 12).**

[Figure]

***Figure R1.*** **Comparison of results from Unterstrasser & Gierens (2010a) (top) to a simulation in APCEMM (bottom) which attempted to reproduce the same input conditions. Data for the upper plot was kindly provided by the authors of the original paper.**

*Technical corrections*

*p.3, l.14: the effects OF these*

**The changes have been made.**

*p.7, l.21/22: No complete sentence.*

**The sentence has been removed.**

*p.7, l.42: repetition of "a measure of local .."*

**The second instance has been removed.**

*p.11, l.24: please reformulate the sentence.*

**We have reformulated the paragraph:**
**"The instant dilution approach overestimates ozone production for any emission time, with emission conversion factors in the box model which are up to three times their respective values in the plume model. These discrepancies are greatest in summertime due to the larger ozone production term. The size of the ozone perturbation is sensitive to background concentrations of $NO_x$ in both models.**

**During summertime, increasing the background concentration of $NO_x$ from 100 to 200 pptv reduces the net (positive) ozone perturbation by 30-45% in both models. During wintertime, the same change in background $NO_x$ has a negligible effect in the plume model, as shown in Figure 8. However, the instant dilution approach is still sensitive to this change. It produces a larger (more negative) ozone perturbation when the background $NO_x$ is increased during wintertime. This pattern is explained by a less efficient conversion of $NO_x$ to reservoir species at night. The transition between net positive and net negative ozone also changes as a function of the background $NO_x$. At 50 pptv of background $NO_x$, the plume model simulates net ozone production for 10 months, compared to 8 months in the instant dilution model. At 200 pptv, net production is simulated for 6 and 5 months by the two models respectively. This inconsistency in the magnitude and sign of the error between the two models means that the true impact of aviation emissions will be inconsistently modeled by an instant dilution approach." (pages 15, lines 6-31)**

**References:**

**Beard, Kenneth V., and Harry T. Ochs III. "Collisions between small precipitation drops. Part II: Formulas for coalescence, temporary coalescence, and satellites."** *Journal of the atmospheric sciences* **52.22 (1995): 3977-3996.**

**Gierens, Klaus, and Sebastian Bretl. "Analytical treatment of ice sublimation and test of sublimation parameterisations in two-moment ice microphysics models."** *Atmospheric Chemistry and Physics* **9 (2009): 7481-7490.**

**Kärcher, B., et al. "Factors controlling contrail cirrus optical depth."** *Atmos. Chem. Phys* **9.16 (2009): 6229-6254.**

**Lewellen, D. C., O. Meza, and W. W. Huebsch. "Persistent contrails and contrail cirrus. Part I: Large-eddy simulations from inception to demise."** *Journal of the Atmospheric Sciences* **71.12 (2014a): 4399-4419.**

Lewellen, D. C. "Persistent contrails and contrail cirrus. Part II: Full lifetime behavior." *Journal of the Atmospheric Sciences* 71.12 (2014b): 4420-4438.

Paoli, Roberto, and Karim Shariff. "Contrail modeling and simulation." *Annual Review of Fluid Mechanics* 48 (2016).

Schumann, Ulrich. "A contrail cirrus prediction model." *Geoscientific Model Development* 5 (2012): 543-580.

Sölch, Ingo, and Bernd Kärcher. "A large-eddy model for cirrus clouds with explicit aerosol and ice microphysics and Lagrangian ice particle tracking." *Quarterly Journal of the Royal Meteorological Society* 136.653 (2010): 2074-2093.

Unterstrasser, Simon, and Klaus Gierens. "Numerical simulations of contrail-to-cirrus transition – Part 1: An extensive parametric study." *Atmospheric Chemistry and Physics* 10.4 (2010a): 2017-2036.

*Referee: 3*

*This paper describes a detailed chemical and microphysical model that calculates the composition of aircraft plumes and its interaction with the atmosphere. This is certainly one of the most comprehensive studies that accounts for many chemical reactions and interactions between chemistry and microphysics of particles. The results obtained confirm previous studies, like for example the overestimation of ozone production due to NOx emissions when instantaneous dilution is adopted in CTMs and the role of heterogeneous processes that convert part of the emitted NOx into HNO3. All those results are interesting but in my opinion there are several issues that should be further discussed. Follows my major points.*

**We thank the reviewer for their interest in our manuscript and we are grateful for their detailed comments and in-depth analysis. Please find our responses to each individual comment below.**

*1/ In the introduction it is claimed that almost all the CTM use the instantaneous dilution (ID) approximation to account for aircraft emissions. The authors seem to ignore the many attempts of modelers to introduce plume processing in their large-scale model. The authors should refer to the review paper by Paoli et al. (2011) that gives a comprehensive review of the different approaches that have been followed to account for plume effects: Effective Emission Indexes, Emission Conversion Factors or Emission Reaction Rates. In the same paper are listed the CTM that use those parameterizations and the limitations of each approach.*

**Paoli et al. (2011) gives an in-depth overview of the methods used to account for a plume-scale processing of emissions and should have been included as a reference in the current paper. We have added a discussion of this previous work on page 2, lines 29-36.**

*2/ In contrast to the detailed chemistry and microphysics introduced in their model the authors have chosen a very simplified representation of the contrail and plume dynamics based on a simple diffusion model. This is a very crude approach that for instance ignores the details of contrail dynamics with the role of the crow instability, the formation of secondary vortices that maintain a significant fraction of the emissions at flight level with often persisting ice particles, and the complex nature of atmospheric turbulence and its interaction with radiation (e.g. Paoli et al., 2017). For instance wind shear and diffusion are considered as separate processes, although depending on the scale considered wind shear and diffusions are both the results of turbulence in a stratified atmosphere. Thus, it is very difficult to evaluate how the approximations made can influence the results of the model. Is it a balanced approach to introduce a detailed chemical and microphysics schemes with such a simplified dynamical scheme?*

**We agree that our approach to model the mixing between the plume and the background air is simplistic and does not account for the Crow instability and other plume-scale dynamical phenomena. We prioritized chemistry and microphysics on the expectation of being able to make significant new insights into the chemical behavior of the plume. However we agree that a more complex representation of plume dynamics could result in changes in the results, in particular with regards to contrail formation and persistence. This is reflected in the comparison now performed against the large eddy simulation results of Unterstrasser and Geirens (2010), in Appendix D.**

**We have added several paragraphs discussing the vortex dynamics in the early plume (Section 2.1) and the simplified representation used in APCEMM. As discussed in the response to Reviewer #2, we now mention in Section 2.2.1 that the current parameterization for plume mixing is not able to capture the spatial heterogeneity that could arise from vortex dynamics. We also added discussion in Section 2.3.1 which clarifies that the enhanced diffusion in the mature plume representation is a simplification, and that a**

**higher-fidelity approach would be needed to capture the effects of vortex structures on the optical and chemical properties of the plume (page 7, lines 17-19).**

*3/ Little is said on the validation of the model. Do we have measurements to confront to the model outputs? Can we constraint ozone formation rates and the conversion fraction of the emitted NOx to nitrogen reservoirs?*

**We agree that validating the model is a crucial step when developing software. We have compared as best as we can with previous papers dealing with the same topic and the results presented in this paper are in good agreement with previous publications (Petry et al., 1998; Meijer et al, 2000; Kraabøl et al, 2000). Although observational data is sparse, we also note in the discussion that direct comparison to *in-situ* measurements of both plume chemistry and ice would be a valuable future step (page 24, lines 52-54).**

*4/ In the conclusion it is concluded that plume effects are important and should be included in CTM. This is not new (see all the articles referenced by Paoli et al. 2011) and leaves open the difficulty to do that in a consistent manner with the chemistry and microphysics in place in those CTMs or GCMs. It is often because this consistency is difficult to preserve that the CTM and MCG modelers keep the ID approach despite its limitation.*

**We agree that maintaining the current capabilities of the chemical and microphysical processes of the CTMs and GCMs while adding a new capability - with an online treatment of aircraft emissions - could prove to be tricky. We now state explicitly on page 22, lines 46-48 that finding and maintaining an efficient implementation in a global simulation is a non-trivial challenge which we intend to confront in a future study.**

**References :**

**Paoli, R., O. Thouron, D. Cariolle, M. Garcia and J. Escobar. Three-dimensional large-eddy simulations of the early phase of contrail-to-cirrus transition: effects of atmo-spheric turbulence and radiative transfert. Meteorologische Zeitschrift, Vol. 26, 6, 597-620, 2017.**

**Paoli, R., D. Cariolle, R. Sausen. Review of effective emissions modeling and computation. Geosci. Mod. Dev., 4, 643-667, 2011.**

**Unterstrasser, Simon, and Klaus Gierens. "Numerical simulations of contrail-to-cirrus transition – Part 1: An extensive parametric study." *Atmospheric Chemistry and Physics* 10.4 (2010a): 2017-2036.**

Thank you again for arranging this review. We look forward to your response.

Sincerely,

Sebastian Eastham

---

## Author Response (AR2)

**Sebastian D. Eastham**
**Research Scientist,**
**Laboratory for Aviation and the Environment**
**Department of Aeronautics and Astronautics**

[Figure]

**Massachusetts Institute of Technology**
77 Massachusetts Avenue, Building 33-322A
Cambridge, Massachusetts 02139–4307, USA
http://lae.mit.edu

Atmospheric Chemistry and Physics Editorial Office

30th March 2020

Dear Editor,

**Re: Revisions for "The role of plume-scale processes in long-term impacts of aircraft emissions" after submission to *Atmospheric Chemistry and Physics***

Thank you for considering our submission and for arranging this additional round of detailed and careful reviews. We appreciate the time and effort taken to provide feedback on this manuscript. We are grateful for the detailed comments received. Where possible, we have modified the manuscript to better satisfy the reviewers' concerns.

Most importantly, in light of the comments of reviewer #3, we have attempted to make as clear as possible to the reader that the APCEMM model is of intermediate complexity which is designed to resolve plume chemistry and microphysical processes. We wanted to make sure that the reader knows that the model is designed to provide more detail and accuracy than a Gaussian plume approach but is not designed to achieve the same level of dynamical fidelity in contrail modeling as (say) a Large Eddy Simulation (LES) model. It is also for this reason that we sought the assistance of Dr. Unterstraßer, an LES modeller who specializes in contrails, during the preparation and review of this manuscript. It is for this reason that Dr. Unterstraßer is credited in the manuscript's acknowledgements.

Please find below our responses (in **bold**) to each of the reviewers' comments (in *italics*). Since the second reviewer's comments were embedded as PDF annotations on our previous response, we have extracted those comments for our reply, adding brief context where helpful. In addition to the "clean" version of the manuscript which we have submitted through the usual channels, we have attached to this document a copy of the manuscript in which all changes are highlighted. Line numbers below correspond to the "markup" version.

*Referee: 1*

Submitted on 6 Mar 2020

*No comments made.*

*Referee: 2*

Submitted on 13 Mar 2020

*Page 10, line 45: "As the relative humidity with respect to ice falls below saturation, the particles start to melt." I would suggest that the microphysics is different: that subsaturation would induce sublimation, which would keep the particles frozen, and not induce melting. The subsequent statement about maintaining saturation would then follow from sublimation shifting ice mass from the particle phase to the gas phase. If the term "melt" is changed to "sublimate", then I think that the description would make more sense physically.*

**We agree with this interpretation. The word "melt" has been replaced by "sublimate" as suggested (page 10, line 79).**

Submitted on 18 Mar 2020

[Regarding appropriate use of "that" and "which"]
*Did you really check the whole manuscript? In the rather short section 3.6 I find two mistakes.*

**We have gone carefully through the manuscript a second time and modified remaining incorrect uses of "that" and "which" accordingly.**

[Regarding a statement given in the response to the reviewer] *No, this statement reveals that the authors still have not understood the physics behind thermal stratification. The contrail cools due to adiabatic expansion (you need a vertical pressure gradient). The temperature gradient of the surrounding air is irrelevant for this. It is not the reviewer's task to explain content of introductory meteorology textbooks. Lack of basic meteorlogical knowledge does not boost my confidence in the model. How can I be sure that everything is correctly implemented if you fail to give a consistent description of simple meteorology?*
*You invested a lot of effort in devising the model, so I strongly encourage you to have more discussions with atmospheric physicists to make more reasonable (design) choices.*

**We realize that the response previously given to the reviewer was misleading. The plume does indeed cool through adiabatic expansion as it rises. Although the effect on (e.g.) contrail ice mass is still a function of the ambient temperature gradient (as discussed by e.g. Lewellen 2014), our original response was inaccurate and for this we apologize.**

*Moreover, I think the wording is misleading. Updraft/uplift usually refers to a rising air layer (together with the aircraft plume and/or contrail). If you would like to say that the contrail moves relative to the surrounding air please write at least "contrail updraft".*

**We agree that the use of the words "updraft" and "uplift" can be misleading. We now refer to "contrail updraft" in the manuscript (page 18, line 54; page 19, line 21; page 22, line 60).**

*What is the benefit of adding a radiatively driven contrail uplift in your setup? What are the reason for choosing 10cm/s over 1h? And the cooling is supposed to be same, indepedently of the contrail properties? To be honest, this pretends to include radiation in the model but I do not see a benefit of doing it so crudely. Then better leave it out.*
*Your second motivation for including a contrail updraft is the buoyancy of the displaced plume. The Brunt-Vaisala frequency tells you how fast the displaced air parcel reaches its original level (it is around 10min), not an hour. Anyway your contrail spatial init is simplified, so why bother about bouyant sloshing (Lewellen uses this term for the process you talk about)? Considering the simplifications in other model parts, it would be more adequate to ignore those buoyancy-related short term effects and simply prescribe a contrail of a certain depth and whose top is at flight altitude.*

**We choose to include a simple representation of contrail uplift due to the effect that this process has on contrail depth. This "vertical stretching" modifies the contrail's depth in a way which our model is able to capture, due to the interaction with wind shear, in spite of its limitations. We fully recognize that this is a highly simplified approach, as we are aiming to provide an intermediate complexity representation, and we now clarify our aims in the text (page 2, lines 95-102).**

**In terms of our specific choices of parameters, we base these on *in situ* measurements of large-scale vertical motion in an aircraft wake (Heymsfield et al, 1998; Jensen et al, 1998). This vertical motion, the result of buoyancy and the absorption of upwelling infrared radiation, was observed to last on order of one hour**

**with a peak vertical velocity of around 5 to 10 cm/s. In this study we simulate this using an initial velocity of 0-10 cm/s (varying between simulations as described in the text), specifically to identify the effect this has on our metrics of interest (see Section 3.6.1). We also apply an exponential decay for this vertical velocity, such that it falls by 40% every 30 minutes. We have added a clarifying statement at the end of Section 2.3.1 (pages 7-8, lines 25-12)**

**We are aware that the contributions of buoyancy and radiative imbalance to the vertical stretching occur on different timescales, the former being a function of the Brunt-Väisala frequency. As currently designed, APCEMM can either perform the sort of simplified simulation we outline above, or - as suggested - a plume can simply be initialized with enhanced vertical depth and no vertical movement (or indeed only a synoptic uplift, as discussed below). The inclusion of more realistic contrail initialization and dynamics is an area for future improvement as now emphasized in Sections 2.2.1 and 2.3.1 (page 4, lines 24-29; page 8, lines 8-12).**

**To be clear, we realize that our representation of contrail dynamics is highly simplified. In this paper, our goal is to enable online estimation of ice mass and surface area in a chemically-active exhaust plume, so that we can produce a first estimate of the magnitude of the contrail-induced chemical perturbation. We now clarify this in Section 2 (page 2, lines 95-101). We also state that improvements in the model's representation and calculation of plume motion should be considered if APCEMM is intended to be used for advanced contrail modeling (page 8, lines 8-12).**

*Moreover, you state in section 3.6 that you prescribe an uplift of 0 to 10 cm/s that causes the contrail to move relative to the surrounding air. What drives this contrail uplift? Only diabatic heating through radiation comes to my mind. This confuses me! Earlier on you wrote that a 10cm/s updraft during the first hour is prescribed that is supposed to cover radiative heating effects.So do you account for radiative effects twice?*

**We do not double count the radiative effects. We apply a single updraft with time-varying vertical velocity over a timescale of one hour. This is made explicit in the manuscript (page 7, lines 43-51) . The initial magnitude of this updraft was varied between 0 and 10 cm/s for different simulations (page 18, lines 75-76).**

*I could understand if you prescribed a (synoptic) uplift of the contrail together with the surrounding air. This would make sense, but surely has an opposite effect compared to your current approach.*
*In your current approach, the contrail moves out of the supersaturated into a drier layer whereas a synoptic updraught causes a cooling and hence increases the local relative humidity.*

**A synoptic uplift of the contrail with the surrounding air would indeed cause a cooling. Synoptic uplift could be modeled in APCEMM and we have added a sentence describing this at the end of Section 2.3.1 (page 7, lines 48-51). We have chosen to not include results of simulations with synoptic uplift as the focus of this paper is mostly on the plume-scale non-linear chemistry.**

*This nicely shows how relevant the exact choice of the aggregation implementation is. It does not really matter. It did not even matter that your original approach was for cloud droplets, and not for ice crystals.*
*I hope this is insightful to the author, as it reveals that simply including a complicated scheme does not necessarily give better results.*
*In carefully developing a model one should have checked results with a constant aggregation efficiency and should have compared it to the complicated approach in order to see the potential extra benefit. You should question all your design choices in the development stage.*

**We agree with the reviewer. Early-design choices in the development stage are critical. We realize that the inclusion of a sophisticated aggregation efficiency was not necessary for solid ice crystals.**

[Regarding the choice to investigate changes in the chemical composition after 24 hours] *I do not fully understand your reasoning.*

*Why is the important to have the same background conditions as at the time of the emission? If you compare the two models, it is only important that the time of the day at which the two models are evaluated are the same. Or do you somewhere subtract the base state at emission time from the 24h-state?*

**It is true that a comparison could be performed at any time of day. However, a goal of this study was to produce a model which could predict the perturbation to be added to a global chemistry-transport model (CTM) to represent an aircraft exhaust plume's products. In order to do this, we would subtract the base state from the 24-hour state exactly as outlined by the reviewer. However, we also found that 24 hours was a convenient comparison time. In addition to minimizing spurious differences when comparing to the base state, this was also a long enough period that any simulated contrails would sublimate, and the simulated chemical plume is diffuse enough that non-linear chemical phenomena such as ozone titration are no longer occurring. This motivation is outlined on page 22 (lines 65-72).**

We would like to thank you again for arranging these reviews. We look forward to your response.

Sincerely,

Sebastian Eastham

[revised manuscript text omitted]
}^{\mathrm{DE}} = \begin{cases} K_{i,j}^{\mathrm{B}} 0.45 \mathrm{Re}_j^{1/3} \mathrm{Sc}_{p,i}^{1/3} & \text{if } \mathrm{Re}_j \leq 1, r_j \geq r_i \\ K_{i,j}^{\mathrm{B}} 0.45 \mathrm{Re}_j^{1/2} \mathrm{Sc}_{p,i}^{1/3} & \text{if } \mathrm{Re}_j > 1, r_j \geq r_i, \end{cases} \quad (B2)$$

where Re and Sc are the particle Reynolds and Schmidt numbers.

The sedimentation-induced aggregation kernel is described by:

$$K_{i,j}^{\mathrm{SI}} = \mathrm{E}_{\mathrm{agg}} \pi (r_i+r_j)^2 \, | \, \mathrm{V}_{\mathrm{f},i} - \mathrm{V}_{\mathrm{f},j} \, |, \quad (B3)$$

where $\mathrm{E}_{\mathrm{agg}}$ is a collision efficiency.

The turbulent inertial motion and turbulent shear kernel are defined by:

$$K_{i,j}^{\mathrm{TI}} = \frac{\pi \varepsilon_d^{3/4}}{g \nu_a^{1/4}} (r_i+r_j)^2 \, | \, \mathrm{V}_{\mathrm{f},i} - \mathrm{V}_{\mathrm{f},j} \, | \quad (B4)$$

$$K_{i,j}^{\mathrm{TS}} = \left( \frac{8\pi \varepsilon_d}{15 \nu_a} \right)^{1/2} (r_i+r_j)^3, \quad (B5)$$

where $\varepsilon_d$ is the rate of dissipation of turbulent kinetic energy, $g$ the acceleration due to gravity and $\nu_a$ the kinematic viscosity.

The total coagulation kernel is equal to the sum of each individual kernel:

$$K_{i,j} = K_{i,j}^{\mathrm{B}} + K_{i,j}^{\mathrm{DE}} + K_{i,j}^{\mathrm{SI}} + K_{i,j}^{\mathrm{TI}} + K_{i,j}^{\mathrm{TS}}. \quad (B6)$$

**Appendix C: Plume-averaged $\mathrm{NO_x}$ chemical rate**

We assume that the conversion of $\mathrm{NO_x}$ to reservoir species is dictated by the daytime conversion pathway through the following reactions:

$$\mathrm{NO_2} + \mathrm{OH} \xrightarrow{\mathrm{M}} \mathrm{HNO_3} \quad (CR1)$$

$$\mathrm{NO_2} + \mathrm{HO_2} \xrightarrow{\mathrm{M}} \mathrm{HO_2NO_2} \quad (CR2)$$

The chemical reaction rate can be written as:

$$\frac{d[\mathrm{NO_2}]}{dt} = -(k_1[\mathrm{OH}] + k_2[\mathrm{HO_2}])[\mathrm{NO_2}]$$

$$\frac{d[\mathrm{NO_2}]}{dt} = -k_{\mathrm{eff}}[\mathrm{HO_x}][\mathrm{NO_2}]$$

where $\mathrm{HO_x}$ has been defined such that $[\mathrm{HO_x}] = \frac{k_1[\mathrm{OH}] + k_2[\mathrm{HO_2}]}{k_1+k_2}$.

We assume that the concentration field at a fixed point can be expressed as the sum of spatially-averaged quantity and the instantaneous fluctuation, such that:

$$[\mathrm{NO_2}] = \overline{[\mathrm{NO_2}]} + [\mathrm{NO_2}]'$$

$$[\mathrm{HO_x}] = \overline{[\mathrm{HO_x}]} + [\mathrm{HO_x}]'$$

The chemical conversion rate of $\mathrm{NO_x}$ can therefore be written as:

$$\frac{d\overline{[\mathrm{NO_2}]}}{dt} = -k_{\mathrm{eff}} \times \overline{[\mathrm{HO_x}][\mathrm{NO_2}]}$$

$$= -k_{\mathrm{eff}} \times \left( \overline{[\mathrm{HO_x}]} \times \overline{[\mathrm{NO_2}]} + \overline{[\mathrm{HO_x}]' \times [\mathrm{NO_2}]'} \right)$$

The first term on the right hand side leads to a net depletion. $\mathrm{NO_2}$ is an emitted species. Therefore, the $\mathrm{NO_2}$ fluctuation is positive in the core of the plume, while it is negative far away. $\mathrm{HO_x}$, however, gets depleted to form $\mathrm{HNO_3}$ and $\mathrm{HO_2NO_2}$. Therefore, $[\mathrm{
[revised manuscript text omitted]

---

## Author Response (AR3)

**Sebastian D. Eastham**
**Research Scientist,**
**Laboratory for Aviation and the Environment**
**Department of Aeronautics and Astronautics**

[Figure]

**Massachusetts Institute of Technology**
77 Massachusetts Avenue, Building 33-322A
Cambridge, Massachusetts 02139–4307, USA
http://lae.mit.edu

Atmospheric Chemistry and Physics Editorial Office

17[th] April 2020

Dear Editor,

**Re: Revisions for "The role of plume-scale processes in long-term impacts of aircraft emissions" after submission to *Atmospheric Chemistry and Physics***

Thank you again for taking the time to review our manuscript. In response to the comments of reviewer #2, we have modified Appendix D of the manuscript to try and more clearly explore the different processes which might be responsible for the discrepancies observed between APCEMM and Large Eddy Simulations (LES), as performed by Dr. Unterstraßer. We have also attempted to make it clear that this list is not exhaustive, and that additional insight could be gained through a larger intercomparison attempt.

In addition, we now include a "Code availability" section at the end of the manuscript, as requested. This directs readers to a stable location from which they can acquire the APCEMM code.

Please find below our responses (in **bold**) to the reviewer's comment (in *italics*). We have attached two copies of the manuscript: one "clean" version incorporating all changes, and a "markup" version in which all changes are highlighted. Line numbers below correspond to the "markup" version.

*Referee: 2*

Submitted on 18 Mar 2020

*I appreciate the contrail model validation in Appendix D. It is good to see contrail cross sections produced by your model. Even though the explanation in D2 reads nicely, it is pure speculation. It makes life easy if you pick one process that is included only in one of the two models and blame it for the differences in the model outputs. Your model is by design a simplifying model, uses a much coarser resolution and process description. Also the numerical schemes differ. Are you really convinced that including the missing process would drastically reduce the model differences?*

**We have expanded Appendix D2 (page 25, lines 40-85). We have tried to make it clear that the explanations we offer may explain some part of the discrepancy, but that a more comprehensive study would be needed to provide a quantitative breakdown of the causes of these differences. We also now offer some additional possible explanations for the differences between the results from APCEMM and the Large Eddy Simulations (LES) from Unterstraßer and Gierens (2010). This includes some areas where APCEMM is using a much simpler approach, such as in the representation of transport processes, and some areas where APCEMM is using a comparable but different approach such as in aerosol microphysics.**

**Most significantly we have changed the manuscript to make clear that we lack sufficient data to be able to say that adding any one missing process might be able to resolve these differences. We instead say simply that a full explanation of these differences would require a quantitative approach, ideally with the ability to run comparable simulations in both models simultaneously.**

We would like to thank you again for arranging these reviews. We look forward to your response.

Sincerely,

Sebastian Eastham

*References*

[revised manuscript text omitted]